# WHY DO TRANSFORMERS FAIL TO FORECAST TIME SERIES IN-CONTEXT?

## ABSTRACT

Time series forecasting (TSF) remains a challenging and largely unsolved problem in machine learning, despite significant recent efforts leveraging Large Language Models (LLMs), which predominantly rely on Transformer architectures. Empirical evidence consistently shows that even powerful Transformers often fail to outperform much simpler models, e.g., linear models, on TSF tasks; however, a rigorous theoretical understanding of this phenomenon remains limited. In this paper, we provide a theoretical analysis of Transformers' limitations for TSF through the lens of In-Context Learning (ICL) theory. Specifically, under AR($p$) data, we establish that: (1) Linear Self-Attention (LSA) models *cannot* achieve lower expected MSE than classical linear models for in-context forecasting; (2) as the context length approaches to infinity, LSA asymptotically recovers the optimal linear predictor; and (3) under Chain-of-Thought (CoT) style inference, predictions collapse to the mean exponentially. We empirically validate these findings through carefully designed experiments. Our theory not only sheds light on several previously underexplored phenomena but also offers practical insights for designing more effective forecasting architectures. We hope our work encourages the broader research community to revisit the fundamental theoretical limitations of TSF and to critically evaluate the direct application of increasingly sophisticated architectures without deeper scrutiny.

> "The only thing we know about the future is that it will be different."
>
> — *Peter Drucker*

## 1 INTRODUCTION

Time series forecasting (TSF), a fundamental and longstanding challenge in machine learning, involves predicting future observations based on historical data (Brockwell & Davis, 2002; Box et al., 2015; De Gooijer & Hyndman, 2006; Hamilton, 2020). TSF has broad applicability across diverse fields such as electronic health records, traffic analysis, energy consumption, and financial market predictions (Montgomery et al., 2015; Masini et al., 2023). In contrast, Transformers (Vaswani et al., 2017) have emerged as a cornerstone architecture in modern deep learning, achieving groundbreaking success across a wide array of sequence modeling tasks, including language modeling (Hurst et al., 2024; Jaech et al., 2024; Grattafiori et al., 2024; Yang et al., 2025; Guo et al., 2025), computer vision (Dosovitskiy et al., 2021; Peebles & Xie, 2023; Ma et al., 2024), visual-language modeling (Liu et al., 2023; Jin et al., 2024d), and video modeling (Deng et al., 2025; Jin et al., 2024c).

Encouraged by their remarkable performance in language modeling, substantial efforts have been dedicated to adapting Transformers and Large Language Models (LLMs) to TSF (Zhou et al., 2021; Liu et al., 2022; Wu et al., 2021; Zhou et al., 2022; Gruver et al., 2023; Cao et al., 2024; Pan et al., 2024; Jin et al., 2024a;b). Nevertheless, empirical evidence consistently reveals that Transformer-based models frequently *underperform* compared to simpler, linear forecasting methods, despite their quadratic time complexity and significantly larger parameter counts (Zeng et al., 2023; Tan et al., 2024; Eldele et al., 2024; Li et al., 2025). Such findings have prompted the development of lightweight linear models and frequency-domain approaches that typically outperform Transformers on long-horizon forecasting tasks (Xu et al., 2024; Li et al., 2023b; Yue et al., 2025; Wang et al.,

2024; Eldele et al., 2024). However, a comprehensive theoretical understanding of why Transformers exhibit such limitations remains scarce.

Existing theoretical studies on Transformer for TSF mainly relied on Neural Tangent Kernel analyses or generic In-Context Learning (ICL) theory, yielding theoretical bounds often disconnected from practical relevance and failing to provide clear representational insights (Ke et al., 2025; Cole et al., 2025; Sander et al., 2024; Li et al., 2023a; Wu et al., 2025). In sharp contrast, our work uniquely addresses the core representational limitations of Transformers through a rigorous theoretical examination grounded explicitly in classical Auto-Regressive (AR) models, fundamental frameworks dominating traditional TSF (Hamilton, 2020). By adopting Linear Self-Attention (LSA), a simplified yet powerful abstraction that eliminates the Softmax function for analytical tractability (Ahn et al., 2023; Mahankali et al., 2024; Zhang et al., 2024a; Ahn et al., 2024; Yang et al., 2024), we uncover novel and essential constraints inherent in the attention mechanism itself.

Despite AR models' inherent linearity rendering linear methods optimal, assessing the representational gap between optimal linear models and Transformers is a highly non-trivial endeavor. Under minimal assumptions—specifically, only assuming data adheres to a stable $AR(p)$ process—we establish substantial theoretical results that clearly delineate the representational boundaries of Transformers. A related setting was studied in Cole et al. (2025), which analyzes LSA on a one-dimensional linear dynamical system (a special case of AR(2)), while we generalize to $AR(p)$ under minimal stability assumptions, significantly increasing difficulty and scope. Our findings indicate that even optimally parameterized LSA Transformers cannot outperform classical linear predictors in terms of expected MSE. With infinitely long historical context their predictions can theoretically converge to those of linear regression if training is sufficiently good, yet even this convergence arises not from any structural advantage of LSA but from the inherent stability of time series, which collapses their representational space to that of linear models on AR processes. In contrast, for any finite context length there exists a provable strictly positive gap, which diminishes at a rate no faster than $1/n$ as the context length grows. We further analyze how predictions evolve and how errors accumulate under iterative Chain-of-Thought (CoT) inference.

Our theoretical analyses are complemented by empirical validations, providing practical insights into Transformer architectural design and clarifying their fundamental limitations in TSF contexts. Ultimately, our work calls for a reconsideration of the suitability and effectiveness of naively applying complex Transformer-based architectures to TSF. We advocate for deeper theoretical exploration to systematically unravel the foundational differences driving Transformers' divergent performance across domains, bridging the gap between representational capability and practical efficacy.

Our primary contributions are summarized as follows:

- We show that linear self-attention is essentially a restricted/compressed representation of linear regression, so it cannot outperform linear predictors (Section 3.1).

- We establish a strictly positive performance gap between LSA and linear predictor, and proves that this gap vanishes at a rate no faster than $1/n$ as context length increases (Section 3.2).

- We characterize prediction behavior and error compounding rate in Chain-of-Thought inference, highlighting fundamental limitations of iterative Transformer predictions (Section 3.3).

- We empirically corroborate our theoretical findings, offering insights into architectural implications and emphasizing the inherent representational limitations of Transformers for TSF (Section 4).

## 1.1 RELATED WORK

**Negative Results of Transformers on TSF.** Empirical studies consistently find Transformer- and LLM-based models struggle to surpass simpler linear baselines for time series forecasting. Language modeling components add minimal value (Tan et al., 2024), and linear variants (NLinear, DLinear) outperform Transformers on long-horizon benchmarks (Zeng et al., 2023). Analyses further question the utility of self-attention (Kim et al., 2024), show limited gains from scaling model size (Li et al., 2025), and motivate lightweight methods such as FITS (Xu et al., 2024), RLinear (Li et al., 2023b), OLinear (Yue et al., 2025), TimeMixer (Wang et al., 2024), and CNN-based TSLANet (Eldele et al., 2024). Beyond forecasting, Transformers also struggle with zero-shot temporal reasoning (Merrill et al., 2024) and anomaly detection tasks (Zhou & Yu, 2025).

Existing theoretical explanations remain limited. Kernel-based analyses attribute Transformer failures to asymmetric feature learning within a Neural Tangent Kernel regime (Jacot et al., 2018), but their synthetic assumptions and lack of universal lower bounds restrict their practical applicability (Ke et al., 2025). Similarly, recent In-Context Learning theory for linear dynamical systems provides a data-dependent lower bound largely reflecting intrinsic noise rather than representational shortcomings (Cole et al., 2025). In contrast, our work studies general AR($p$) processes, employs distinct analytic techniques, and explicitly identifies representational constraints of Transformers relative to classical linear forecasters. Due to the space constraint, we leave more related work to Appendix A.

## 2 PRELIMINARIES

**Notations.** We write $[n] := \{1, 2, \ldots, n\}$. For $a \in \mathbb{R}^d$, let $a = (a_1, \ldots, a_d)^\top$. For $X = [x_1, \ldots, x_n] \in \mathbb{R}^{m \times n}$, $x_i \in \mathbb{R}^m$ is the $i$-th column; $X_{i,:}$ and $X_{:,j}$ denote the $i$-th row and $j$-th column; $X_{a:b,:}$ and $X_{:,c:d}$ denote row/column submatrices. $\mathbf{1}_d, \mathbf{0}_d$ and $\mathbf{1}_{d \times d}, \mathbf{0}_{d \times d}$ are all-ones/all-zeros vectors and matrices; $I_d$ is the $d$-identity. $\| \cdot \|$ denotes certain norm for a vector or matrix. For symmetric $A, B \in \mathbb{R}^{d \times d}$, $A \succeq B$ ($A \succ B$) iff $A - B$ is positive semidefinite (definite). For a sequence $\{a_t\}$, $a_t \nearrow a$ means $a_t$ increases monotonically to $a$. For $A \in \mathbb{R}^{m \times n}$ and $B \in \mathbb{R}^{p \times q}$, the Kronecker product $A \otimes B \in \mathbb{R}^{mp \times nq}$ satisfies $(A \otimes B)_{(i-1)p+k,\,(j-1)q+\ell} := A_{i,j} B_{k,\ell}$ ($i \in [m]$, $j \in [n]$, $k \in [p]$, $\ell \in [q]$). For $X \in \mathbb{R}^{p \times p}$ symmetric, $\mathrm{vech}(X) \in \mathbb{R}^{p(p+1)/2}$ stacks the lower triangle (including diagonal); for $X \in \mathbb{R}^{m \times n}$, $\mathrm{vec}(X) \in \mathbb{R}^{mn}$ stacks columns.

### 2.1 TIME SERIES

We begin by formally defining the notion of time series considered in this paper.

**Definition 2.1** (Time Series). *A time series is a finite sequence of random variables $\{x_t\}_{t=1}^T$, indexed by discrete time $t \in \{1, \ldots, T\}$. We write $x_{1:T} := (x_1, \ldots, x_T)$ for the full sequence. The process is called* multivariate *if each $x_t \in \mathbb{R}^d$ with $d > 1$, and* univariate *if $d = 1$.*

In this work, we primarily focus on *univariate* Auto-Regressive processes, particularly the AR($p$) model, a cornerstone of classical time series analysis (Hamilton, 2020; Box et al., 2015).

**Definition 2.2** (AR($p$) Process (Hamilton, 2020)). *A real-valued stochastic process $\{x_i\}_{i=1}^T$ follows an autoregressive model of order $p$, denoted AR(p), if there exist coefficients $\rho_1, \ldots, \rho_p \in \mathbb{R}$ and white noise $\varepsilon_i \stackrel{i.i.d.}{\sim} \mathcal{N}(0, \sigma_\varepsilon^2)$ such that for all $i > 0$,*

$$x_{i+1} = \sum_{j=1}^p \rho_j\, x_{i-j+1} + \varepsilon_{i+1},$$

*with fixed initial values $\{x_{-p+1}, \ldots, x_0\}$. Assuming the characteristic polynomial $1 - \rho_1 z - \cdots - \rho_p z^p$ has all roots outside the unit circle, i.e. $|z| > 1$, to ensure weak stationarity, the process satisfies: (1) $\mathbb{E}[x_i] = 0$, (2) $\mathbb{E}[x_i^2] = \gamma_0$, and (3) $\mathbb{E}[x_i x_{n+1}] = \gamma_{n+1-i}$, where $\gamma_k := \mathbb{E}[x_i x_{i+k}]$ and $r_k := \gamma_k / \gamma_0$.*

Further classical results—including the ordinary least squares (OLS) solution for AR models, as well as the formulation and properties of linear predictors—are deferred to Appendix D.

### 2.2 TRANSFORMER ARCHITECTURE

For theoretical tractability, we adopt the *Linear Self-Attention (LSA)*, which omits Softmax and has been widely used in prior theoretical works (Von Oswald et al., 2023; Ahn et al., 2023; Zhang et al., 2024a; Mahankali et al., 2024; Vladymyrov et al., 2024; Gatmiry et al., 2024; Giannou et al., 2025; Sun et al., 2025; Zhang et al., 2025), with growing empirical interest (Katharopoulos et al., 2020; Schlag et al., 2021; Ahn et al., 2024; Dao & Gu, 2024; Yang et al., 2024).

**Definition 2.3** (Linear Self-Attention (LSA)). *Let $H \in \mathbb{R}^{(d+1) \times (m+1)}$ be the input matrix and define the causal mask $M := \begin{bmatrix} I_m & 0 \\ 0 & 0 \end{bmatrix} \in \mathbb{R}^{(m+1) \times (m+1)}$. We denote the attention weights $P, Q \in \mathbb{R}^{(d+1) \times (d+1)}$. Then the linear self-attention output is defined as*

$$\mathsf{LSA}(H) := H + \frac{1}{m} PHM(H^\top QH) \in \mathbb{R}^{(d+1) \times (m+1)}.$$

Throughout this paper, we focus on LSA-only Transformers.

**Definition 2.4** (*L*-Layer LSA-Only Transformer)**.** *Let* $\mathsf{LSA}_1, \ldots, \mathsf{LSA}_L$ *be a sequence of L linear self-attention layers as defined in Definition 2.3. The L-layer Transformer is defined recursively via function composition:*

$$\mathsf{TF}(H) := \mathsf{LSA}_L \circ \mathsf{LSA}_{L-1} \circ \cdots \circ \mathsf{LSA}_1(H) \in \mathbb{R}^{(d+1)\times(m+1)}.$$

### 2.3 In-Context Time Series Forecasting

Given a univariate sequence $x_{1:n}$ of AR order $p$ (Definition 2.2), we build a Hankel matrix $H_n \in \mathbb{R}^{(p+1)\times(n-p+1)}$ (Definition 2.5) whose final column is zero-padded in its last entry as a *label slot* for $x_{n+1}$. Setting $d = p$ and $m = n - p$, we feed $H_n$ into the $L$-layer LSA-only Transformer TF (Definition 2.4) and read the forecast directly from the label slot:

$$\widehat{x}_{n+1} := [\mathsf{TF}(H_n)]_{(p+1,\, n-p+1)} \in \mathbb{R}.$$

The Hankel construction (Definition 2.5) encodes the autoregressive structure in context: identifying an AR($p$) process requires at least $p$ lags, and each column of the Hankel matrix provides exactly this information. Unlike raw token sequences, the Hankel layout already fixes the relative order of observations, so it implicitly carries position encoding. This both respects the time-series autoregressive dependencies and avoids the need for additional positional embeddings that can make Transformers harder to train. The formal justification is given in Appendix E.3.

**Definition 2.5** (Hankel Matrix)**.** *For* $(x_1, \ldots, x_n) \in \mathbb{R}^n$ *and* $p \le n$, *define*

$$H_n := \begin{bmatrix} x_1 & x_2 & \cdots & x_{n-p} & x_{n-p+1} \\ x_2 & x_3 & \cdots & x_{n-p+1} & x_{n-p+2} \\ \vdots & \vdots & & \vdots & \vdots \\ x_p & x_{p+1} & \cdots & x_{n-1} & x_n \\ x_{p+1} & x_{p+2} & \cdots & x_n & 0 \end{bmatrix} \in \mathbb{R}^{(p+1)\times(n-p+1)},$$

*where each column is a sliding window of length* $p+1$, *with the last zero marking the prediction.*

## 3 Main Results

We organize our theoretical contributions into three parts. First, in Section 3.1 we provide a high-level feature-space perspective: By Hankelizing the input and analyzing the induced $\sigma$-algebra, we show that one-layer linear self-attention (LSA) effectively compresses history into a restricted cubic feature class which asymptotically collapses to the last $p$ lags, thereby anticipating a structural disadvantage relative to linear regression (LR). Building on this intuition, Section 3.2 establishes our core result: for autoregressive (AR) and more generally linear stationary processes, the optimal one-layer LSA predictor suffers a *strict finite-sample excess risk* over LR, quantified by a positive Schur–complement gap that vanishes only asymptotically at an explicit $1/n$ rate. While stacking additional LSA layers yields monotone improvements, LR remains the fundamental benchmark that cannot be surpassed. Finally, Section 3.3 turns to multistep forecasting: we prove that chain-of-thought (CoT) rollout, in stark contrast to its benefits in language tasks, compounds errors exponentially and collapses forecasts to the mean, with LSA uniformly dominated by LR at every horizon. All formal proofs are deferred to Appendices E, F, G and H. In contrast to prior work that mainly studies LR in ICL settings (Ahn et al., 2023; Zhang et al., 2024a), time series settings introduce intrinsic temporal dependencies among input variables, making the analysis substantially more complex and non-trivial.

### 3.1 Feature-Space View

**Restricted feature class.** We first reparameterize $P, Q$ in Definition 2.3 to $A, b$ to obtain the simplified form of the LSA prediction (Lemma E.3). Let $\Phi = \Phi(H_n; A, b)$ denote the one-layer LSA features induced by query–key weighting and value aggregation. The predictor admits a cubic lifting:

**Lemma 3.1** (Cubic lifting for one-layer LSA)**.** *There exist coefficients* $\{\beta_{j,r,k}\}$ *such that*

$$\widehat{x}_{n+1}^{\mathrm{LSA}}(A, b) = \sum_{j,r,k} \beta_{j,r,k}\, \varphi_{j,r,k}^{(p)}(x_{1:n}), \qquad \varphi_{j,r,k}^{(p)} \text{ are degree-3 monomials in } \{x_t\}.$$

*Hence one-layer LSA is a linear functional over a cubic feature space $\mathcal{H}_{\mathrm{LSA}}^{(p)}$.*

*Proof deferred to Appendix E.* Lemma 3.1 shows that the LSA readout lives in a *cubic* feature space of the raw inputs. By the $L_2$-projection property of conditional expectation (orthogonality onto sub-$\sigma$-algebras), we obtain Proposition 3.2: any predictor operating on $\Phi$ cannot achieve lower MSE than the optimal predictor operating on the full context $x_{1:n}$. In short, the attention-derived representation is *informationally coarser* than the raw context; adding architectural complexity does not reveal additional predictive signal beyond the last $p$ lags, but only reweights existing information.

**Proposition 3.2** (Information monotonicity). $\sigma(\Phi) \subseteq \sigma(x_{1:n})$. *Hence, by conditional orthogonality/Jensen,*

$$\inf_f \ \mathbb{E}\big[(x_{n+1} - f(x_{1:n}))^2\big] \ \leq \ \inf_g \ \mathbb{E}\big[(x_{n+1} - g(\Phi))^2\big].$$

*Proof deferred to Appendix E.* Furthermore, Proposition 3.3 formalizes the asymptotic picture: as the number of observed contexts $n \to \infty$, the $(p+1)$-row Hankel design ensures access to exactly the $p$ relevant lags; by ergodicity, empirical Hankel Gram blocks converge to their Toeplitz limits, cross-row correlations stabilize, and the cubic coordinates concentrate onto the last-$p$-lag subspace. Consequently, one-layer LSA *cannot outperform* LR and can at best *match it asymptotically* (Proposition E.14). See Appendix E for an optimal LSA parameter choice achieving this limit.

**Proposition 3.3** (Asymptotic collapse of LSA features). *As $n \to \infty$ for a stable AR(p), the coordinates $\varphi_{j,r,k}^{(p)}(x_{1:n})$ converge in $L_2$ to scaled copies of $\{x_{n-p+1}, \ldots, x_n\}$. Thus the optimal one-layer LSA readout asymptotically reduces to a linear function of the last $p$ lags.*

> **Takeaway 1: Attention is Not All You Need for TSF**
>
> **Feature-Space.** LSA operates on a strictly coarser $\sigma$-algebra than the raw context; it can at best reweight the last-$p$ lags and cannot unlock signal beyond them. For time series with pronounced linear structure (e.g., AR/ARMA), this is a *fundamental representational limitation*. This aligns with empirical findings that simple linear models often outperform Transformers in TSF (Zeng et al., 2023; Kim et al., 2024; Tan et al., 2024).

### 3.2 A Strict Finite-Sample Gap (Core Result)

In this section we rigorously establish and quantitatively characterize the *fundamental representational limitation* of one-layer LSA. Our main technical device is a *Kronecker-product lifting* of the Hankel-derived features: the one-layer LSA readout can be written as

$$\widehat{x}_{n+1}^{\mathrm{LSA}} = \widetilde{\eta}^\top Z, \qquad Z := \big(\operatorname{vech} G\big) \otimes x, \ \ x := x_{n-p+1:n} \in \mathbb{R}^p,$$

where $G = G(x_{1:n})$ is the Hankel Gram matrix, $\operatorname{vech}(\cdot)$ denotes half-vectorization, and $\widetilde{\eta}$ is an affine reparameterization of $(A, b)$ (see Appendix F.2). This lifts the *cubic* dependence of LSA into an ordinary *linear regression* in the lifted space. Let $Z \in \mathbb{R}^{qp}$ with $q := \dim(\operatorname{vech} G)$. We further define $\widetilde{S} := \mathbb{E}[ZZ^\top]$, $\widetilde{r} := \mathbb{E}[Zx^\top]$, $\Gamma_p := \mathbb{E}[xx^\top]$, and the induced Schur complement is $\Delta_n := \Gamma_p - \widetilde{r}^\top \widetilde{S}^{-1} \widetilde{r}$. Intuitively, $\Delta_n$ captures the component of the linear signal in the last $p$ lags that remains *orthogonal* to the span of lifted LSA features—namely, the exact *representation gap* characterized in Theorem 3.4. Moreover, we prove in Theorem 3.4 that $\Delta_n \succ 0$ for any finite $n$, establishing a strict finite-sample gap. We then derive an explicit first-order expansion $\Delta_n = \frac{1}{n} B_p + o(1/n)$ under Gaussianity in Theorem 3.5, and show in Theorems H.3 and H.5 and Lemma H.4 that both the strictness and the $1/n$ rate persist for general linear stationary processes, with the leading constant adjusted by cumulant spectra (Appendix H).

**Theorem 3.4** (Strict finite-sample gap: AR(p)). *For any $n \geq p$ and stable AR(p),*

$$\min_{A,b} \mathbb{E}\big[(\widehat{x}_{n+1}^{\mathrm{LSA}} - x_{n+1})^2\big] \ \geq \ \min_w \mathbb{E}\big[(w^\top x_{n-p+1:n} - x_{n+1})^2\big] \ + \ \rho^\top \Delta_n \rho, \qquad \Delta_n \succ 0.$$

**What this says.** Even after optimizing over all one-layer LSA parameters, the *best-in-class* LSA risk is strictly larger than the *best-in-class* linear risk by the explicit quadratic form $\rho^\top \Delta_n \rho$; the gap is *structural* (positive definite), not an estimation or optimization artifact.

*Proof Sketch.* (i) Since the lifted loss is a strictly convex quadratic in $\widetilde{\eta}$, it has the unique minimizer $\widetilde{\eta}^* = \widetilde{S}^{-1}\widetilde{r}, \rho$, yielding the class optimum $\min_{\widetilde{\eta}} \mathcal{L}(\widetilde{\eta}) = \sigma_\varepsilon^2 + \rho^\top \Delta_n \rho$. (ii) Under AR($p$), the optimal linear predictor attains the Bayes one-step risk $\sigma_\varepsilon^2$, so the excess risk equals $\rho^\top \Delta_n \rho$. (iii) Strictness follows from block positive definiteness of the joint covariance

$$\mathbb{E}\left[\begin{pmatrix} Z \\ x \end{pmatrix} \begin{pmatrix} Z \\ x \end{pmatrix}^\top\right] = \begin{pmatrix} \widetilde{S} & \widetilde{r} \\ \widetilde{r}^\top & \Gamma_p \end{pmatrix} \succ 0.$$

Specifically, we reduce the claim to showing that the (non-lifted) joint covariance of $(\operatorname{vech} G, x)$ is positive definite. We prove this via an innovation-based elimination from the newest to older indices, establishing that no nontrivial linear combination can have zero variance. Using the Schur-complement property then yields $\Delta_n \succ 0$. All blocks admit closed forms as Hankel–Toeplitz moments up to orders 4 and 6 (from earlier definitions of $\widetilde{S}, \widetilde{r}$ in this section). Hence, $\Delta_n$ is *computable* from process moments; in the Gaussian case these moments follow from Isserlis' theorem. A warm-up AR(1) calculation is given in Appendix F.1. See Appendix F.2 for the complete proof. □

**First-order gap rate under Gaussianity.** We now quantify the finite-$n$ excess risk in Theorem 3.4. For zero-mean stationary Gaussian AR($p$), we expand the lifted moments around their population (rank-one) limit and evaluate the Schur complement via a singular block inverse.

**Theorem 3.5** (Explicit $1/n$ rate: Gaussian)**.** *For Gaussian AR(p),*

$$\Delta_n = \Gamma_p - \widetilde{r}_n^\top \widetilde{S}_n^{-1} \widetilde{r}_n = \frac{1}{n}B_p + o(1/n), \qquad B_p \succeq 0 \text{ (generically } B_p \succ 0\text{)}.$$

*Consequently, for any fixed $\|\rho\| \geq r > 0$ there exists $c_r > 0$ such that*

$$\min_{A,b} \mathbb{E}[(\widehat{x}_{n+1}^{\mathrm{LSA}} - x_{n+1})^2] \geq \min_w \mathbb{E}[(w^\top x_{n-p+1:n} - x_{n+1})^2] + \frac{c_r}{n}.$$

*Proof Sketch.* The argument proceeds in two steps: *(1) Moment expansions.* We first expand the lifted covariance quantities using standard tools (Isserlis' theorem for Gaussian moments and Toeplitz summation for time averages). This shows that the main term has a simple block-diagonal structure determined entirely by the process autocovariances, while the finite-sample corrections appear at order $1/n$. *(2) Singular block inversion.* Because part of the leading block vanishes in the limit, the inverse matrix develops a component that scales linearly with $n$. Applying a first-order block-inverse expansion reveals that the excess-risk matrix $\Delta_n$ itself scales like $1/n$, with a computable correction term depending on the second-order moment structure of the process.

In short, the proof shows that after separating the dominant block structure and carefully controlling the inverse, the residual term $\Delta_n$ has an explicit $1/n$-expansion governed entirely by process moments. A complete proof is provided in Appendix F.3. □

**Remark 3.6** (Why the rate is $1/n$)**.** *Let $u = \operatorname{vech}(\Gamma_{p+1})$. At the population limit $\widetilde{S}_\infty = (uu^\top) \otimes \Gamma_p$ is rank-one along $u$. Finite $n$ introduces $\Theta(1/n)$ perturbations that regularize the orthogonal directions, so the Schur complement $\Gamma_p - \widetilde{r}_n^\top \widetilde{S}_n^{-1}\widetilde{r}_n$ is $\Theta(1/n)$. The overlap of Hankel windows is the source of these first–order terms.*

**Beyond Gaussianity: strict gap and $1/n$ rate.** We work under *linear stationarity* with Wold representation $x_t = \sum_{k \geq 0} \psi_k \varepsilon_{t-k}$, $\sum_k |\psi_k| < \infty$, and i.i.d. *symmetric* innovations $\{\varepsilon_t\}$ with $\mathbb{E}[\varepsilon_t] = 0$, possessing finite fourth and sixth moments. Replacing Isserlis' theorem by the moment–cumulant formula preserves positive definiteness of the joint covariance, so the **strictly positive** gap still holds (Theorem H.3); furthermore, the convergence rate remains $1/n$ via the non-Gaussian expansions (Lemma H.4) culminating in the rate result (Theorem H.5).

**Multi-layer LSA yields monotone improvement.** Because the stacked Kronecker feature set enlarges with depth, the optimal LSA risk is monotone nonincreasing in $L$ by projection isotonicity (Proposition F.18, using the Moore–Penrose formulation Equations (12) and (13) and the residual-covariance view Lemma F.17). Moreover, if the $(L+1)$-st layer contributes any $L_2$-nonredundant

direction—equivalently $\mathrm{Var}\big(P_{\mathcal{H}_L^{\perp}}(g^{(L)} \otimes x)\big) > 0$ for $\mathcal{H}_L = \mathrm{span}\{g^{(0)} \otimes x, \ldots, g^{(L-1)} \otimes x\}$—then the Moore–Penrose Schur complement strictly decreases (Proposition F.18), yielding

$$\min_{\{b^{(\ell)}, A^{(\ell)}\}_{\ell=0}^{L}} \mathbb{E}\big[(\widehat{x}_{n+1}^{(L+1)} - x_{n+1})^2\big] \;<\; \min_{\{b^{(\ell)}, A^{(\ell)}\}_{\ell=0}^{L-1}} \mathbb{E}\big[(\widehat{x}_{n+1}^{(L)} - x_{n+1})^2\big].$$

**Remark 3.7.** *A potential misinterpretation is that we claim Transformers cannot solve or fit time series at all. This is NOT the case. Rather, our result shows that Transformers cannot outperform simple linear models beyond a certain extent. While they may be capable of solving time series tasks to some degree, their performance does not substantially exceed that of linear models.*

> **Takeaway 2: Strictly Positive Gap between LSA and Linear Predictor**
>
> **Strictness.** For any finite $n$, one-layer LSA has a *strict* excess risk over $p$-lag LR equal to $\rho^{\top} \Delta_n \rho$ with $\Delta_n \succ 0$ (Theorem 3.4).
> **Rate.** The gap admits an explicit $1/n$ expansion with PSD leading constant $B_p$ (generically PD) (Theorem 3.5).
> **Robustness.** Under linear stationarity with finite moments, strictness and the $1/n$ rate persist; non-Gaussianity only alters the constant via cumulants (Theorems H.3 and H.5 and Lemmas H.4 and H.6).
> **Depth.** Stacking layers enlarges the feature span, so risk is *monotone nonincreasing* in $L$; generically it *strictly* improves when the new layer adds a nonredundant direction (Proposition F.18), yet the LR baseline remains unbeatable at finite $n$ (Equations (12) and (13)).

### 3.3 Chain-of-Thought Rollout: Multi-Step Collapse

In this section, we show that under CoT-style inference, LSA collapses to the mean with an error rate that grows exponentially. We first define CoT in Definition 3.8.

**Definition 3.8** (Chain-of-Thought (CoT) Inference)**.** *Given a time series $(x_1, \ldots, x_n)$ and context length $p$, initialize the Hankel matrix $H_n \in \mathbb{R}^{(p+1) \times (n-p+1)}$ as in Definition 2.5 with the last column zero-padded. Let* TF *be the $L$-layer LSA-based Transformer in Definition 2.4. For each step $t = 1, 2, \ldots, T$:*

1. *Predict the next value: $\widehat{x}_{n+t} := [\mathsf{TF}(H_{n+t-1})]_{(p+1,\, n-p+t)}$.*

2. *Overwrite the zero in the last column of $H_{n+t-1}$ with $\widehat{x}_{n+t}$.*

3. *Append the column $(x_{n-p+t+1}, \ldots, x_n, \widehat{x}_{n+1}, \ldots, \widehat{x}_{n+t})^{\top}$ to form $H_{n+t}$ with last entry set to $0$.*

*Repeating yields CoT rollouts $\widehat{x}_{n+1}, \ldots, \widehat{x}_{n+T}$ by feeding model outputs back into the Hankel input.*

**Theorem 3.9** (Collapse and error compounding for CoT)**.** *Under AR($p$), the Bayes $h$-step forecast equals the noise-free recursive rollout of the one-step Bayes predictor. Any stable linear CoT recursion $\widehat{s}_{t+1} = A(w)\widehat{s}_t$ collapses exponentially to $0$. For Bayes $w = \rho$,*

$$\mathrm{MSE}^*(h) = \mathbb{E}[(x_{n+h} - \widehat{x}_{n+h}^*)^2] = \sigma_{\varepsilon}^2 \sum_{k=0}^{h-1} \psi_k^2 \nearrow \mathrm{Var}(x_t), \quad \mathrm{Var}(x_t) - \mathrm{MSE}^*(h) \leq \frac{C^2 \sigma_{\varepsilon}^2}{1-\beta^2} \beta^{2h}.$$

*Thus, even for the optimal predictor, CoT error compounds to the unconditional variance at an exponential rate governed by the spectrum of $A(\rho)$. Here $\beta < 1$ and $C > 0$ are constants depending only on the AR($p$) process.*

*Proof deferred to Appendix G.* Because conditional expectation is the $L_2$ projection, for any measurable $g = g(x_{1:n})$ (including any $L$-layer LSA CoT rollout) one has

$$\mathbb{E}[(x_{n+h} - g)^2] = \mathrm{MSE}^*(h) + \mathbb{E}[(\widehat{x}_{n+h}^* - g)^2] \;\geq\; \mathrm{MSE}^*(h),$$

with strictness unless $g \equiv \widehat{x}_{n+h}^*$ a.s. (Equations (16) and (17)). Since one-layer LSA already has a strict finite-sample gap at $h = 1$ (Theorem 3.4), equality fails generically for all $h$.

**Corollary 3.10** (Earlier failure of LSA CoT)**.** *Define the failure horizon $H_{\tau}(g) := \inf\{h \geq 1 : \mathbb{E}[(x_{n+h} - g_h)^2] \geq \tau \mathrm{Var}(x_t)\}$. Then $H_{\tau}(\widehat{x}^{\mathrm{LSA}}) \leq H_{\tau}(\widehat{x}^*)$ for all $\tau \in (0, 1)$, with strict inequality on a set of $\tau$ of positive measure. In words: LSA CoT reaches the large-error regime no later than (and generically earlier than) Bayes linear regression.*

*Proof deferred to Appendix G.* Quantitatively, $\text{Var}(x_t) - \text{MSE}^{\text{LSA}}(h) \leq \text{Var}(x_t) - \text{MSE}^*(h)$, so whenever the gap to variance remains positive, LSA's CoT error approaches $\text{Var}(x_t)$ at least exponentially fast; if it overshoots (the left side becomes negative), Corollary 3.10 still guarantees earlier threshold crossing. For AR(1), closed forms make the compounding explicit: $\text{MSE}^*(h) = \sigma^2(1 - \rho^{2h})$ with half-life $h_{1/2} = \log(1/2)/\log(\rho^2)$.

> **Takeaway 3: CoT Collapse in TSF**
>
> **TSF.** CoT rollout forms a stable linear dynamical system that *collapses to the mean* and whose error *compounds exponentially* to $\text{Var}(x_t)$; Bayes/LR is horizonwise optimal and LSA CoT is uniformly dominated at each horizon (Proposition G.1, Lemma G.2, Theorem G.3, Equation (17), and Corollary G.4).
>
> **Contrast.** Notably, CoT behaves very differently in TSF compared to other domains: in language tasks, test-time scaling shows longer inference chains *improve* problem solving (Li et al., 2024; Snell et al., 2025), and CoT can help in in-context linear regression (Huang et al., 2025); in stark contrast, in TSF, CoT leads to *rapidly compounding* forecast errors.

## 4 NUMERICAL VERIFICATION

We defer experimental details, including datasets, model configurations, and the Softmax attention vs. LSA comparison, to Appendix C.

### 4.1 EVALUATION

To comprehensively assess model performance in TSF, we employ two complementary inference modes for evaluation and visualization:

- **Teacher-Forcing (TF) TSF:** This method evaluates the model under idealized conditions by providing ground-truth historical values as inputs at *each* time step. It is commonly used to measure predictive accuracy and to visualize the model's capacity to fit the true data distribution.

- **Chain-of-Thought (CoT) TSF:** This iterative inference approach simulates real-world deployment by using the model's own past predictions as inputs for future steps. It enables the evaluation of long-horizon stability and the extent of error accumulation during rollout.

**Evaluation Metrics.** Let $\{x_1, x_2, \ldots, x_T\}$ denote a ground-truth test time series and $\{\widehat{x}_1, \widehat{x}_2, \ldots, \widehat{x}_T\}$ the corresponding model predictions. We evaluate forecasting accuracy using the Mean Squared Error (MSE): $\text{MSE} := \frac{1}{T} \sum_{t=1}^{T} (x_t - \widehat{x}_t)^2$. In rollouts where predictions are generated via TF or CoT, we further compute the *cumulative MSE* up to step $k$: $\text{CumMSE}(k) := \frac{1}{k} \sum_{t=1}^{k} (x_t - \widehat{x}_t)^2, k \in [T]$. This captures how errors accumulate as inference progresses.

### 4.2 EXPERIMENTS

For all experiments, we adopt a common set of hyperparameters. Synthetic AR series are generated with Gaussian noise of zero mean and standard deviation $\sigma_\varepsilon = 0.05$ and total length 50,000, split into training/validation/test with proportions $0.70/0.15/0.15$. Models are trained on 10 RTX 2080 GPUs for up to 100 epochs, using a batch size of 512 and the Adam optimizer with learning rate $10^{-3}$.

**Teacher Forcing vs. Chain-of-Thought.** We compare 50-step predictions under Teacher-Forcing (TF) and Chain-of-Thought (CoT) rollout (Section 4.1) using a one-layer LSA with $n = 8$ on AR(5). Figures 1(a–b,c–d) show trajectories and cumulative MSEs. Under TF, both LSA and OLS track the AR(5) process, but OLS consistently yields lower MSE, indicating LSA fits yet never exceeds the linear baseline. Under CoT rollout, errors accumulate: both methods collapse to the mean, with LSA failing earlier, consistent with Section 3.3.

**Context and Layers Scaling.** We vary history length $n$ and LSA layers $L$ with $p \in \{3, 5, 7\}$, averaging over 5 seeds with 95% CIs. For context scaling, $n = p + \{5, 25, 50, 100, 200\}$ with one LSA layer; for layer scaling, $L \in \{1, \ldots, 6\}$ with $n = 100$. Figures 1(e–f) show Teacher-Forcing

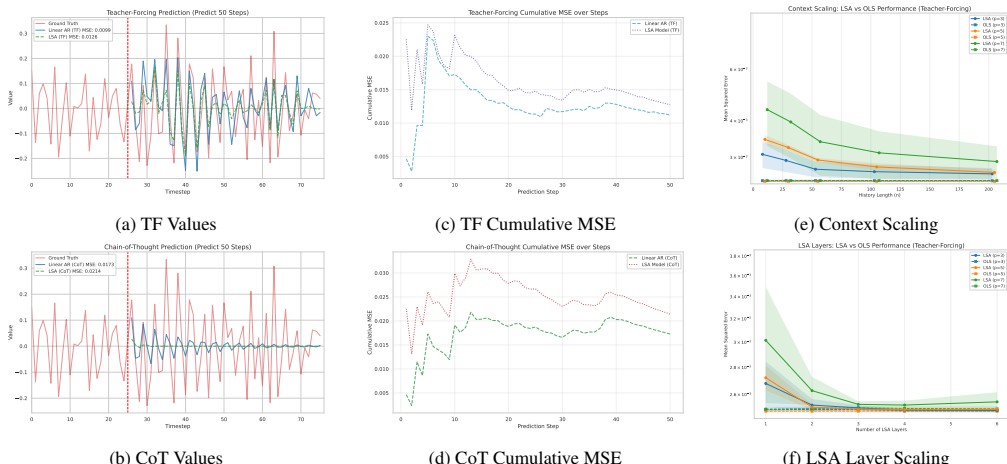

(a) TF Values      (c) TF Cumulative MSE      (e) Context Scaling

(b) CoT Values      (d) CoT Cumulative MSE      (f) LSA Layer Scaling

Figure 1: Experimental results. **(a–b)** Predictions under Teacher-Forcing (TF) and Chain-of-Thought (CoT). **(c–d)** Cumulative MSE for TF and CoT rollouts. **(e–f)** Scaling experiments varying the history length and the number of LSA layers. Overall, LSA tracks $AR(p)$ but never surpasses the OLS baseline, confirming its representational limits.

MSE: larger $n$ improves LSA but never closes the gap to OLS, and more layers give diminishing gains saturating at the OLS level, consistent with Section 3.2.

## 5 DISCUSSION

**Architectural Considerations.** Beyond LSA, several components may influence Transformer performance. The Softmax in standard attention may enhance expressivity by exponentiating inputs, effectively expanding the representation space and approximating an infinite series to capture richer dependencies (see experimental results in Appendix C). Given the limitations of single-head LSA, simply aggregating multiple heads over the same data source is unlikely to improve expressivity; however, allocating different heads to distinct modalities may offer benefits through data fusion. Additionally, the role of feedforward MLP layers deserves closer scrutiny. Although not the focus of our analysis, prior work (Zeng et al., 2023; Tan et al., 2024; Kim et al., 2024) suggests that MLPs play as key contributors in time series tasks—potentially explaining the performance of LLMs in TSF. We leave these directions to future work.

**Difference Between Language and Time Series.** Attention serves as a learned compression mechanism, essential in language modeling where meaning depends on long-range, abstract dependencies (Delétang et al., 2024; Sutskever, 2023; Goldblum et al., 2024; Huang et al., 2024b). In contrast, time series with low-order dynamics (e.g., $AR(p)$) hinge on local, position-specific patterns, where such compression can obscure predictive signals. Because attention applies fixed contextual weights, it often fails to capture these direct dependencies, explaining the locality-agnostic behavior noted in (Li et al., 2019). When compression is misaligned with the data-generating process, classical models typically outperform attention.

## 6 CONCLUSION

We study the limits of Transformers in time-series forecasting via in-context learning theory. For autoregressive (AR) processes, we prove that Linear Self-Attention (LSA) cannot beat the optimal linear predictor, yielding a strictly positive gap in expectation. Our analysis further shows that although LSA asymptotically recovers the linear predictor under teacher-forcing, errors compound under Chain-of-Thought rollout, ultimately causing collapse-to-mean behavior. Experiments across teacher-forcing, CoT, and scaling corroborate our theory: LSA matches but never exceeds the linear baseline. These findings clarify the inherent limits of attention in time-series forecasting and highlight the need for architectures beyond self-attention to capture temporal structure.

## LLMs Usage Statement

We disclose that LLMs were employed exclusively as auxiliary tools, limited to refining the exposition of the manuscript for clarity and conciseness.

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

CONTENTS

# Appendix

**Roadmap.** In Appendix A, we review related work on Transformers for time series forecasting, in-context learning theory, and representational limitations of attention. Appendix B provides further discussion on our perspective of TSF. In Appendix C, we include more experiment details. Appendix D reviews classical results on time series. Appendix E analyzes the expressivity of LSA Transformers for TSF. Appendix F establishes the finite-sample gap between LSA and linear models, with detailed proofs. Appendix G extends our analysis to Chain-of-Thought rollout, characterizing collapse-to-mean and error compounding. Finally, Appendix H relaxes Gaussian assumptions and generalizes our results to linear stationary processes.

## A MORE RELATED WORK

### A.1 TRANSFORMERS FOR TIME SERIES FORECASTING

Early adaptations of Transformers for time series forecasting primarily modified attention mechanisms to capture long-term dependencies efficiently. Informer introduced ProbSparse attention to mitigate quadratic complexity (Zhou et al., 2021), while Pyraformer employed hierarchical pyramidal attention for multi-scale modeling (Liu et al., 2022). Autoformer and FEDformer further integrated domain-specific inductive biases, utilizing auto-correlation and frequency decomposition, respectively, to model seasonal-trend components explicitly (Wu et al., 2021; Zhou et al., 2022). Additional variants include locality-enhanced attention (Li et al., 2019), inverted architectures (Liu et al., 2024b), and tokenization-based representations treating sequences as textual patches (Nie et al., 2023). A comprehensive overview is presented in (Wen et al., 2023).

More recent studies leverage pretrained Large Language Models (LLMs) to transfer NLP-style capabilities to forecasting tasks. Zero-shot forecasting was initially demonstrated using pretrained LLMs without task-specific tuning (Gruver et al., 2023). Subsequent works explored specialized prompt-based strategies (Cao et al., 2024; Pan et al., 2024), reprogramming pretrained LLMs directly (Jin et al., 2024a), and unified, dataset-agnostic training paradigms (Woo et al., 2024; Lu et al., 2025). Broader frameworks proposed foundational-model perspectives for time series tasks (Goswami et al., 2024), discrete vocabulary tokenization (Ansari et al., 2024), multi-patch prediction (Bian et al., 2024), and generalized decoder-only architectures (Das et al., 2024; Zhou et al., 2023; Liu et al., 2024c). Recent surveys systematically summarize these emerging paradigms and highlight open challenges and opportunities in deploying LLMs for time series analysis (Zhang et al., 2024b; Liang et al., 2024; Liu et al., 2025; Jin et al., 2024b).

### A.2 IN-CONTEXT LEARNING THEORY

Recent work theoretically interprets in-context learning (ICL) as a Transformer forward pass implicitly performing variants of gradient descent (GD). Early studies empirically demonstrated Transformers can closely approximate ordinary-least-squares predictors (Garg et al., 2022), while subsequent constructive analyses showed one linear self-attention (LSA) layer corresponds exactly to one GD step, with the global training objective implementing a preconditioned, Bayes-optimal GD step (Von Oswald et al., 2023; Akyürek et al., 2023; Ahn et al., 2023; Mahankali et al., 2024). Further training-dynamics analyses establish gradient flow convergence of LSA to learn the class of linear models (Zhang et al., 2024a), provide finite-time convergence guarantees and parameter evolution for multi-head Softmax attention (Huang et al., 2024a; He et al., 2025), and identify phase transitions revealing when linear-attention mimics full Transformer behaviors (Ahn et al., 2024; Zhang et al., 2025). Extensions include multi-step GD via chain-of-thought prompting (Huang et al., 2025) and kernelized polynomial regression through gated linear units (Sun et al., 2025). Other works establish positive approximation guarantees for ICL in dynamical and autoregressive settings but lack universal lower bounds or explicit representational constraints (Li et al., 2023a; Sander et al., 2024; Wu et al., 2025; Cole et al., 2025).

## A.3 REPRESENTATIONAL LIMITATIONS OF TRANSFORMERS

Despite their success, Transformers exhibit fundamental limitations in expressivity. Pure self-attention without MLPs suffers doubly exponential rank collapse with depth, severely constraining representational capacity (Dong et al., 2021). Self-attention cannot model periodic finite-state languages unless the depth or number of heads scales with input length (Hahn, 2020). Complexity analyses show that log-precision Transformers are no more powerful than $TC^0$ circuits, implying provable failure on linear systems and context-free languages under standard complexity separations (Merrill & Sabharwal, 2023). Communication-complexity arguments further reveal that Transformers cannot compose functions over sufficiently large input domains (Peng et al., 2024), and their performance is task-dependent, achieving logarithmic complexity in input size for sparse averaging tasks, but requiring linear complexity for triple-detection (Sanford et al., 2023). (Chen et al., 2024) establish unconditional depth–width trade-offs, proving that solving sequential $L$-step function composition tasks over input of $n$ tokens requires either $\Omega(L)$ layers or $n^{\Omega(1)}$ hidden dimensions. Empirically, Transformers struggle with compositional generalization (Dziri et al., 2023) and fail to outperform RNNs in modeling Hidden Markov dynamics (Hu et al., 2024a). While chain-of-thought prompting and positional embeddings can recover arithmetic and step-wise reasoning (Feng et al., 2023; McLeish et al., 2024), these function as external aids, underscoring the architecture's inherent limitations.

## B  MORE DISCUSSION

**Other Data Distributions.**    Although our analysis focuses on AR processes, similar considerations apply more broadly. The difficulty for attention arises from its misalignment in capturing input dependencies, as seen in AR($p$). We further discuss Moving Average (MA) processes and, more generally, ARMA models in Appendix H.

**Multivariate Time Series.**    In the case of uncorrelated multivariate time series, each dimension evolves independently, reducing training to separate LSA models. This eliminates the opportunity to exploit cross-variable dependencies and limits the potential of learning shared structure. Consequently, pretraining on a collection of uncorrelated time series may fail to produce useful shared representations. For the correlated case, one could impose structural assumptions on inter-variable dependencies to enable tractable analysis. We leave the investigation of such multivariate models, such as Vector Auto-Regression (VAR) (Hamilton, 2020), to future work.

**Optimization Dynamics.**    While our analysis primarily addresses representational limitations of attention mechanisms, future work could explore optimization dynamics and training difficulties of Transformers for time series forecasting. Understanding these issues might yield complementary insights into observed empirical shortcomings.

**Real-World Complexities.**    Real-world forecasting tasks include many complexities not modeled in this study, such as data randomness, intricate temporal dependencies, training instability, noisy signals, and external factors like market sentiment (Liang et al., 2024; Liu et al., 2025). Exploring these practical challenges could help bridge theoretical findings with real-world performance.

**Architectural and Framework Varieties.**    Our framework intentionally abstracts away practical components such as Rotary Position Embeddings (RoPE) (Su et al., 2024) and Mixture-of-Experts (MoE) (Fedus et al., 2022). Future work may assess whether these enhancements alleviate the representational limitations we identify. State Space Models like Mamba (Gu & Dao, 2024; Dao & Gu, 2024) also present a promising alternative for time series forecasting (Wang et al., 2025). Beyond architectural changes, generative modeling paradigms such as diffusion models (Ho et al., 2020) and flow matching (Lipman et al., 2023) offer alternative approaches for time series (Tashiro et al., 2021; Yuan & Qiao, 2024; Hu et al., 2024b; Liu et al., 2024a; Fan et al., 2024), potentially overcoming the limitations of Transformer-based Next-Token Prediction objectives.

## C  EXPERIMENTAL DETAILS

This section provides additional details for Section 4.

## C.1   DATASET AND MODEL CONFIGURATION

**Synthetic data.**   We generate *stable* AR($p$) processes (Definition 2.2) by sampling coefficient vectors (roots outside the unit circle), adding Gaussian noise, discarding a short burn-in, and retaining a long sequence. Each sequence is split into train/validation/test segments.

**From long sequences to training examples.**   We fix a history length $n > p$. For each series $x_{1:T}$, a sliding window of length $n$ with stride 1 defines training pairs with input history $x_{t-n+1:t}$ and target $x_{t+1}$. Each history is transformed into a Hankel matrix $H_n^{(t)} \in \mathbb{R}^{(p+1)\times(n-p+1)}$ (Definition 2.5), which serves as the input to the LSA model.

**Models.**   Our main model is an $L$-layer LSA-only Transformer TF (Definition 2.4) with feature dimension $d = p$. We read prediction from the label slot: $\widehat{x}_{t+1} = \big[\mathsf{TF}(H_n^{(t)})\big]_{(p+1,n-p+1)}$. As a baseline, we fit a classical AR($p$) predictor by OLS on the same training series used for LSA.

**Training.**   All windows are shuffled and batched. Models are trained with teacher forcing using MSE loss and Adam. We sweep $p$, $n$, and $L$, while keeping the noise level and optimization hyperparameters fixed. Performance is reported on the held-out test split.

## C.2   SOFTMAX ATTENTION VS. LSA

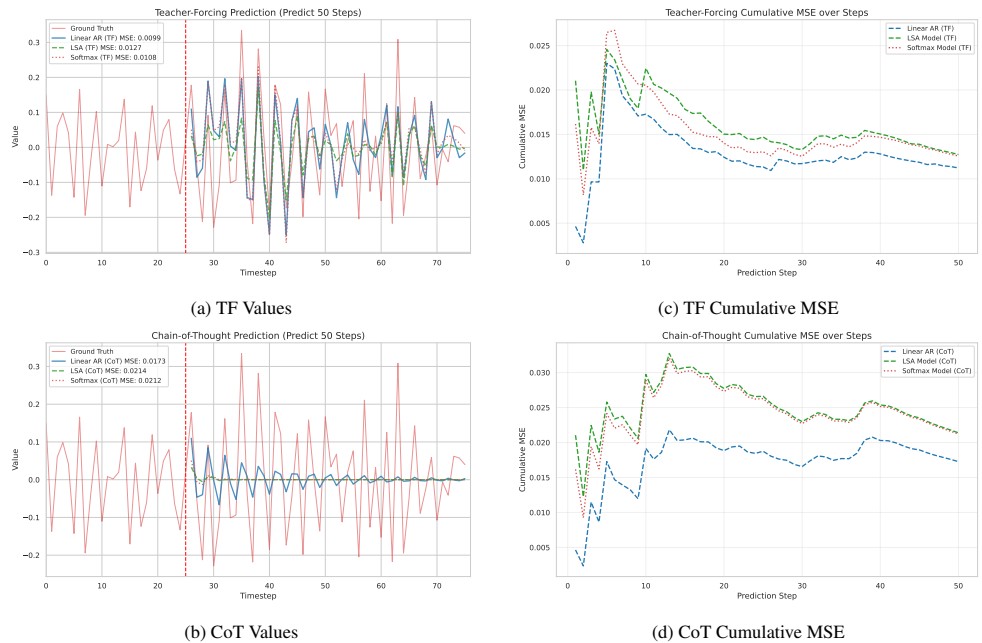

(a) TF Values

(c) TF Cumulative MSE

(b) CoT Values

(d) CoT Cumulative MSE

Figure 2: Experimental results on comparison of LSA and Softmax Attention. **(a–b)** Predictions under Teacher-Forcing (TF) and Chain-of-Thought (CoT). **(c–d)** Cumulative MSE for TF and CoT rollouts. Overall, both LSA and Softmax Attention tracks AR($p$) but never surpass the OLS baseline. Moreover, Softmax Attention is slightly better than LSA.

Though theoretically more challenging, we could examine the experimental behavioral comparison between Softmax attention and LSA. We define Softmax attention as follows:

**Definition C.1** (Softmax Attention). *Let $H \in \mathbb{R}^{(d+1)\times(m+1)}$ be the input matrix and define the causal mask*

$$M := \begin{bmatrix} I_m & 0 \\ 0 & 0 \end{bmatrix} \in \mathbb{R}^{(m+1)\times(m+1)}.$$

We denote the reparameterized weights $P, Q \in \mathbb{R}^{(d+1)\times(d+1)}$. Then the (masked) softmax attention output is defined as

$$\mathsf{Attn}(H) := PHM\,\mathsf{Softmax}\big(H^\top QH\big),$$

where the $\mathsf{Softmax}(\cdot)$ is applied column-wise to ensure attention weights are normalized. Thus $\mathsf{Attn}(H) \in \mathbb{R}^{(d+1)\times(m+1)}$.

We compare 1-layer LSA and softmax attention models under identical settings (Adam optimizer and hyperparameters as in Section 4), differing only in architecture. Empirically, Softmax Attention performs slightly better than LSA (Figure 2), which is unsurprising given the greater expressivity underlying the strong performance of Transformers. However, the analytical complexity of the Softmax operator makes theoretical understanding more challenging, and we leave a deeper investigation of this gap to future work.

## D  TIME SERIES FUNDAMENTALS

We first recall classical results for stationary autoregressive (AR) processes, which form the theoretical backbone for our later analysis. These results connect population-level covariances to model coefficients, show how consistent estimates can be obtained from finite data, and establish a natural linear performance baseline.

**Definition D.1** (Yule–Walker Equations (Hamilton, 2020)). *Let $\{x_i\}_{i=1}^T$ follow a stationary AR(p) process as in Definition 2.2. Define the Toeplitz autocovariance matrix $\Gamma_p \in \mathbb{R}^{p\times p}$ as*

$$\Gamma_p := \begin{pmatrix} \gamma_0 & \gamma_1 & \cdots & \gamma_{p-1} \\ \gamma_1 & \gamma_0 & \cdots & \gamma_{p-2} \\ \vdots & \vdots & \ddots & \vdots \\ \gamma_{p-1} & \gamma_{p-2} & \cdots & \gamma_0 \end{pmatrix}, \quad \gamma := \begin{pmatrix} \gamma_1 \\ \gamma_2 \\ \vdots \\ \gamma_p \end{pmatrix}.$$

*Then the AR coefficient vector $\rho := (\rho_1, \ldots, \rho_p)^\top$ satisfies the Yule–Walker system:*

$$\Gamma_p \rho = \gamma.$$

*This moment-matching condition links the autocovariance structure to the autoregressive coefficients.*

**Theorem D.2** (OLS Consistency for AR(p) Estimation (Hamilton, 2020)). *Let $\{x_i\}_{i=1}^T$ be generated by an AR(p) process as in Definition 2.2. Let $\widehat{\rho}_n := (X^\top X)^{-1}X^\top y$ be the ordinary least squares estimator obtained from*

$$y := (x_{p+1}, \ldots, x_n)^\top, \quad X := \begin{pmatrix} x_p & x_{p-1} & \cdots & x_1 \\ x_{p+1} & x_p & \cdots & x_2 \\ \vdots & \vdots & \ddots & \vdots \\ x_{n-1} & x_{n-2} & \cdots & x_{n-p} \end{pmatrix}.$$

*Then, as $n \to \infty$,*

$$\widehat{\rho}_n \xrightarrow{\text{a.s.}} \rho.$$

*Hence, in the large-sample limit, OLS recovers the Bayes-optimal linear predictor in mean squared error.*

**Theorem D.3** (Linear Baseline under AR(p) Dynamics). *Let $\{x_i\}_{i=1}^T$ be generated by an AR(p) process as in Definition 2.2. Fix a context window $x_{1:n} \in \mathbb{R}^n$ with $n \geq p$. For any linear predictor $\widehat{x}_{n+1}^{\mathrm{LR}} = w^\top x_{1:n}$ we have*

$$\min_{w\in\mathbb{R}^n} \mathbb{E}\left[(w^\top x_{1:n} - x_{n+1})^2\right] < \mathrm{Var}(x_{n+1}),$$

*where the expectation is over the stationary joint distribution of $(x_1, \ldots, x_{n+1})$.*

*Proof.* By Definition 2.2, $\mathbb{E}[x_i] = 0$ and the noise variance is $\sigma^2$. The Bayes-optimal (and hence optimal linear) predictor is

$$\widehat{x}_{n+1}^{\mathrm{LR}} = \mathbb{E}[x_{n+1} \mid x_{1:n}] = \sum_{j=1}^p \rho_j x_{n-j+1}.$$

Its mean squared error is

$$\mathbb{E}\left[\left(\widehat{x}_{n+1}^{\mathrm{LR}} - x_{n+1}\right)^2\right] = \sigma^2 \ < \ \sigma^2 + \mathrm{Var}(x_n) = \mathrm{Var}(x_{n+1}),$$

which is strictly below the variance baseline. □

## E TRANSFORMERS EXPRESSIVITY FOR TSF

### E.1 FUNDAMENTALS OF LINEAR SELF-ATTENTION

We work with Hankelized inputs for time-series forecasting. For context length $p$ and total length $n \geq p$, define the Hankel matrix

$$H = \begin{bmatrix} x^{(1)} \ x^{(2)} \ \cdots \ x^{(n-p+1)} \end{bmatrix} \in \mathbb{R}^{(p+1)\times(n-p+1)},$$

where each $x^{(i)} \in \mathbb{R}^{p+1}$ stacks a length-$p$ window and a $(p+1)$-st *label slot*; we reserve the last column $x^{(n-p+1)}$ as the prediction token (see Definition 2.5 for construction details).

**Definition E.1** (Linear Self-Attention (single layer)). *Let $P, Q \in \mathbb{R}^{(p+1)\times(p+1)}$ be trainable. Define*

$$S := H^\top Q H \in \mathbb{R}^{(n-p+1)\times(n-p+1)}, \qquad M := \mathrm{diag}\big(I_{n-p}, 0\big) \in \mathbb{R}^{(n-p+1)\times(n-p+1)}.$$

*The one-layer LSA update is*

$$\mathsf{LSA}(H) := H + \frac{1}{n}\, P\left(H\,M\,S\right) \in \mathbb{R}^{(p+1)\times(n-p+1)}, \tag{1}$$

*where $M$ masks the prediction token to avoid self-use during the update.*

The multi-layer variant is defined by composing $L$ residual bilinear updates.

**Definition E.2** ($L$-layer LSA Transformer). *For parameters $\{P_\ell, Q_\ell\}_{\ell=1}^L$, define each layer as*

$$\mathsf{LSA}_\ell(H) := H + \frac{1}{n}\, P_\ell\big(H\,M\,(H^\top Q_\ell H)\big),$$

*and the overall L-layer LSA Transformer as*

$$\mathsf{TF}(H) := \mathsf{LSA}_L \circ \cdots \circ \mathsf{LSA}_1(H).$$

We now formalize the readout at the prediction slot for a single LSA layer, which will later serve as the basic building block in our theoritical analysis.

**Lemma E.3** (Closed-form readout of one-layer LSA on Hankel input (Ahn et al., 2023)). *Following the construction in Definition 2.5 and Definition E.1, the final column of $H$ is the prediction token,*

$$x^{(n-p+1)} = \begin{bmatrix} x_{\,n-p+1:n} \\ 0 \end{bmatrix}, \qquad x_{\,n-p+1:n} \in \mathbb{R}^p.$$

*Suppose*

$$P = \begin{bmatrix} \mathbf{0}_{p\times(p+1)} \\ b^\top \end{bmatrix}, \qquad Q = \begin{bmatrix} A \ \mathbf{0}_{(p+1)\times 1} \end{bmatrix},$$

*with $b \in \mathbb{R}^{p+1}$ and $A \in \mathbb{R}^{(p+1)\times p}$. Then the prediction-slot entry satisfies*

$$\big[\mathsf{LSA}(H)\big]_{p+1,\, n-p+1} = b^\top G_n A\, x_{\,n-p+1:n},$$

*where we define the empirical Gram matrix*

$$G_n := \frac{1}{n} \sum_{i=1}^{n-p} x^{(i)} x^{(i)\top}.$$

*Proof.* Let $S = H^\top Q H$. Since the last column of $M$ is zero,

$$HM = \begin{bmatrix} x^{(1)} \ \cdots \ x^{(n-p)} \ 0 \end{bmatrix}.$$

Because $[H]_{p+1,\, n-p+1} = 0$ and $e_{p+1}^\top P = b^\top$,

$$\big[\mathsf{LSA}(H)\big]_{p+1,\, n-p+1} = \frac{1}{n}\, b^\top (HMS) e_{\,n-p+1} = \frac{1}{n} \sum_{j=1}^{n-p} b^\top x^{(j)}\, S_{j,\, n-p+1}.$$

For $j \leq n-p$,

$$S_{j,\, n-p+1} = x^{(j)\top} Q x^{(n-p+1)} = x^{(j)\top} A\, x_{\,n-p+1:n},$$

which gives the stated expression. □

### E.2 FUNCTION SPACE CONSTRAINTS OF LINEAR SELF-ATTENTION

In this subsection we take a function-space perspective on linear self-attention. Rather than analyzing specific computations, we characterize the class of functions that a one-layer LSA readout can realize on Hankelized inputs. This higher-level view makes explicit that LSA predictions are confined to a restricted polynomial feature space, and thus—despite the nontrivial attention mechanism—cannot achieve fundamentally lower risk than classical linear regression on AR($p$) processes.

**Lemma E.4** (Doob–Dynkin (Kallenberg, 2006)). *Let $f : \Omega \to S$ and $g : \Omega \to T$ be measurable mappings between measurable spaces. Then $f$ is $\sigma(g)$-measurable if and only if there exists a measurable $h : T \to S$ such that $f = h \circ g$.*

**Theorem E.5** (Projection Monotonicity in $L^2$). *Let $H$ be a Hilbert space and $M_2 \subseteq M_1 \subseteq H$ be closed subspaces. For any $Z \in H$, letting $P_{M_i}$ denote the orthogonal projection onto $M_i$, we have*

$$\|Z - P_{M_1}Z\| \ \leq \ \|Z - P_{M_2}Z\|.$$

**Functional form of the one-layer readout.** Let $H = [\, x^{(1)} \ \dots \ x^{(n-p+1)} \,] \in \mathbb{R}^{(p+1)\times(n-p+1)}$ be the Hankel matrix from Definition 2.5. Following Definition E.1, the one-layer LSA update is

$$\mathsf{LSA}(H) \ = \ H \ + \ \frac{1}{n} P\big(H\, M\, (H^\top Q H)\big).$$

Following Definition 2.5, the final column of $H$ is the prediction token,

$$x^{(n-p+1)} \ = \ \begin{bmatrix} x_{\,n-p+1:n} \\ 0 \end{bmatrix}, \qquad x_{\,n-p+1:n} \in \mathbb{R}^p.$$

By Lemma E.3, for suitable trainable $P, Q$,

$$\big[\mathsf{LSA}(H)\big]_{p+1,\,n-p+1} \ = \ b^\top G_n\, A\, x_{\,n-p+1:n}, \qquad G_n \ := \ \frac{1}{n}\sum_{i=1}^{n-p} x^{(i)}x^{(i)\top}.$$

**Definition E.6** (LSA feature space). *For $j, r \in [p+1]$ and $k \in [p]$, define the cubic coordinates*

$$\phi^{(p)}_{j,r,k}(x_{1:n}) \ := \ \frac{1}{n}\sum_{i=1}^{n-p} x_{\,i+j-1}\, x_{\,i+r-1}\, x_{\,n-p+k}.$$

*Collect them into the feature map*

$$\Phi^{(p)}(x_{1:n}) \ := \ \big(\phi^{(p)}_{j,r,k}(x_{1:n})\big)_{j,r\in[p+1],\, k\in[p]} \in \mathbb{R}^m, \qquad m \ = \ p\,(p+1)^2,$$

*and define the linear self-attention feature space*

$$\mathcal{H}^{(p)}_{\mathrm{LSA}} \ := \ \mathrm{span}\Big\{\phi^{(p)}_{j,r,k}(\,\cdot\,) :\ j,r \in [p+1],\ k \in [p]\Big\}.$$

*Since each coordinate is a finite sum of degree-3 monomials, $\Phi^{(p)}$ is continuous.*

**Theorem E.7** (Representation of one-layer LSA readout). *There exist coefficients $\{\beta_{j,r,k}\}$, depending on the trainable parameters in $P, Q$, such that*

$$\widehat{x}_{n+1} \ = \ \big[\mathsf{LSA}(H)\big]_{p+1,\,n-p+1} \ = \ \sum_{j=1}^{p+1}\sum_{r=1}^{p+1}\sum_{k=1}^{p} \beta_{j,r,k}\, \phi^{(p)}_{j,r,k}(x_{1:n}) \ \in \ \mathcal{H}^{(p)}_{\mathrm{LSA}}.$$

*Proof.* By Lemma E.3, for suitable trainable $P, Q$,

$$\big[\mathsf{LSA}(H)\big]_{p+1,\,n-p+1} = b^\top G_n A x_{\,n-p+1:n}, \qquad G_n := \frac{1}{n}\sum_{i=1}^{n-p} x^{(i)}x^{(i)\top}.$$

Expanding the product gives

$$b^\top G_n A x_{n-p+1:n} = \sum_{j=1}^{p+1}\sum_{r=1}^{p+1}\sum_{k=1}^{p} (b_j A_{r,k})\, \phi^{(p)}_{j,r,k}(x_{1:n}),$$

where we define $\beta_{j,r,k} := b_j A_{r,k}$. This matches the claimed representation. □

**Theorem E.8** (Risk monotonicity under LSA features). *Let $(x_{1:n}, x_{n+1})$ be jointly defined random variables. Under squared loss,*

$$\inf_f \ \mathbb{E}\left[\left(x_{n+1} - f(x_{1:n})\right)^2\right] \ \leq \ \inf_g \ \mathbb{E}\left[\left(x_{n+1} - g(\Phi^{(p)}(x_{1:n}))\right)^2\right].$$

*Equality holds if and only if $x_{n+1} \perp\!\!\!\perp x_{1:n} \,|\, \Phi^{(p)}(x_{1:n})$.*

*Proof.* By optimality of conditional expectations, the minimizers are $\mathbb{E}[x_{n+1} \mid x_{1:n}]$ and $\mathbb{E}[x_{n+1} \mid \Phi^{(p)}(x_{1:n})]$. Since $\Phi^{(p)}$ is a deterministic function of $x_{1:n}$, Doob–Dynkin (Lemma E.4) gives $\sigma(\Phi^{(p)}(x_{1:n})) \subseteq \sigma(x_{1:n})$. Conditional expectation is the $L^2$ projection, so Theorem E.5 yields the inequality; the equality condition is standard. $\square$

**Theorem E.9** (Single-layer LSA cannot beat the linear predictor under AR($p$)). *Suppose $\{x_t\}$ follows a stationary AR($p$) process with context $x_{1:n}$ and $n \geq p$. Let $\widehat{x}_{n+1}^{\mathrm{LSA}}$ denote any one-layer LSA readout as in Theorem E.7. Then*

$$\inf \ \mathbb{E}\left[\left(\widehat{x}_{n+1}^{\mathrm{LSA}} - x_{n+1}\right)^2\right] \ \geq \ \inf_{w \in \mathbb{R}^n} \mathbb{E}\left[\left(w^\top x_{1:n} - x_{n+1}\right)^2\right].$$

*Proof.* By Theorem E.7, $\widehat{x}_{n+1}^{\mathrm{LSA}}$ is a measurable function of $\Phi^{(p)}(x_{1:n})$, hence its best risk is at least that of the Bayes predictor given $\Phi^{(p)}(x_{1:n})$; Theorem E.8 then lower-bounds this by the Bayes risk given $x_{1:n}$. Under AR($p$), $\mathbb{E}[x_{n+1} \mid x_{1:n}]$ is linear in $x_{1:n}$, so the Bayes risk equals the optimal linear risk. Hence the claim. $\square$

### E.3 NESTED TRANSFORMER FEATURE SPACES AND RISK MONOTONICITY IN TIME SERIES PREDICTION

We take a function-space perspective: with Hankel inputs and a masked label slot, the one-layer LSA readout operates in a restricted feature class. As the sequence length grows, these features collapse to scaled copies of the last $p$ lags, which clarifies both our Hankel design ($p+1$ rows) and why one-layer LSA cannot fundamentally beat linear regression on AR($p$).

**Setup.** Throughout this subsection we work under the AR($p$) model in Definition 2.2, the Hankel construction in Definition 2.5, the one-layer LSA update in Definition E.1, and the cubic coordinates in Definition E.6. Let $M$ be the mask from Definition E.1.

**Theorem E.10** (Ergodic Theorem, Birkhoff (Kallenberg, 2006)). *Let $\xi$ be a random element in a measurable space $S$ with distribution $\mu$, and let $T : S \to S$ be a $\mu$-preserving transformation with invariant $\sigma$-field $\mathcal{I}$. Then for any measurable function $f \geq 0$ on $S$, we have*

$$\frac{1}{n} \sum_{k=0}^{n-1} f(T^k \xi) \xrightarrow{a.s.} \mathbb{E}[f(\xi) \mid \mathcal{I}],$$

*where the convergence is almost surely.*

**Lemma E.11** (Asymptotic feature collapse). *Define the normalized masked Hankel Gram*

$$G_n \ := \ \frac{1}{n} \, H \, M \, H^\top.$$

*Then, entrywise, $G_n \xrightarrow{a.s.} \Gamma_{p+1}$, where $[\Gamma_{p+1}]_{ij} = \gamma_{|i-j|}$ and $\gamma_h = \mathbb{E}[x_t x_{t+h}]$. Moreover, for the cubic coordinates $\phi_{j,r,k}^{(p)}$ in Definition E.6,*

$$\phi_{j,r,k}^{(p)}(x_{1:n}) \ \xrightarrow{a.s.} \ \gamma_{|j-r|} \, x_{n-p+k}, \qquad j, r \in [p+1], \ k \in [p].$$

*Hence,*

$$\mathrm{span}\big\{\phi_{j,r,k}^{(p)}(x_{1:n})\big\} \ \xrightarrow{a.s.} \ \mathrm{span}\{x_{n-p+1}, \ldots, x_n\}.$$

*Proof.* Each entry of $G_n$ is a time average of $x_{i+a-1} x_{i+b-1}$; by Birkhoff's ergodic theorem (Theorem E.10), $G_n \to \Gamma_{p+1}$ almost surely. For $\phi_{j,r,k}^{(p)}$, factor out $x_{n-p+k}$ and apply the same theorem to $\frac{1}{n} \sum_i x_{i+j-1} x_{i+r-1}$. $\square$

**Corollary E.12** (Nested spaces). *Let $\widetilde{\mathcal{H}}_{\mathrm{LSA}}^{(p)} := \mathrm{span}\{x_{n-p+1}, \dots, x_n\}$. Then $\widetilde{\mathcal{H}}_{\mathrm{LSA}}^{(p)} \subseteq \widetilde{\mathcal{H}}_{\mathrm{LSA}}^{(p+1)}$ for all $p$.*

**Theorem E.13** (Risk plateau on AR($p$)). *Under AR($p$), the Bayes predictor is linear in the last $p$ lags: $x_{n+1} = \sum_{j=1}^{p} \rho_j x_{n-j+1} + \varepsilon_{n+1}$ with $\mathbb{E}[\varepsilon_{n+1} \mid x_{1:n}] = 0$. By Lemma E.11 and Corollary E.12, any one-layer LSA based on Hankel features operates (as $n \to \infty$) on $\mathrm{span}\{x_{n-p+1}, \dots, x_n\}$. Therefore, the minimal achievable MSE equals the linear Bayes risk $\sigma_\varepsilon^2$ and does not decrease for $q \geq p$:*

$$\inf_f \mathbb{E}\Big[(x_{n+1} - f(\text{LSA features of order } q))^2\Big] = \sigma_\varepsilon^2, \qquad \forall\, q \geq p.$$

**Proposition E.14** (Asymptotic exact recovery: a constructive parameter choice). *Let the one-layer readout at the prediction slot be written as in Lemma E.3:*

$$\big[\mathsf{LSA}(H)\big]_{p+1,\,n-p+1} \;=\; b^\top G_n A\, x_{n-p+1:n},$$

*with $b \in \mathbb{R}^{p+1}$ and $A \in \mathbb{R}^{(p+1)\times p}$. Define*

$$b^* \;:=\; (0, \dots, 0, 1)^\top \in \mathbb{R}^{p+1}, \qquad A^* \;:=\; \begin{bmatrix} J\,\Gamma_p^{-1} \\ \mathbf{0}_{1\times p} \end{bmatrix} \in \mathbb{R}^{(p+1)\times p},$$

*where $J \in \mathbb{R}^{p\times p}$ is the anti-diagonal permutation (reversal) matrix and $\Gamma_p$ is the $p\times p$ autocovariance Toeplitz matrix. Then, as $n \to \infty$,*

$$b^{*\top} G_n A^*\, x_{n-p+1:n} \;\xrightarrow{a.s.}\; \rho^\top x_{n-p+1:n},$$

*i.e., the one-layer LSA readout exactly recovers the Bayes-optimal linear predictor in the limit.*

*Proof.* By Lemma E.11, $G_n \to \Gamma_{p+1}$. Since $b^{*\top}\Gamma_{p+1} = [\gamma_p, \dots, \gamma_1, \gamma_0]$, we get

$$b^{*\top}\Gamma_{p+1} A^* \;=\; [\gamma_p, \dots, \gamma_1]\, J\,\Gamma_p^{-1} \;=\; [\gamma_1, \dots, \gamma_p]\,\Gamma_p^{-1} \;=\; \rho^\top,$$

using the Yule–Walker relation $\rho = \Gamma_p^{-1}\gamma$ with $\gamma = (\gamma_1, \dots, \gamma_p)^\top$. $\qquad\qquad\square$

**Why exactly $p+1$ Hankel rows.** The first $p$ rows are necessary to capture the true $p$ lags of the AR($p$) process, while the $(p+1)$-st row serves as the masked prediction slot. With fewer than $p+1$ rows, the model cannot access all $p$ relevant lags. With more than $p+1$ rows, the extra rows are asymptotically redundant, since the optimal predictor ultimately depends only on the last $p$ lags.

# F CLOSED-FORM GAP CHARACTERIZATION BETWEEN LSA AND LINEAR MODELS

## F.1 WARM UP VIA AR(1)

*Warm-up, not global optimum.* Motivated by the asymptotic constructive choice in Proposition E.14 (where $b$ has the last entry 1 and the last row of $A$ is 0), we study the univariate AR(1) case and *restrict* $(b, A)$ to the one-dimensional ray induced by that limiting structure. This gives a computable warm start that illustrates how Isserlis' theorem (Theorem F.3) enters the calculation; it is not the global optimum over all $(b, A)$, but it tends to linear regression as $n \to \infty$.

**Proposition F.1** (AR(1) warm start along the asymptotic ray). *Let $\{x_t\}$ be the stationary Gaussian AR(1) process of Definition 2.2:*

$$x_t = \rho\, x_{t-1} + \varepsilon_t, \quad |\rho| < 1, \quad \varepsilon_t \overset{\text{i.i.d.}}{\sim} \mathcal{N}(0, \sigma_\varepsilon^2),$$

*and set $\sigma^2 := \sigma_\varepsilon^2/(1 - \rho^2) = \mathrm{var}(x_t)$. For context $p = 1$ and $n \geq 3$, let $H_n$ be the Hankel matrix (Definition 2.5) with mask $M = \mathrm{diag}(I_{n-1}, 0)$ and define the normalized Gram*

$$G_n := \frac{1}{n} H_n M H_n^\top = \frac{1}{n} \sum_{i=1}^{n-1} \begin{pmatrix} x_i^2 & x_i x_{i+1} \\ x_i x_{i+1} & x_{i+1}^2 \end{pmatrix}.$$

*Let $S_n := \frac{1}{n} \sum_{i=1}^{n-1} x_i x_{i+1}$. Restrict $(b, A)$ to*

$$b = \begin{bmatrix} 0 \\ 1 \end{bmatrix}, \qquad A = \begin{bmatrix} \alpha \\ 0 \end{bmatrix}, \qquad \alpha \in \mathbb{R},$$

*so the one-layer LSA readout at the prediction slot is*

$$\widehat{y}_n(\alpha) = (b^\top G_n A)\, x_n = \alpha\, S_n\, x_n.$$

*Consider the surrogate loss against the AR(1) linear term:*

$$\mathcal{L}(\alpha) = \mathbb{E}[(\widehat{y}_n(\alpha) - \rho x_n)^2] = \mathbb{E}[x_n^2(\alpha S_n - \rho)^2] = \alpha^2 D_n - 2\alpha\rho N_n + \rho^2 \sigma^2.$$

*Define the geometric sums (for $m \geq 1$)*

$$K_m := \sum_{k=1}^m \rho^{2k} = \frac{\rho^2(1 - \rho^{2m})}{1 - \rho^2}, \qquad H_m := \sum_{k=1}^m k\, \rho^{2k} = \frac{\rho^2\big(1 - (m+1)\rho^{2m} + m\rho^{2(m+1)}\big)}{(1 - \rho^2)^2}.$$

*Then, with $\Gamma_k := \mathrm{Cov}(x_t, x_{t+k}) = \sigma^2 \rho^{|k|}$,*

$$N_n := \mathbb{E}[x_n^2 S_n] = \frac{\sigma^4}{n}\Big[(n-1)\rho + \frac{2}{\rho}\, K_{n-1}\Big], \tag{2}$$

$$D_n := \mathbb{E}[x_n^2 S_n^2] = \frac{\sigma^6}{n^2}\Big[(n-1)n\,\rho^2 + (n-1) + \big(4(n-1) - 2\big)K_{n-1}$$

$$+ \big(4(n-1) + 2\big)K_{n-2} + 8H_{n-1} + 4H_{n-2}\Big]. \tag{3}$$

*Consequently the unique minimizer along this ray and its value are*

$$\alpha^* = \frac{\rho\, N_n}{D_n}, \qquad \min_\alpha \mathcal{L}(\alpha) = \rho^2 \sigma^2 - \frac{\rho^2 N_n^2}{D_n}.$$

*Equivalently,*

$$\min_\alpha \mathbb{E}\big[(\widehat{x}_{n+1}^{\mathrm{LSA}} - x_{n+1})^2\big] = \sigma_\varepsilon^2 + \rho^2 \sigma^2 - \frac{\rho^2 N_n^2}{D_n},$$

*and, by the law of large numbers and dominated convergence,*

$$\alpha^* \xrightarrow[n\to\infty]{} \frac{1}{\sigma^2}, \qquad \min_\alpha \mathcal{L}(\alpha) \xrightarrow[n\to\infty]{} 0,$$

*i.e., this warm start tends to the linear-regression limit.*

*Proof.* **Step 1: $N_n$—pairwise expansion (4th order).**

$$N_n = \frac{1}{n} \sum_{i=1}^{n-1} \mathbb{E}[x_n^2 x_i x_{i+1}].$$

For the zero-mean Gaussian quadruple $(x_n, x_n, x_i, x_{i+1})$, by Isserlis (Theorem F.3) the three pairings give

$$
\begin{aligned}
(x_n, x_n)(x_i, x_{i+1}) &\Rightarrow \Gamma_0 \Gamma_1 = \sigma^4 \rho, \\
(x_n, x_i)(x_n, x_{i+1}) &\Rightarrow \Gamma_{n-i} \Gamma_{n-i-1} = \sigma^4 \rho^{2n-2i-1}, \\
(x_n, x_{i+1})(x_n, x_i) &\Rightarrow \Gamma_{n-i-1} \Gamma_{n-i} = \sigma^4 \rho^{2n-2i-1}.
\end{aligned}
$$

Hence $\mathbb{E}[x_n^2 x_i x_{i+1}] = \sigma^4 (\rho + 2\rho^{2n-2i-1})$, and summing over $i$ yields (2).

**Step 2: $D_n$—pairwise expansion (6th order).**

$$D_n = \frac{1}{n^2} \sum_{i=1}^{n-1} \sum_{j=1}^{n-1} \mathbb{E}[x_n^2 x_i x_{i+1} x_j x_{j+1}]$$

splits into diagonal ($i = j$) and off-diagonal ($i < j$) parts.

*Diagonal $i = j$.* With $X := x_n$, $Y := x_i$, $Z := x_{i+1}$,

$$\mathbb{E}[X^2 Y^2 Z^2] = \Gamma_0^3 + 2\Gamma_{XY}^2 \Gamma_0 + 2\Gamma_{XZ}^2 \Gamma_0 + 2\Gamma_{YZ}^2 \Gamma_0 + 8\Gamma_{XY} \Gamma_{XZ} \Gamma_{YZ}.$$

Substituting $\Gamma_0 = \sigma^2$, $\Gamma_{XY} = \sigma^2 \rho^{n-i}$, $\Gamma_{XZ} = \sigma^2 \rho^{n-i-1}$, $\Gamma_{YZ} = \sigma^2 \rho$ and summing over $i = 1, \ldots, n-1$ gives the diagonal contribution

$$\sigma^6 \left( (n-1) + 2\rho^2(n-1) + 2\sum_{k=1}^{n-2} \rho^{2k} + 10\sum_{k=1}^{n-1} \rho^{2k} \right) = \sigma^6 \left( (n-1) + 2\rho^2(n-1) + 2K_{n-2} + 10K_{n-1} \right).$$

*Off-diagonal $i < j$.* Let $Y_1 := x_i$, $Y_2 := x_{i+1}$, $Z_1 := x_j$, $Z_2 := x_{j+1}$. Grouping the 15 pairings by how the two copies of $x_n$ are paired gives the per–$(i,j)$ contributions listed below (the factor "$\times 2$" indicates the symmetric counterpart):

$$
\begin{aligned}
(x_n, x_n)(Y_1, Y_2)(Z_1, Z_2) &\Rightarrow \sigma^6 \rho^2 \\
(x_n, x_n)(Y_1, Z_1)(Y_2, Z_2) &\Rightarrow \sigma^6 \rho^{2(j-i)} \\
(x_n, x_n)(Y_1, Z_2)(Y_2, Z_1) &\Rightarrow \sigma^6 \rho^{2(j-i)} \\
\hline
\{(x_n, Y_1), (x_n, Y_2)\}, (Z_1, Z_2) \ (\times 2) &\Rightarrow 2\sigma^6 \rho^{2(n-i)} \\
\{(x_n, Z_1), (x_n, Z_2)\}, (Y_1, Y_2) \ (\times 2) &\Rightarrow 2\sigma^6 \rho^{2(n-j)} \\
\{(x_n, Y_1), (x_n, Z_1)\}, (Y_2, Z_2) \ (\times 2) &\Rightarrow 2\sigma^6 \rho^{2(n-i)} \\
\{(x_n, Y_1), (x_n, Z_2)\}, (Y_2, Z_1) \ (\times 2) &\Rightarrow 2\sigma^6 \rho^{2(n-i)} \\
\{(x_n, Y_2), (x_n, Z_1)\}, (Y_1, Z_2) \ (\times 2) &\Rightarrow 2\sigma^6 \rho^{2(n-i-1)} \\
\{(x_n, Y_2), (x_n, Z_2)\}, (Y_1, Z_1) \ (\times 2) &\Rightarrow 2\sigma^6 \rho^{2(n-i-1)}
\end{aligned}
$$

Summing over $1 \leq i < j \leq n-1$ and reindexing,

$$
\begin{aligned}
\sum_{1 \leq i < j \leq n-1} &\mathbb{E}[x_n^2 x_i x_{i+1} x_j x_{j+1}] \\
&= \sigma^6 \left( \binom{n-1}{2} \rho^2 + 2\sum_{k=1}^{n-1}(n-1-k)\rho^{2k} + 6\sum_{i=1}^{n-1}(n-1-i)\rho^{2(n-i)} \right. \\
&\qquad \left. + 4\sum_{i=1}^{n-1}(n-1-i)\rho^{2(n-1-i)} + 2\sum_{k=1}^{n-2}(n-1-k)\rho^{2k} \right) \\
&= \sigma^6 \left( \binom{n-1}{2} \rho^2 + (2n-8)K_{n-1} + 4H_{n-1} + 2(n-1)K_{n-2} + 2H_{n-2} \right).
\end{aligned}
$$

Finally, we obtain the final expression of $D_n$

$$D_n = \frac{1}{n^2} \sum_{i=1}^{n-1} \sum_{j=1}^{n-1} \mathbb{E}[x_n^2 x_i x_{i+1} x_j x_{j+1}]$$

$$= \frac{1}{n^2} \sum_{1 \leq i=j \leq n-1} \mathbb{E}[x_n^2 x_i x_{i+1} x_j x_{j+1}] + \frac{2}{n^2} \sum_{1 \leq i < j \leq n-1} \mathbb{E}[x_n^2 x_i x_{i+1} x_j x_{j+1}]$$

$$= \frac{\sigma^6}{n^2} \Big( (n-1) + 2\rho^2(n-1) + 2K_{n-2} + 10K_{n-1} \Big)$$

$$\quad + \frac{2\sigma^6}{n^2} \Big( \binom{n-1}{2} \rho^2 + (2n-8)K_{n-1} + 4H_{n-1} + 2(n-1)K_{n-2} + 2H_{n-2} \Big).$$

$$= \frac{\sigma^6}{n^2} \Big[ (n-1)n\,\rho^2 + (n-1) + (4(n-1)-2)\,K_{n-1} + (4(n-1)+2)\,K_{n-2} + 8H_{n-1} + 4H_{n-2} \Big].$$

Adding the diagonal part and multiplying by $1/n^2$ yields (3).

**Step 3: Limit and conclusion.** Since $D_n > 0$, $\mathcal{L}$ is strictly convex, so the minimizer and value follow. As $S_n \to \mathbb{E}[x_t x_{t+1}] = \rho\sigma^2$ almost surely and $x_n^2$ is integrable, $\alpha^* \to 1/\sigma^2$ and $\min_\alpha \mathcal{L}(\alpha) \to 0$. $\square$

**Remark F.2** (Warm start vs. global optimum). *The optimization above is restricted to the asymptotic ray $b = [0,1]^\top$, $A = [\alpha, 0]^\top$. It serves as a computational warm-up to practice Isserlis-based moment calculations and to quantify the finite-sample gap. The global optimum over all $(b, A)$ may differ, but this warm start already converges to the linear-regression limit as $n \to \infty$.*

AUXILIARY LEMMAS USED IN APPENDIX F.1

**Theorem F.3** (Isserlis' Theorem (Wick's Formula) (Isserlis, 1918)). *Let $(X_1, \ldots, X_n)$ be a zero-mean multivariate normal random vector.*

- *If $n$ is odd, i.e., $n = 2m + 1$, then*

$$\mathbb{E}[X_1 X_2 \cdots X_{2m+1}] = 0.$$

- *If $n$ is even, i.e., $n = 2m$, then*

$$\mathbb{E}[X_1 X_2 \cdots X_{2m}] = \sum_{p \in \mathcal{P}_n} \prod_{\{i,j\} \in p} \mathbb{E}[X_i X_j],$$

*where $\mathcal{P}_n$ denotes the set of all possible pairwise partitions (perfect matchings) of $\{1, 2, \ldots, 2m\}$.*

F.2    A FINITE-SAMPLE OPTIMALITY GAP FOR ONE-LAYER LSA

**Goal.**    We prove that for any fixed sample size $n$, the best one-layer LSA readout on Hankel inputs *cannot* match linear regression on the last $p$ lags. The argument rewrites the readout as a quadratic form in a Kronecker–lifted feature, solves the induced convex problem in closed form, and identifies a strictly positive Schur–complement gap. In other words, this section provides the first rigorous separation between LSA and linear regression: the gap is not an artifact of optimization, but a structural property of the linear self attention.

**Setup (recalled).**    Let $\{x_t\}$ be a zero-mean stationary AR($p$) process as in Definition 2.2. For $n \geq p$, let $H_n$ be the Hankel matrix from Definition 2.5 and put

$$G := \frac{1}{n} H_n M H_n^\top = \frac{1}{n} \sum_{i=1}^{n-p} x^{(i)} x^{(i)\top} \in \mathbb{R}^{(p+1) \times (p+1)}, \qquad M := \mathrm{diag}(I_{n-p}, 0).$$

Denote the prediction window $x := x_{n-p+1:n} \in \mathbb{R}^p$ and $y := x_{n+1} = \rho^\top x + \varepsilon_{n+1}$ with $\mathbb{E}[\varepsilon_{n+1}] = 0$, $\mathbb{E}[\varepsilon_{n+1}^2] = \sigma_\varepsilon^2$ and $\varepsilon_{n+1} \perp (G, x)$. Given parameters $b \in \mathbb{R}^{p+1}$ and $A \in \mathbb{R}^{(p+1) \times p}$, the one-layer LSA readout at the prediction slot is

$$\widehat{x}_{n+1}(b, A) = b^\top G A x. \tag{4}$$

**Kronecker reparameterization.** Let $D_{p+1}$ be the $(p+1)$-dimensional duplication matrix so that $\mathrm{vec}(G) = D_{p+1}\,\mathrm{vech}(G)$. Using $\mathrm{vec}(AXB) = (B^\top \otimes A)\,\mathrm{vec}(X)$ and the mixed–product rule,

$$x^\top(A^\top G b) = \big((\mathrm{vech}\,G)^\top D_{p+1}^\top \otimes x^\top\big)\big(b \otimes \mathrm{vec}(A^\top)\big) = \big((\mathrm{vech}\,G)^\top \otimes x^\top\big)\widetilde{\eta},$$

where

$$\widetilde{\eta} := (D_{p+1}^\top \otimes I_p)\big(b \otimes \mathrm{vec}(A^\top)\big) \in \mathbb{R}^{qp}, \qquad q := \tfrac{(p+1)(p+2)}{2}.$$

Introduce the lifted feature and its moments

$$Z := (\mathrm{vech}\,G) \otimes x \in \mathbb{R}^{qp}, \qquad \widetilde{S} := \mathbb{E}[ZZ^\top], \qquad \widetilde{r} := \mathbb{E}[Zx^\top], \qquad \Gamma_p := \mathbb{E}[xx^\top].$$

Then the mean–squared error decomposes as a strictly convex quadratic in $\widetilde{\eta}$:

$$\mathcal{L}(b, A) := \mathbb{E}\big[(\widehat{x}_{n+1}(b, A) - y)^2\big]$$
$$= \sigma_\varepsilon^2 + \rho^\top \Gamma_p \rho + \widetilde{\eta}^\top \widetilde{S}\widetilde{\eta} - 2\widetilde{\eta}^\top \widetilde{r}\rho. \tag{5}$$

**Two technical facts we will establish and use.**

(F1) The block second-moment matrix

$$\Sigma^\otimes := \mathbb{E}\left[\begin{pmatrix} Z \\ x \end{pmatrix}\begin{pmatrix} Z \\ x \end{pmatrix}^\top\right] = \begin{pmatrix} \widetilde{S} & \widetilde{r} \\ \widetilde{r}^\top & \Gamma_p \end{pmatrix} \quad \text{is } \textit{strictly} \text{ positive definite.}$$

Equivalently, $\widetilde{S} \succ 0$ and the Schur complement $\Gamma_p - \widetilde{r}^\top \widetilde{S}^{-1}\widetilde{r} \succ 0$.

(F2) The unconstrained minimizer of (5) is $\widetilde{\eta}^* = \widetilde{S}^{-1}\widetilde{r}\rho$ with minimum value

$$\mathcal{L}_{\min} = \sigma_\varepsilon^2 + \rho^\top \Gamma_p \rho - \rho^\top \widetilde{r}^\top \widetilde{S}^{-1}\widetilde{r}\rho.$$

Because the original parameters $(b, A)$ satisfy the rank-one Kronecker constraint $\widetilde{\eta} = (D_{p+1}^\top \otimes I_p)(b \otimes \mathrm{vec}(A^\top))$, their achievable risk is *no smaller* than $\mathcal{L}_{\min}$.

The core of the proof is thus (F1); (F2) is elementary convex optimization once (F1) holds.

STEP 1: STRICT POSITIVE DEFINITENESS OF A BASIC BLOCK

We begin by showing that the basic statistics at time $n$ already enjoy strict nondegeneracy.

**Lemma F.4** (Strict covariance of $(\mathrm{vech}\,G, x)$). *Let $g := \mathrm{vech}\,G \in \mathbb{R}^q$ and $x := x_{n-p+1:n} \in \mathbb{R}^p$. Then the covariance matrix*

$$\mathrm{Cov}\big([\,g^\top, x^\top\,]^\top\big) \succ 0.$$

*Proof.* Write $Z_0 := [\,g^\top, x^\top\,]^\top$ and suppose, for contradiction, that there exists a nonzero vector $v$ with $\mathrm{Var}(v^\top Z_0) = 0$. We will force all coordinates of $v$ to be zero by an *innovation-by-innovation* elimination. Let $\mathcal{F}_t := \sigma(\varepsilon_s : s \le t)$. Proceed by traversing the rearranged vector $Z_0$ as shown below, in a *row–wise from bottom to top*, eliminating coefficients accordingly.

$$\widetilde{Z}_0 = \left[\begin{array}{ccccc|c}
\sum_{m=1}^{n-p} x_m^2 & & & & & \\
\sum_{m=1}^{n-p} x_m x_{m+1} & \sum_{m=1}^{n-p} x_{m+1}^2 & & & & x_{n-p+1} \\
\sum_{m=1}^{n-p} x_m x_{m+2} & \sum_{m=1}^{n-p} x_{m+1} x_{m+2} & \sum_{m=1}^{n-p} x_{m+2}^2 & & & x_{n-p+2} \\
\vdots & \vdots & \vdots & \ddots & & \vdots \\
\sum_{m=1}^{n-p} x_m x_{m+p} & \sum_{m=1}^{n-p} x_{m+1} x_{m+p} & \sum_{m=1}^{n-p} x_{m+2} x_{m+p} & \cdots & \sum_{m=1}^{n-p} x_{m+p}^2 & x_n
\end{array}\right]$$

*Bottom block (involving $x_n$).* The last row of the Hankel Gram contributes, among others, the terms $\sum_{m=1}^{n-p} x_{m+p}^2$ and $\sum_{m=1}^{n-p} x_{m+p-1}x_{m+p}$, whose final summands are $x_n^2$ and $x_{n-1}x_n$. Collecting the coefficients in $v$ that multiply $\{x_n^2, x_{n-1}x_n, \ldots, x_{n-p}x_n, x_n\}$, we can write

$$v^\top Z_0 = (\text{terms } \mathcal{F}_{n-1}\text{-measurable}) + \left(u^\top x_{n-p:n-1} + a\,x_n + c\right)x_n,$$

for some reals $a, c$ and a vector $u \in \mathbb{R}^p$ determined by $v$. Since $x_n = \rho^\top x_{n-p:n-1} + \varepsilon_n$ with $\varepsilon_n \perp \mathcal{F}_{n-1}$ and $\varepsilon_n$ independent of all earlier innovations, the conditional variance is

$$\mathrm{Var}\!\left(v^\top Z_0 \mid \mathcal{F}_{n-1}\right) = \mathrm{Var}\!\left((u^\top x_{n-p:n-1} + c)\varepsilon_n + a\,\varepsilon_n^2 \mid \mathcal{F}_{n-1}\right).$$

If $a \neq 0$ on a set of positive probability and $\varepsilon_n$ has finite fourth moment (true e.g. for Gaussian innovations), then this conditional variance is $> 0$ on that set; hence $\mathrm{Var}(v^\top Z_0) > 0$, a contradiction. Thus $a = 0$ a.s. With $a = 0$, the conditional variance reduces to $\mathrm{Var}\!\left((u^\top x_{n-p:n-1} + c)\varepsilon_n \mid \mathcal{F}_{n-1}\right) = (u^\top x_{n-p:n-1} + c)^2\sigma_\varepsilon^2$, forcing $u^\top x_{n-p:n-1} + c = 0$ a.s. By Lemma F.10, this implies $u = 0$ and $c = 0$. Hence *all* coefficients of $v$ that touch $x_n$ vanish.

*Induction upward.* Assume we have eliminated all coefficients attached to $x_n, x_{n-1}, \ldots, x_{n-r+1}$ and to every Gram entry that involves these variables (this means we have removed the last $r$ block-rows of the lower-triangular tableau of the sums defining $g$). Focus on the remaining first time that appears, $x_{n-r}$. Exactly the same conditioning on $\mathcal{F}_{n-r-1}$, writing $x_{n-r} = \phi^\top x_{n-r-p:n-r-1} + \varepsilon_{n-r}$, shows that the coefficient of $x_{n-r}^2$ must be 0, and then the linear coefficient of $x_{n-r}$ must be 0. By Lemma F.10 again, *all* coefficients in $v$ that touch $x_{n-r}$ vanish.

Proceeding for $r = 1, \ldots, p$ removes all entries of $v$ associated with $x$ and with the last $p$ block-rows of $g$. Finally, the block that involves no recent $x$'s also collapses by the same argument (conditioning on $\mathcal{F}_t$ at the appropriate times). We conclude $v = 0$, contradicting the assumption. Therefore $\mathrm{Cov}(Z_0) \succ 0$. $\qquad\square$

**Lemma F.5** (Two linear–algebra tools). *(i) If $\Sigma = \begin{pmatrix} \Sigma_{AA} & \Sigma_{AB} \\ \Sigma_{BA} & \Sigma_{BB} \end{pmatrix} \in \mathbb{R}^{(m+n)\times(m+n)}$ is positive definite, then the Schur complement $\Sigma_{AA} - \Sigma_{AB}\Sigma_{BB}^{-1}\Sigma_{BA} \succ 0$. (ii) If $X$ is square–integrable with $\mathrm{Cov}(X) \succ 0$ then, for every nonzero $v$, $\mathrm{Var}(v^\top X) > 0$.*

*Proof.* (i) is standard; see, e.g., any matrix analysis text. (ii) is immediate from $v^\top \mathrm{Cov}(X)v = \mathrm{Var}(v^\top X)$. $\qquad\square$

Step 2: lifting to the Kronecker level

We now show that the lift $Z = (\mathrm{vech}\,G) \otimes x$ is also strictly nondegenerate in the block sense.

**Lemma F.6** (Strict PD of the Kronecker–lifted block). *With $g := \mathrm{vech}\,G$, $x := x_{n-p+1:n}$ and $Z := g \otimes x$,*

$$\Sigma^\otimes = \begin{pmatrix} \widetilde{S} & \widetilde{r} \\ \widetilde{r}^\top & \Gamma_p \end{pmatrix} = \mathbb{E}\!\left[\begin{pmatrix} Z \\ x \end{pmatrix}\begin{pmatrix} Z \\ x \end{pmatrix}^\top\right] \succ 0.$$

*Proof.* Take any $(u, w) \neq (0, 0)$ with $u \in \mathbb{R}^{qp}$, $w \in \mathbb{R}^p$, and reshape $u$ into a matrix $U \in \mathbb{R}^{q\times p}$ so that $u = \mathrm{vec}(U)$. Consider the scalar

$$Y := u^\top Z + w^\top x = \sum_{s=1}^{p} x_s\Big(w_s + \sum_{\ell=1}^{q} U_{\ell s}\,g_\ell\Big) = x^\top\big(w + Ug\big).$$

Condition on $x$. Using the tower property,

$$\mathrm{Var}(Y) = \mathbb{E}\big[\mathrm{Var}(Y \mid x)\big] + \mathrm{Var}\big(\mathbb{E}[Y \mid x]\big) \geq \mathbb{E}\big[\mathrm{Var}\big(x^\top(Ug) \mid x\big)\big].$$

Given $x$, $x$ is a deterministic vector and $g$ is still random. Hence

$$\mathrm{Var}\big(x^\top(Ug) \mid x\big) = x^\top U\,\mathrm{Cov}(g \mid x)\,U^\top x.$$

By Lemma F.4 and the Schur–complement Lemma F.5(i),

$$\mathrm{Cov}(g \mid x) = \mathrm{Cov}(g) - \mathrm{Cov}(g, x)\,\Gamma_p^{-1}\,\mathrm{Cov}(x, g) \succ 0.$$

Therefore, for any $U \neq 0$, $x^\top U \operatorname{Cov}(g \mid x) U^\top x > 0$ on a set of positive probability (because $\Gamma_p \succ 0$ and $x$ is non-degenerate); taking expectations yields $\operatorname{Var}(Y) > 0$. If instead $U = 0$ then $u = 0$ and $Y = w^\top x$; since $\Gamma_p \succ 0$, Lemma F.5(ii) implies $\operatorname{Var}(Y) > 0$ unless $w = 0$. Thus no nonzero $(u, w)$ can make $\operatorname{Var}(Y) = 0$, which proves $\Sigma^\otimes \succ 0$. $\qquad\square$

STEP 3: THE GAP VIA A SCHUR COMPLEMENT

We are ready to state and prove the main result.

**Theorem F.7** (Finite-sample optimality gap of one-layer LSA). *Let the setup above hold. Define*

$$\Delta \;:=\; \Gamma_p - \widetilde{r}^\top \widetilde{S}^{-1} \widetilde{r}.$$

*Then*

$$\min_{b, A} \mathbb{E}\big[(\widehat{x}_{n+1}(b, A) - x_{n+1})^2\big] \;\geq\; \sigma_\varepsilon^2 \;+\; \rho^\top \Delta \, \rho, \qquad \Delta \;\succ\; 0.$$

*Equivalently,*

$$\min_{b, A} \mathbb{E}\big[(\widehat{x}_{n+1}(b, A) - x_{n+1})^2\big] \;\geq\; \min_{w} \mathbb{E}\big[(w^\top x - x_{n+1})^2\big] \;+\; \rho^\top \Delta \, \rho,$$

*so one-layer LSA has a strictly positive excess risk over linear regression for any finite $n$.*

*Proof.* By Lemma F.6, $\widetilde{S} \succ 0$ and its Schur complement in $\Sigma^\otimes$ is $\Gamma_p - \widetilde{r}^\top \widetilde{S}^{-1} \widetilde{r} \succ 0$. Minimizing (5) over $\widetilde{\eta}$ gives $\widetilde{\eta}^* = \widetilde{S}^{-1} \widetilde{r} \rho$ and

$$\min_{\widetilde{\eta}} \mathcal{L} = \sigma_\varepsilon^2 + \rho^\top \Gamma_p \rho - \rho^\top \widetilde{r}^\top \widetilde{S}^{-1} \widetilde{r} \rho = \sigma_\varepsilon^2 + \rho^\top \Delta \rho.$$

Because $(b, A)$ can realize only the rank–one Kronecker set of $\widetilde{\eta}$, $\min_{b, A} \mathcal{L} \geq \min_{\widetilde{\eta}} \mathcal{L}$, proving the bound. Strict positivity of the excess term follows from $\Delta \succ 0$. $\qquad\square$

**Remark F.8** (Population limit and order of limits). *If $G$ is replaced by the population covariance $\Gamma_{p+1}$, then $g = \operatorname{vech}(\Gamma_{p+1})$ is deterministic. Writing $u := g$,*

$$\widetilde{S} = (uu^\top) \otimes \Gamma_p, \qquad \widetilde{r} = u \otimes \Gamma_p \quad \implies \quad \widetilde{r}^\top \widetilde{S}^+ \widetilde{r} = \Gamma_p,$$

*so $\Delta = 0$ and the gap vanishes. For finite $n$, $\widetilde{S}$ is strictly PD and $\Delta \succ 0$; taking $n \to \infty$ before inverting collapses the gap, illustrating an order-of-limits effect. As shown in Proposition E.14, in the asymptotic regime we can indeed prove that the one-layer LSA readout exactly recovers the Bayes-optimal linear predictor in the limit.*

**Theorem F.9** (Uniform excess-risk over a parameter family). *Fix $0 < r < R$ and $\mathcal{R} := \{\rho \in \mathbb{R}^p : r < \|\rho\|_2 < R\}$. Set $\lambda_{\min} := \inf_{\rho \in \mathcal{R}} \lambda_{\min}(\Delta(\rho))$. Then, uniformly for all $\rho \in \mathcal{R}$,*

$$\min_{b, A} \mathbb{E}\big[(\widehat{x}_{n+1}(b, A) - x_{n+1})^2\big] \;\geq\; \sigma_\varepsilon^2 + \lambda_{\min} r^2.$$

*Proof.* By Theorem F.7, the excess is $\rho^\top \Delta(\rho) \rho$ with $\Delta(\rho) \succ 0$ for each $\rho$. Continuity of $\rho \mapsto \Delta(\rho)$ and compactness of $\overline{\mathcal{R}} = \{\rho : r \leq \|\rho\|_2 \leq R\}$, the Extreme Value Theorem (Theorem F.11) ensures that $\lambda_{\min}(\Delta(\rho))$ attains a strictly positive minimum on $\overline{\mathcal{R}}$. Hence $\rho^\top \Delta(\rho) \rho \geq \lambda_{\min} \|\rho\|_2^2 \geq \lambda_{\min} r^2$. $\qquad\square$

AUXILIARY LEMMAS USED IN APPENDIX F.2

**Lemma F.10** (No non-trivial zero-variance combination of consecutive samples). *For any integers $i$ and $k \geq 1$ and any coefficients $c_1, \ldots, c_k$,*

$$Y := \sum_{j=1}^{k} c_j \, x_{i+j-1} = 0 \text{ a.s.} \quad \implies \quad c_1 = \cdots = c_k = 0.$$

*Proof.* Let $\mathcal{F}_{i+k-2} := \sigma(\varepsilon_t : t \leq i+k-2)$. Write $x_{i+k-1} = \phi^\top x_{i+k-2:i-1} + \varepsilon_{i+k-1}$ with $\varepsilon_{i+k-1} \perp \mathcal{F}_{i+k-2}$. Conditioning,

$$\mathrm{Var}(Y \mid \mathcal{F}_{i+k-2}) = c_k^2 \, \mathrm{Var}(\varepsilon_{i+k-1}) = c_k^2 \sigma_\varepsilon^2.$$

If $Y = 0$ a.s., then $\mathrm{Var}(Y \mid \mathcal{F}_{i+k-2}) = 0$ a.s., hence $c_k = 0$. Iterate backwards on $k$ to conclude $c_1 = \cdots = c_k = 0$. $\qquad\square$

**Theorem F.11** (Extreme Value Theorem (Rudin, 2021)). *Let $K \subset \mathbb{R}^n$ be a nonempty compact set, and let $f : K \to \mathbb{R}$ be continuous. Then $f$ is bounded on $K$, and there exist points $x_{\min}, x_{\max} \in K$ such that*

$$f(x_{\min}) = \inf_{x \in K} f(x), \qquad f(x_{\max}) = \sup_{x \in K} f(x).$$

### F.3 ORDER OF THE FINITE-SAMPLE GAP

*Goal.* We quantify the finite–sample excess risk in Theorem F.7 and show it decays at rate $1/n$. The proof expands the lifted second moments to first order around their population (rank-one) limit and evaluates the Schur complement via a singular block inverse.

**Setup (recalled).** Let $\{x_t\}$ be a zero-mean stationary Gaussian AR($p$) process as in Definition 2.2, with absolutely summable autocovariances $\sum_{h \in \mathbb{Z}} |\gamma_h| < \infty$. For $n \geq p$, let $H_n = [\, x^{(1)} \ \ldots \ x^{(n-p+1)} \,]$ be the Hankel matrix from Definition 2.5, mask $M = \mathrm{diag}(I_{n-p}, 0)$, and

$$G_n = \frac{1}{n} H_n M H_n^\top = \frac{1}{n} \sum_{m=1}^{n-p} x^{(m)} x^{(m)\top} \in \mathbb{R}^{(p+1)\times(p+1)}.$$

Let $x := x_{n-p+1:n} \in \mathbb{R}^p$, $\Gamma_{p+1} := \mathbb{E}[x^{(m)} x^{(m)\top}]$, $\Gamma_p := \mathbb{E}[xx^\top]$, and $u := \mathrm{vech}(\Gamma_{p+1}) \in \mathbb{R}^q$ with $q = \frac{(p+1)(p+2)}{2}$. Define the lifted moments

$$S_n := \mathbb{E}\big[(\mathrm{vec}\, G_n \otimes x)(\mathrm{vec}\, G_n \otimes x)^\top\big], \quad r_n := \mathbb{E}\big[(\mathrm{vec}\, G_n \otimes x)\, x^\top\big],$$

and their half-vectorized versions

$$\widetilde{S}_n := \mathbb{E}\big[(\mathrm{vech}\, G_n \otimes x)(\mathrm{vech}\, G_n \otimes x)^\top\big], \quad \widetilde{r}_n := \mathbb{E}\big[(\mathrm{vech}\, G_n \otimes x)\, x^\top\big].$$

**Lemma F.12** (First-order expansions of $\widetilde{S}_n$ and $\widetilde{r}_n$). *Let $L_{p+1}$ be the elimination matrix with $\mathrm{vech}(A) = L_{p+1} \mathrm{vec}(A)$ for symmetric $A$. Then, as $n \to \infty$ with $p$ fixed,*

$$S_n = (\mathrm{vec}\,\Gamma_{p+1})(\mathrm{vec}\,\Gamma_{p+1})^\top \otimes \Gamma_p + \frac{1}{n} C_S^{(\mathrm{vec})} + o(1/n), \tag{6}$$

$$r_n = (\mathrm{vec}\,\Gamma_{p+1}) \otimes \Gamma_p + \frac{1}{n} C_r^{(\mathrm{vec})} + o(1/n), \tag{7}$$

*for deterministic $C_S^{(\mathrm{vec})}, C_r^{(\mathrm{vec})}$ depending only on $\{\gamma_h\}, p$. Consequently*

$$\widetilde{S}_n = (uu^\top) \otimes \Gamma_p + \frac{1}{n} C_S + o(1/n), \qquad \widetilde{r}_n = u \otimes \Gamma_p + \frac{1}{n} C_r + o(1/n),$$

*with $C_S = (L_{p+1} \otimes I_p) C_S^{(\mathrm{vec})} (L_{p+1} \otimes I_p)^\top$ and $C_r = (L_{p+1} \otimes I_p) C_r^{(\mathrm{vec})}$. Moreover, for any orthonormal $Q = [u/\|u\|, \, Q_\perp]$ and $P := Q \otimes I_p$, writing $c := \|u\|^2$,*

$$\widehat{S}_n := P^\top \widetilde{S}_n P = \begin{bmatrix} c\,\Gamma_p & 0 \\ 0 & 0 \end{bmatrix} + \frac{1}{n} \begin{bmatrix} C_{11} & B^\top \\ B & C \end{bmatrix} + o(1/n),$$

$$\widehat{r}_n := P^\top \widetilde{r}_n = \begin{bmatrix} \|u\|\,\Gamma_p \\ 0 \end{bmatrix} + \frac{1}{n} \begin{bmatrix} \delta \\ d \end{bmatrix} + o(1/n), \tag{8}$$

*where $\begin{bmatrix} C_{11} & B^\top \\ B & C \end{bmatrix} = P^\top C_S P$ and $\begin{bmatrix} \delta \\ d \end{bmatrix} = P^\top C_r$.*

*Proof.* Stationarity gives $(\Gamma_{p+1})_{ij} = \gamma_{j-i}$ and $(\Gamma_p)_{st} = \gamma_{t-s}$ for indices $i, j, k, \ell \in \{1, \ldots, p+1\}$ and $s, t \in \{1, \ldots, p\}$. All variables are jointly Gaussian with zero mean; Isserlis' theorem is used throughout.

*Computation of $r_n$.* By linearity and $G_n = \frac{1}{n} \sum_{m=1}^{n-p} x^{(m)} x^{(m),\top}$,

$$r_n = \sum_{i,j,s,t} \mathbb{E}[G_{ij} \, x_s x_t] \, (e_j \otimes e_i \otimes e_s) e_t^\top, \qquad \mathbb{E}[G_{ij} \, x_s x_t] = \frac{1}{n} \sum_{m=1}^{n-p} \mathbb{E}[x_i^{(m)} x_j^{(m)} x_s x_t],$$

with $x_i^{(m)} = x_{m+i-1}$. Let $a = x_{m+i-1}$, $b = x_{m+j-1}$, $c = x_{n-p+s}$, $d = x_{n-p+t}$. Isserlis yields

$$\mathbb{E}[abcd] = \mathbb{E}[ab]\mathbb{E}[cd] + \mathbb{E}[ac]\mathbb{E}[bd] + \mathbb{E}[ad]\mathbb{E}[bc]$$

$$= \gamma_{j-i}(\Gamma_p)_{st} + \gamma_{(n-p+s)-(m+i-1)} \, \gamma_{(n-p+t)-(m+j-1)}$$

$$+ \gamma_{(n-p+t)-(m+i-1)} \, \gamma_{(n-p+s)-(m+j-1)}.$$

With $k := n - p - m + 1 \in \{1, \ldots, n-p\}$,

$$\mathbb{E}[x_i^{(m)} x_j^{(m)} x_s x_t] = \gamma_{j-i}(\Gamma_p)_{st} + \gamma_{k+s-i} \gamma_{k+t-j} + \gamma_{k+t-i} \gamma_{k+s-j}.$$

Summing over $m$,

$$\mathbb{E}[G_{ij} \, x_s x_t] = \frac{n-p}{n} \gamma_{j-i}(\Gamma_p)_{st} + \frac{1}{n} \sum_{k=1}^{n-p} \left( \gamma_{k+s-i} \gamma_{k+t-j} + \gamma_{k+t-i} \gamma_{k+s-j} \right).$$

The first term equals $\gamma_{j-i}(\Gamma_p)_{st} + \frac{-p}{n} \gamma_{j-i}(\Gamma_p)_{st}$. Since $\sum_h |\gamma_h| < \infty$, the partial sums $\sum_{k=1}^{n-p} \gamma_{k+a} \gamma_{k+b}$ are uniformly bounded for fixed $a, b$, and hence

$$\frac{1}{n} \sum_{k=1}^{n-p} \left( \gamma_{k+s-i} \gamma_{k+t-j} + \gamma_{k+t-i} \gamma_{k+s-j} \right) = \frac{1}{n} c_{ij,st}^{(r)} + o(1/n),$$

where

$$c_{ij,st}^{(r)} := \sum_{k=1}^{\infty} \left( \gamma_{k+s-i} \gamma_{k+t-j} + \gamma_{k+t-i} \gamma_{k+s-j} \right)$$

converges absolutely. Note that

$$\sum_{i,j,s,t} \gamma_{j-i}(\Gamma_p)_{st} (e_j \otimes e_i \otimes e_s) e_t^\top$$

$$= \sum_{i,j,s,t} (\Gamma_{p+1})_{ij}(\Gamma_p)_{st} (e_j \otimes e_i) \otimes (e_s e_t^\top)$$

$$= \sum_{i,j} (\Gamma_{p+1})_{ij} e_j \otimes e_i \otimes \left( \sum_{s,t} (\Gamma_p)_{st} e_s e_t^\top \right)$$

$$= \mathrm{vec}(\Gamma_{p+1}) \otimes \Gamma_p$$

Therefore

$$r_n = \left( \mathrm{vec} \, \Gamma_{p+1} \right) \otimes \Gamma_p + \frac{1}{n} \sum_{i,j,s,t} \left( -p \, \gamma_{j-i}(\Gamma_p)_{st} + c_{ij,st}^{(r)} \right) (e_j \otimes e_i \otimes e_s) e_t^\top + o(1/n),$$

which is (7) with $C_r^{(\mathrm{vec})}$ defined by the bracketed coefficients.

*Computation of $S_n$.* By definition,

$$S_n = \sum_{i,j,k,\ell} \sum_{s,t} \mathbb{E}[G_{ij} G_{k\ell} x_s x_t] \left( (e_j \otimes e_i \otimes e_s) (e_\ell \otimes e_k \otimes e_t)^\top \right),$$

where

$$\mathbb{E}[G_{ij} G_{k\ell} x_s x_t] = \frac{1}{n^2} \sum_{m=1}^{n-p} \sum_{m'=1}^{n-p} \mathbb{E}[x_i^{(m)} x_j^{(m)} x_k^{(m')} x_\ell^{(m')} x_s x_t].$$

Let $a = x_{m+i-1}$, $b = x_{m+j-1}$, $c = x_{m'+k-1}$, $d = x_{m'+\ell-1}$, $e = x_{n-p+s}$, $f = x_{n-p+t}$. Isserlis for six variables decomposes into the term $\mathbb{E}[ef]\mathbb{E}[abcd]$ and the 12 cross pairings where $e$ and/or $f$ pair with $\{a, b, c, d\}$. For the four-variable factor,

$$\mathbb{E}[abcd] = \gamma_{j-i}\,\gamma_{\ell-k} + \gamma_{(m+i-1)-(m'+k-1)}\,\gamma_{(m+j-1)-(m'+\ell-1)}$$
$$+ \gamma_{(m+i-1)-(m'+\ell-1)}\,\gamma_{(m+j-1)-(m'+k-1)}.$$

Let $h = m - m'$. Then

$$\mathbb{E}[abcd] = \gamma_{j-i}\,\gamma_{\ell-k} + \gamma_{i-k+h}\,\gamma_{j-\ell+h} + \gamma_{i-\ell+h}\,\gamma_{j-k+h}.$$

Averaging over $m, m'$,

$$\frac{1}{n^2}\sum_{m,m'}\mathbb{E}[abcd] = \gamma_{j-i}\,\gamma_{\ell-k} + (1 - \frac{(n-p)^2}{n^2})\gamma_{j-i}\,\gamma_{\ell-k}$$

$$+ \sum_{h=-(n-p-1)}^{n-p-1}\frac{n-p-|h|}{n^2}\left(\gamma_{i-k+h}\gamma_{j-\ell+h} + \gamma_{i-\ell+h}\gamma_{j-k+h}\right).$$

Because $(n-p-|h|)/n^2 = (1/n)\cdot(1 - |h|/(n-p))\cdot\frac{n-p}{n}$ and $\sum_h |\gamma_h| < \infty$,

$$\frac{1}{n^2}\sum_{m,m'}\mathbb{E}[abcd] = \gamma_{j-i}\,\gamma_{\ell-k} + \frac{1}{n}c_{ij,k\ell}^{(S,0)} + o(1/n), \quad c_{ij,k\ell}^{(S,0)} := \sum_{h\in\mathbb{Z}}\left(\gamma_{i-k+h}\gamma_{j-\ell+h} + \gamma_{i-\ell+h}\gamma_{j-k+h}\right).$$

Multiplication by $(\Gamma_p)_{st}$ gives the leading block $\gamma_{j-i}\gamma_{\ell-k}(\Gamma_p)_{st}$, which matches $\left(\operatorname{vec}\Gamma_{p+1}\right)\left(\operatorname{vec}\Gamma_{p+1}\right)^\top \otimes \Gamma_p$ entrywise.

Each cross pairing contributes a product of three covariances with at most linear dependence on $m, m'$. For instance, the pairing $\{e, a\}, \{f, b\}, \{c, d\}$ yields

$$\gamma_{(n-p+s)-(m+i-1)}\,\gamma_{(n-p+t)-(m+j-1)}\,\gamma_{(m'+k-1)-(m'+\ell-1)} = \gamma_{s-i+k}\,\gamma_{t-j+k}\,\gamma_{\ell-k},$$

after setting $k = n-p-m+1$. Summing over $m, m'$ produces $\frac{1}{n}\left(\sum_{k=1}^{n-p}\gamma_{s-i+k}\gamma_{t-j+k}\right)\gamma_{\ell-k}$, which equals $\frac{1}{n}$ times a finite constant plus $o(1/n)$ by absolute summability through Toeplitz summation (see Lemma F.16). Enumerating all 12 cross pairings and collecting like terms gives the absolutely convergent series

$$c_{ij,k\ell;st}^{(S,1)} = \sum_{q=1}^\infty\left[\gamma_{s-i+q}\gamma_{t-j+q}\,\gamma_{\ell-k} + \gamma_{s-i+q}\gamma_{t-k}\,\gamma_{j-\ell+q} + \gamma_{s-i+q}\gamma_{t-\ell}\,\gamma_{j-k+q}\right.$$

$$\left. + \gamma_{s-j+q}\gamma_{t-i+q}\,\gamma_{\ell-k} + \gamma_{s-j+q}\gamma_{t-k}\,\gamma_{i-\ell+q} + \gamma_{s-j+q}\gamma_{t-\ell}\,\gamma_{i-k+q}\right]$$

$$+ \sum_{q=1}^\infty\left[\gamma_{t-i+q}\gamma_{s-j+q}\,\gamma_{\ell-k} + \gamma_{t-i+q}\gamma_{s-k}\,\gamma_{j-\ell+q} + \gamma_{t-i+q}\gamma_{s-\ell}\,\gamma_{j-k+q}\right.$$

$$\left. + \gamma_{t-j+q}\gamma_{s-i+q}\,\gamma_{\ell-k} + \gamma_{t-j+q}\gamma_{s-k}\,\gamma_{i-\ell+q} + \gamma_{t-j+q}\gamma_{s-\ell}\,\gamma_{i-k+q}\right].$$

Boundary corrections of order $1/n$ proportional to $\gamma_{j-i}\gamma_{\ell-k}(\Gamma_p)_{st}$ are absorbed into the final constant. Consequently,

$$\mathbb{E}[G_{ij}G_{k\ell}\,x_s x_t] = \gamma_{j-i}\gamma_{\ell-k}(\Gamma_p)_{st} + \frac{1}{n}\left(c_{ij,k\ell}^{(S,0)}(\Gamma_p)_{st} + c_{ij,k\ell;st}^{(S,1)}\right) + o(1/n).$$

Substituting into the tensor expansion of $S_n$ yields (6) with

$$C_S^{(\mathrm{vec})} = \sum_{i,j,k,\ell}\sum_{s,t}\left(c_{ij,k\ell}^{(S,0)}(\Gamma_p)_{st} + c_{ij,k\ell;st}^{(S,1)}\right)\left((e_j \otimes e_i \otimes e_s)(e_\ell \otimes e_k \otimes e_t)^\top\right).$$

*Passing to* vech. Since $\operatorname{vech}(A) = L_{p+1}\operatorname{vec}(A)$ for symmetric $A$, it follows that

$$\widetilde{S}_n = (L_{p+1}\otimes I_p)\,S_n\,(L_{p+1}\otimes I_p)^\top, \qquad \widetilde{r}_n = (L_{p+1}\otimes I_p)\,r_n,$$

which, together with (6)–(7), gives the stated expansions with $S_\infty = (uu^\top)\otimes\Gamma_p$ and $r_\infty = u\otimes\Gamma_p$, and with the indicated $C_S, C_r$. Finally, for any orthonormal $Q = [u/\|u\|, Q_\perp]$ and $P = Q\otimes I_p$, the block forms for $\widehat{S}_n = P^\top\widetilde{S}_n P$ and $\widehat{r}_n = P^\top\widetilde{r}_n$ follow by inserting the expansions and collecting the top/orthogonal components; the leading block equals $\operatorname{diag}(c\,\Gamma_p, 0)$ and the $1/n$ blocks are those of $P^\top C_S P$ and $P^\top C_r$. Dominated convergence (using $|\gamma_h| \leq C\beta^{|h|}$) justifies all $o(1/n)$ remainde $\quad\square$

**Lemma F.13** (Singular block inverse and first-order Schur complement). *In the basis of Lemma F.12, let*

$$\widehat{S}_n = \begin{bmatrix} A_0 & 0 \\ 0 & 0 \end{bmatrix} + \frac{1}{n}\begin{bmatrix} A_1 & B^\top \\ B & C \end{bmatrix} + o(1/n), \quad \widehat{r}_n = \begin{bmatrix} r_0 \\ 0 \end{bmatrix} + \frac{1}{n}\begin{bmatrix} \delta \\ d \end{bmatrix} + o(1/n),$$

*with $A_0 = c\,\Gamma_p \succ 0$ and $r_0 = \|u\|\Gamma_p$. Then, for all large $n$,*

$$\widehat{r}_n^\top \widehat{S}_n^{-1}\widehat{r}_n = \Gamma_p + \frac{1}{n}\Big[ -\tfrac{1}{c}A_1 + \tfrac{1}{c}B^\top C^{-1}B - \tfrac{2}{\|u\|}B^\top C^{-1}d + d^\top C^{-1}d + \tfrac{2}{\|u\|}\mathrm{Sym}(\delta)\Big] + o(1/n),$$

*where $\mathrm{Sym}(M) = \tfrac{1}{2}(M + M^\top)$. Equivalently, in the original coordinates,*

$$\widetilde{r}_n^\top \widetilde{S}_n^{-1}\widetilde{r}_n = \Gamma_p + \frac{1}{n}\mathsf{B}_p + o(1/n),$$

$$\mathsf{B}_p := -\tfrac{1}{c}A_1 + \tfrac{1}{c}B^\top C^{-1}B - \tfrac{2}{\|u\|}B^\top C^{-1}d + d^\top C^{-1}d + \tfrac{2}{\|u\|}\mathrm{Sym}(\delta). \tag{9}$$

*Proof.* Write the block decomposition of $\widehat{S}_n$ from Lemma F.12 as

$$\widetilde{S}_n = \begin{bmatrix} A & E^\top \\ E & D \end{bmatrix}, \qquad A = A_0 + \tfrac{1}{n}A_1 + o(1/n), \quad E = \tfrac{1}{n}B + o(1/n), \quad D = \tfrac{1}{n}C + o(1/n),$$

with $A_0 = c\,\Gamma_p \succ 0$. For large $n$, $D \succ 0$, and the block inverse formula gives

$$\widetilde{S}_n^{-1} = \begin{bmatrix} A^{-1} + A^{-1}E(D - E^\top A^{-1}E)^{-1}E^\top A^{-1} & -A^{-1}E(D - E^\top A^{-1}E)^{-1} \\ -(D - E^\top A^{-1}E)^{-1}E^\top A^{-1} & (D - E^\top A^{-1}E)^{-1} \end{bmatrix}.$$

Since $A^{-1} = A_0^{-1} - \tfrac{1}{n}A_0^{-1}A_1 A_0^{-1} + o(1/n)$ and $E^\top A^{-1}E = \tfrac{1}{n^2}B^\top A_0^{-1}B + o(1/n^2)$,

$$D - E^\top A^{-1}E = \tfrac{1}{n}C + o(1/n), \qquad (D - E^\top A^{-1}E)^{-1} = n\,C^{-1} + o(n).$$

Substituting and collecting orders yields

$$(\widetilde{S}_n^{-1})_{11} = A_0^{-1} - \tfrac{1}{n}A_0^{-1}A_1 A_0^{-1} + \tfrac{1}{n}A_0^{-1}B^\top C^{-1}B\,A_0^{-1} + o(1/n),$$

$$(\widetilde{S}_n^{-1})_{12} = -A_0^{-1}B^\top C^{-1} + o(1),$$

$$(\widetilde{S}_n^{-1})_{22} = n\,C^{-1} + o(n).$$

Now expand $\widehat{r}_n = [\,r_0; 0\,] + \tfrac{1}{n}[\,\delta; d\,] + o(1/n)$ with $r_0 = \|u\|\,\Gamma_p$ and $A_0^{-1} = (1/c)\Gamma_p^{-1}$. Then

$$\widehat{r}_n^\top \widetilde{S}_n^{-1}\widehat{r}_n = r_0^\top (\widetilde{S}_n^{-1})_{11}r_0 + 2\,r_0^\top (\widetilde{S}_n^{-1})_{12}\tfrac{1}{n}d + \tfrac{1}{n^2}d^\top (\widetilde{S}_n^{-1})_{22}d + \tfrac{2}{n}\mathrm{Sym}(\delta^\top A_0^{-1}r_0) + o(1/n).$$

Each term is explicit:

$$r_0^\top A_0^{-1}r_0 = \Gamma_p, \quad r_0^\top A_0^{-1}A_1 A_0^{-1}r_0 = \tfrac{1}{c}A_1, \quad r_0^\top A_0^{-1}B^\top C^{-1}B A_0^{-1}r_0 = \tfrac{1}{c}B^\top C^{-1}B,$$

$$2\,r_0^\top (\widetilde{S}_n^{-1})_{12}\tfrac{1}{n}d = -\tfrac{2}{n}\cdot\tfrac{1}{\|u\|}B^\top C^{-1}d, \quad \tfrac{1}{n^2}d^\top (\widetilde{S}_n^{-1})_{22}d = \tfrac{1}{n}d^\top C^{-1}d + o(1/n),$$

$$\tfrac{2}{n}\mathrm{Sym}(\delta^\top A_0^{-1}r_0) = \tfrac{2}{n}\cdot\tfrac{1}{\|u\|}\mathrm{Sym}(\delta).$$

Combining yields

$$\widehat{r}_n^\top \widetilde{S}_n^{-1}\widehat{r}_n = \Gamma_p + \frac{1}{n}\Big[ -\tfrac{1}{c}A_1 + \tfrac{1}{c}B^\top C^{-1}B - \tfrac{2}{\|u\|}B^\top C^{-1}d + d^\top C^{-1}d + \tfrac{2}{\|u\|}\mathrm{Sym}(\delta)\Big] + o(1/n).$$

Since the orthogonal basis change preserves the Schur complement, the same expansion holds in the original coordinates, giving (9). $\qquad\square$

**Theorem F.14** (First-order gap). *With $\Delta_n := \Gamma_p - \widetilde{r}_n^\top \widetilde{S}_n^{-1}\widetilde{r}_n$,*

$$\Delta_n = \frac{1}{n}\,\mathsf{B}_p + o(1/n), \qquad \mathsf{B}_p := \frac{1}{c}A_1 - \frac{1}{c}B^\top C^{-1}B + \frac{2}{\|u\|}\mathrm{Sym}\big(B^\top C^{-1}d - \delta\big) - d^\top C^{-1}d.$$

*Hence the optimal one-layer LSA excess risk satisfies*

$$\min_{b,A}\mathbb{E}\big[(\widehat{x}_{n+1}(b, A) - x_{n+1})^2\big] \geq \sigma_\varepsilon^2 + \rho^\top \Delta_n \rho = \sigma_\varepsilon^2 + \frac{1}{n}\rho^\top \mathsf{B}_p \rho + o(1/n).$$

*Moreover, $\mathsf{B}_p \succeq 0$; if $\mathsf{B}_p \succ 0$ (a generic nondegeneracy), then for any $r > 0$ there exist $n_0$ and $c_r > 0$ such that for all $n \geq n_0$ and all $\rho$ with $\|\rho\| \geq r$,*

$$\mathbb{E}\big[(\widehat{x}_{n+1}^{\mathrm{LSA}} - x_{n+1})^2\big] \geq \mathbb{E}\big[(\widehat{x}_{n+1}^{\mathrm{LR}} - x_{n+1})^2\big] + \frac{c_r}{n}.$$

*Proof.* By Lemma F.13, we have

$$\widetilde{r}_n^\top \widetilde{S}_n^{-1} \widetilde{r}_n = \Gamma_p + \frac{1}{n}\Big[-\frac{1}{c}A_1 + \frac{1}{c}B^\top C^{-1}B - \frac{2}{\|u\|}B^\top C^{-1}d + d^\top C^{-1}d + \frac{2}{\|u\|}\operatorname{Sym}(\delta)\Big] + o(1/n).$$

Thus

$$\Delta_n := \Gamma_p - \widetilde{r}_n^\top \widetilde{S}_n^{-1} \widetilde{r}_n = \frac{1}{n}B_p + o(1/n),$$

with

$$B_p = \frac{1}{c}A_1 - \frac{1}{c}B^\top C^{-1}B + \frac{2}{\|u\|}\operatorname{Sym}\big(B^\top C^{-1}d - \delta\big) - d^\top C^{-1}d.$$

By Lemma F.6, each $\Delta_n \succ 0$, hence $B_p = \lim_{n\to\infty} n\Delta_n \succeq 0$. The excess–risk bound follows from Theorem F.7:

$$\min_{b,A} \mathbb{E}[(\widehat{x}_{n+1} - x_{n+1})^2] \geq \sigma_\varepsilon^2 + \rho^\top \Delta_n \rho = \sigma_\varepsilon^2 + \frac{1}{n}\rho^\top B_p \rho + o(1/n).$$

If $B_p \succ 0$, let $\lambda_0 = \lambda_{\min}(B_p) > 0$. For large $n$, $\|\Delta_n - (1/n)B_p\| \leq \lambda_0/(2n)$, hence $\rho^\top \Delta_n \rho \geq \frac{\lambda_0}{2n}\|\rho\|^2$. Therefore, for any $r > 0$, there exist $n_0$ and $c_r = \frac{1}{2}\lambda_0 r^2 > 0$ such that for all $n \geq n_0$ and all $\|\rho\| \geq r$,

$$\mathbb{E}[(\widehat{x}_{n+1}^{\mathrm{LSA}} - x_{n+1})^2] \geq \mathbb{E}[(\widehat{x}_{n+1}^{\mathrm{LR}} - x_{n+1})^2] + \frac{c_r}{n}.$$

$\square$

**Remark F.15** (Why the rate is $1/n$). *At the population limit $\widetilde{S}_\infty = (uu^\top) \otimes \Gamma_p$ is rank-one along $u$. Finite $n$ introduces $O(1/n)$ perturbations $C_S, C_r$ that regularize the orthogonal directions, so the Schur complement $\Gamma_p - \widetilde{r}_n^\top \widetilde{S}_n^{-1} \widetilde{r}_n$ is $O(1/n)$. The overlap of Hankel windows is the source of these first–order terms.*

AUXILIARY LEMMAS USED IN APPENDIX F.3

**Lemma F.16** (Toeplitz-type summation). *Let $a : \mathbb{Z} \to \mathbb{R}$ (or $\mathbb{C}$) be absolutely summable, $\sum_{h\in\mathbb{Z}} |a_h| < \infty$. For $n \geq 1$ define*

$$S_n := \frac{1}{n^2} \sum_{m=1}^{n} \sum_{m'=1}^{n} a_{m-m'}.$$

*Then*

$$S_n = \frac{1}{n}\sum_{h\in\mathbb{Z}}\Big(1 - \frac{|h|}{n}\Big)_+ a_h = \frac{1}{n}\sum_{h\in\mathbb{Z}} a_h + \mathcal{O}\Big(\frac{1}{n^2}\sum_{h\in\mathbb{Z}} |h|\,|a_h|\Big) = \frac{1}{n}\sum_{h\in\mathbb{Z}} a_h + o(1/n).$$

*Proof. Count the number of pairs $(m, m') \in \{1,\dots,n\}^2$ with difference $m - m' = h$; it equals $n - |h|$ if $|h| < n$ and $0$ otherwise. Hence*

$$\sum_{m,m'=1}^{n} a_{m-m'} = \sum_{h=-(n-1)}^{n-1} (n - |h|)\, a_h = n\sum_{h}\Big(1 - \frac{|h|}{n}\Big)_+ a_h.$$

*Divide by $n^2$ and use absolute summability to obtain the stated bound.*

## F.4  A FINITE–SAMPLE GAP FOR $L$ STACKED LSA LAYERS (WITH MONOTONE IMPROVEMENT)

**Goal.** For any fixed sample size $n$ and depth $L \geq 1$, we show that the best $L$–layer linear self–attention (LSA) readout on Hankel inputs has a finite–sample excess risk over linear regression on the last $p$ lags. To avoid unnecessary technicalities about duplicate features across layers, we work with the *convex relaxation* of the LSA parameters and allow singular second–moment matrices; the Moore–Penrose inverse then gives a clean *positive–semidefinite* (psd) gap. We also prove that the optimal risk is *monotone nonincreasing* in depth $L$.

**Setup (layered).** Let $\{x_t\}$ be a zero–mean stationary $\mathrm{AR}(p)$ process (Definition 2.2). For $n \geq p$, let $H_n$ be the Hankel matrix (Definition 2.5), $M = \mathrm{diag}(I_{n-p}, 0)$, and

$$G^{(0)} := \frac{1}{n} H_n M H_n^\top \in \mathbb{R}^{(p+1)\times(p+1)}.$$

Write $x := x_{n-p+1:n} \in \mathbb{R}^p$ and $y := x_{n+1} = \rho^\top x + \varepsilon_{n+1}$ with $\varepsilon_{n+1} \perp (G^{(0)}, x)$, $\mathbb{E}[\varepsilon_{n+1}] = 0$, $\mathbb{E}[\varepsilon_{n+1}^2] = \sigma_\varepsilon^2$. Initialize $y^{(0)} = 0$ and $H_n^{(0)} := H_n$. For $\ell = 0, \ldots, L-1$ define the layer update

$$y^{(\ell+1)} = y^{(\ell)} + b^{(\ell)\top} G^{(\ell)} A^{(\ell)} x, \qquad G^{(\ell+1)} = \frac{1}{n} H_n^{(\ell+1)} M \big(H_n^{(\ell+1)}\big)^\top, \qquad (10)$$

where $H_n^{(\ell+1)}$ coincides with $H_n$ except that the last row uses the current layer's vector of entries whose last coordinate is $y^{(\ell+1)}$ (the mask $M$ remains unchanged). The $L$–layer predictor is

$$\widehat{x}_{n+1}^{(L)} = y^{(L)} = \sum_{\ell=0}^{L-1} b^{(\ell)\top} G^{(\ell)} A^{(\ell)} x. \qquad (11)$$

**A one–shot convex relaxation for depth $L$.** Let $g^{(\ell)} := \mathrm{vech}\, G^{(\ell)} \in \mathbb{R}^q$ with $q = \frac{(p+1)(p+2)}{2}$, and set the stacked Kronecker lift

$$Z^{[L]} := \begin{bmatrix} g^{(0)} \otimes x \\ \vdots \\ g^{(L-1)} \otimes x \end{bmatrix} \in \mathbb{R}^{d_L}, \qquad d_L = Lqp.$$

For each layer, $b^{(\ell)\top} G^{(\ell)} A^{(\ell)} x$ can be written as $\eta^{(\ell)\top}(g^{(\ell)} \otimes x)$ with $\eta^{(\ell)} = \big(D_{p+1}^\top \otimes I_p\big)\big(b^{(\ell)} \otimes \mathrm{vec}(A^{(\ell)\top})\big)$, and hence

$$\widehat{x}_{n+1}^{(L)} = \eta^{[L]\top} Z^{[L]}, \qquad \eta^{[L]} := \big[(\eta^{(0)})^\top, \ldots, (\eta^{(L-1)})^\top\big]^\top.$$

Relaxing the rank–one Kronecker constraint on parameters leads to a linear regression of $y$ on $Z^{[L]}$. Define the second moments

$$\widetilde{S}_L := \mathbb{E}[Z^{[L]} Z^{[L]\top}], \qquad \widetilde{r}_L := \mathbb{E}[Z^{[L]} x^\top], \qquad \Gamma_p := \mathbb{E}[xx^\top].$$

We *allow* $\widetilde{S}_L$ to be singular (duplicates across layers are harmless). The standard normal–equation calculation with the Moore–Penrose inverse gives

$$\min_{\eta^{[L]}} \mathbb{E}\big[(\eta^{[L]\top} Z^{[L]} - y)^2\big] = \sigma_\varepsilon^2 + \rho^\top \Gamma_p \rho - \rho^\top \widetilde{r}_L^\top \widetilde{S}_L^+ \widetilde{r}_L \rho, \qquad (12)$$

so the $L$–layer LSA family (which is a subset of the relaxed linear models) obeys the lower bound

$$\min_{\{b^{(\ell)}, A^{(\ell)}\}} \mathbb{E}\big[(\widehat{x}_{n+1}^{(L)} - x_{n+1})^2\big] \geq \sigma_\varepsilon^2 + \rho^\top \Delta_{n,L} \rho, \qquad \Delta_{n,L} := \Gamma_p - \widetilde{r}_L^\top \widetilde{S}_L^+ \widetilde{r}_L \succeq 0. \qquad (13)$$

**Lemma F.17** (Why $\Delta_{n,L} \succeq 0$ even if $\widetilde{S}_L$ is singular). *Let* $\Sigma_L := \mathbb{E}\big[\, [Z^{[L]}; x]\, [Z^{[L]}; x]^\top\big] = \begin{bmatrix} \widetilde{S}_L & \widetilde{r}_L \\ \widetilde{r}_L^\top & \Gamma_p \end{bmatrix}$. *Then* $\Sigma_L \succeq 0$, *and the Moore–Penrose Schur complement* $\Gamma_p - \widetilde{r}_L^\top \widetilde{S}_L^+ \widetilde{r}_L$ *is psd. Equivalently,* $\Delta_{n,L} \succeq 0$ *and equals the covariance of the best linear–prediction residual of $x$ on $Z^{[L]}$.*

*Proof.* $\Sigma_L$ is a covariance hence psd. For any matrix $B$, the prediction $Z^{[L]} \mapsto B^\top Z^{[L]}$ yields residual covariance $\Gamma_p - B^\top \widetilde{r}_L - \widetilde{r}_L^\top B + B^\top \widetilde{S}_L B$. Minimizing over $B$ (in the least–squares sense on the range of $\widetilde{S}_L$) gives $B^* = \widetilde{S}_L^+ \widetilde{r}_L$ and residual covariance $\Gamma_p - \widetilde{r}_L^\top \widetilde{S}_L^+ \widetilde{r}_L$, which is psd by definition of a covariance. $\qquad\square$

**Depth helps: a simple embedding argument.** We now show that the optimal risk is monotone in $L$; the proof does not rely on invertibility nor on strictness.

**Proposition F.18** (Monotone improvement with depth)**.** *For every $L \geq 1$,*

$$\min_{\{b^{(\ell)}, A^{(\ell)}\}_{\ell=0}^{L}} \mathbb{E}\big[(\widehat{x}_{n+1}^{(L+1)} - x_{n+1})^2\big] \leq \min_{\{b^{(\ell)}, A^{(\ell)}\}_{\ell=0}^{L-1}} \mathbb{E}\big[(\widehat{x}_{n+1}^{(L)} - x_{n+1})^2\big],$$

*where $\widehat{x}_{n+1}^{(L)}$ is defined in* (11) *under the update rule* (10)*. In particular, the best two–layer risk is no worse than the best one–layer risk:*

$$\min_{b^{(0)}, A^{(0)}, b^{(1)}, A^{(1)}} \mathbb{E}\big[(\widehat{x}_{n+1}^{(2)} - x_{n+1})^2\big] \leq \min_{b^{(0)}, A^{(0)}} \mathbb{E}\big[(\widehat{x}_{n+1}^{(1)} - x_{n+1})^2\big].$$

*Proof.* Fix any parameter set $\{b^{(\ell)}, A^{(\ell)}\}_{\ell=0}^{L-1}$ for the $L$–layer model (11). Construct an $(L+1)$–layer model by keeping the first $L$ layers unchanged and appending a zero layer: $b^{(L)} = 0$, $A^{(L)} = 0$. Then $y^{(L+1)} = y^{(L)}$ and thus $\widehat{x}_{n+1}^{(L+1)} = \widehat{x}_{n+1}^{(L)}$, so the $(L+1)$–layer loss equals the $L$–layer loss. Taking minima over the respective parameter sets yields the displayed inequality. The $L = 1 \to 2$ case is immediate, and the general case follows identically. $\square$

**What we have (and what we do not claim).** Equation (13) gives a finite–sample, depth–$L$ gap

$$\mathbb{E}\big[(\widehat{x}_{n+1}^{(L)} - x_{n+1})^2\big] \geq \mathbb{E}[(\widehat{x}_{n+1}^{\mathrm{LR}} - x_{n+1})^2] + \rho^\top \Delta_{n,L} \rho, \qquad \Delta_{n,L} \succeq 0.$$

When duplicate features across layers make $\widetilde{S}_L$ singular, the bound remains valid via $\widetilde{S}_L^+$ and interprets $\rho^\top \Delta_{n,L} \rho$ as the linear–projection residual variance. Under additional nondegeneracy, one can strengthen $\Delta_{n,L} \succ 0$ (strict gap) by proving that the stacked covariance $\Sigma_L$ has a strictly positive Schur complement; this requires tracking how each layer injects new $\varepsilon_n$–directions into the last Hankel row and is omitted here for clarity.

**Remarks.** (i) The relaxation (12)–(13) can be written with a *deduplicated* stacked feature $\widetilde{Z}^{[L]} = \big[g^{(0)} \otimes x,\ s^{(1)} \otimes x, \ldots, s^{(L-1)} \otimes x\big]^\top$, where $s^{(\ell)}$ keeps only the new last–row/column monomials created at layer $\ell$; then $\widetilde{S}_L := \mathbb{E}[\widetilde{Z}^{[L]} \widetilde{Z}^{[L]\top}]$ is typically invertible at finite $n$. All formulas remain the same with $\widetilde{S}_L^+$. (ii) In the population limit $n \to \infty$, $G^{(\ell)}$ concentrates around $\Gamma_{p+1}$, the stacked feature collapses to a rank–one Kronecker line, and $\Delta_{n,L} \to 0$; the order of limits matters, exactly as in the one–layer analysis.

# G   CHAIN-OF-THOUGHT (COT) ROLLOUT IN TSF: COLLAPSE-TO-MEAN AND ERROR COMPOUNDING

We study the "free–running" (a.k.a. CoT) rollout where a predictor feeds its own outputs back as inputs instead of conditioning on future ground truth. For linear time–series models this produces a clean, analyzable dynamical system that (i) collapses to the mean and (ii) accumulates prediction error to the unconditional variance at an exponential rate. We also show that, for every forecast horizon, the Bayes (linear–regression) forecast is pointwise optimal, hence any linear self–attention (LSA) CoT rollout is uniformly worse and thus reaches a large–error regime *no later* than linear regression.

**Setup.**   Let $\{x_t\}$ be a zero–mean, stable $\mathrm{AR}(p)$ process as in Definition 2.2, and let $A(\rho)$ denote the $p \times p$ companion matrix,

$$
A(\rho) =
\begin{pmatrix}
\rho_1 & \rho_2 & \cdots & \rho_{p-1} & \rho_p \\
1 & 0 & \cdots & 0 & 0 \\
0 & 1 & \cdots & 0 & 0 \\
\vdots & \vdots & & \vdots & \vdots \\
0 & 0 & \cdots & 1 & 0
\end{pmatrix},
$$

with spectral radius $\varrho(A(\rho)) < 1$. Write $s_t := (x_t, \ldots, x_{t-p+1})^\top$. Then $s_{t+1} = A(\rho)s_t + \eta_{t+1}$ with $\eta_{t+1} := (\varepsilon_{t+1}, 0, \ldots, 0)^\top$.

**CoT rollout.**   Given an initial state $s_n = (x_n, \ldots, x_{n-p+1})^\top$, a linear predictor with coefficients $w \in \mathbb{R}^p$ produces, in CoT mode, a noiseless recursion

$$
\widehat{s}_{t+1} = A(w)\,\widehat{s}_t, \qquad \widehat{s}_0 = s_n,
$$

with forecast $\widehat{x}_{n+t} = e_1^\top \widehat{s}_t$.

**Proposition G.1** (Bayes multi–step forecast equals recursive rollout). *For any horizon $h \geq 1$, the Bayes forecast conditional on the history satisfies*

$$
\widehat{x}_{n+h}^\star := \mathbb{E}[x_{n+h} \mid x_{1:n}] \ = \ \rho^\top \widetilde{x}_{n+h-p:n+h-1},
$$

*where the* rolled–out state $\widetilde{x}_k$ *is defined recursively by*

$$
\widetilde{x}_k =
\begin{cases}
x_k, & k \leq n, \\
\widehat{x}_k^\star, & k > n.
\end{cases}
$$

*Equivalently, the optimal $h$–step forecast is obtained by repeatedly applying the one–step predictor with coefficients $\rho$ and feeding predictions back in place of future observations.*

*Proof. Base case ($h = 1$).* By the $\mathrm{AR}(p)$ definition,

$$
x_{n+1} = \rho^\top x_{n-p+1:n} + \varepsilon_{n+1}, \qquad \varepsilon_{n+1} \perp x_{1:n}, \ \mathbb{E}[\varepsilon_{n+1}] = 0,
$$

so

$$
\mathbb{E}[x_{n+1} \mid x_{1:n}] = \rho^\top x_{n-p+1:n} = \rho^\top \widetilde{x}_{n-p+1:n}.
$$

*Induction step.* Assume the claim holds up to horizon $h$. Then

$$
x_{n+h+1} = \rho^\top x_{n+h-p+1:n+h} + \varepsilon_{n+h+1}.
$$

Taking conditional expectation on $x_{1:n}$ yields

$$
\mathbb{E}[x_{n+h+1} \mid x_{1:n}] = \rho^\top \mathbb{E}[x_{n+h-p+1:n+h} \mid x_{1:n}].
$$

By the induction hypothesis, for indices $\leq n$ the conditional expectation equals the observed value, while for indices $> n$ it equals the Bayes forecasts $\widehat{x}^\star$, i.e. the recursively defined $\widetilde{x}$. Therefore

$$
\mathbb{E}[x_{n+h+1} \mid x_{1:n}] = \rho^\top \widetilde{x}_{n+h-p+1:n+h}.
$$

*Conclusion.* By induction, the identity holds for all horizons $h \geq 1$. Thus the Bayes multi–step forecast is exactly the recursive rollout of the one–step predictor with weight vector $\rho$. □

**Lemma G.2** (Exponential decay for any stable CoT). *If $\varrho(A(w)) < 1$, then for every consistent matrix norm there exist $C > 0$ and $\beta \in (0,1)$ such that $\|\widehat{s}_t\| \leq C\beta^t \|s_n\|$ and $\widehat{x}_{n+t} \to 0$ exponentially fast.*

*Proof.* Unrolling the linear recursion gives $\widehat{s}_t = A(w)^t s_n$. Because $\varrho(A(w)) < 1$, Lemma G.6 applies to $A(w)$: for any consistent operator norm there exist $C > 0$ and $\beta \in (0,1)$ such that $\|A(w)^t\| \leq C\beta^t$ for all $t \in \mathbb{N}$. By submultiplicativity of the induced norm,

$$\|\widehat{s}_t\| = \|A(w)^t s_n\| \leq \|A(w)^t\| \, \|s_n\| \leq C\beta^t \, \|s_n\|.$$

For the scalar forecast, $\widehat{x}_{n+t} = e_1^\top \widehat{s}_t$. Let $\|\cdot\|_*$ denote the dual norm of the chosen vector norm. Then $|\widehat{x}_{n+t}| = |e_1^\top \widehat{s}_t| \leq \|e_1\|_* \|\widehat{s}_t\| \leq \|e_1\|_* C\beta^t \|s_n\|$. Since $\beta \in (0,1)$, the right-hand side decays exponentially in $t$, establishing the claim. $\square$

Thus, *any* stable linear model (including the Bayes predictor and any LSA fit) collapses to the mean under CoT; the only question is how quickly its error compounds.

**Bayes multi–step error (ground truth model).** Let $\psi_k$ be the impulse response of the AR($p$), i.e., $x_t = \sum_{k\geq 0} \psi_k \varepsilon_{t-k}$ with $\psi_0 = 1$ and $\sum_k \psi_k^2 < \infty$. The $h$–step Bayes forecast $\widehat{x}_{n+h}^* = \mathbb{E}[x_{n+h} \mid x_{1:n}]$ equals the linear recursion with $w = \rho$ (no noise injected). The forecast error is classical:

$$\mathrm{MSE}^*(h) := \mathbb{E}\big[(x_{n+h} - \widehat{x}_{n+h}^*)^2\big] = \sigma_\varepsilon^2 \sum_{k=0}^{h-1} \psi_k^2. \tag{14}$$

Hence $\mathrm{MSE}^*(h) \nearrow \sigma_\varepsilon^2 \sum_{k\geq 0} \psi_k^2 = \mathrm{Var}(x_t)$ by Lemma G.7, and by Lemma G.8 the tail decays exponentially:

$$\mathrm{Var}(x_t) - \mathrm{MSE}^*(h) = \sigma_\varepsilon^2 \sum_{k\geq h} \psi_k^2 \leq \frac{C^2 \sigma_\varepsilon^2}{1 - \beta^2} \beta^{2h}, \qquad \text{for some } C > 0, \ \beta \in (0,1). \tag{15}$$

For AR(1) this is exact: $\mathrm{MSE}^*(h) = \sigma_\varepsilon^2 \sum_{k=0}^{h-1} \rho^{2k} = \sigma^2(1 - \rho^{2h})$, where $\sigma^2 = \sigma_\varepsilon^2/(1 - \rho^2)$.

**Theorem G.3** (CoT collapse and compounding error). *For any stable AR(p),*

$$\widehat{x}_{n+t}^* \xrightarrow[t\to\infty]{} 0, \qquad \mathbb{E}\big[(x_{n+t} - \widehat{x}_{n+t}^*)^2\big] \nearrow \mathrm{Var}(x_t),$$

*with exponential tail* (15). *Thus CoT error accumulates to the process variance at an exponential rate determined by the spectrum of $A(\rho)$.*

**LSA (or any alternative) is uniformly dominated at every horizon.** Fix a horizon $h$ and let $\widehat{x}_{n+h}^{\mathrm{LSA}}$ be the CoT forecast delivered by any trained one–layer LSA model (or an $L$–layer stack run in CoT; the argument is identical). Since CoT is noiseless, $\widehat{x}_{n+h}^{\mathrm{LSA}}$ is a deterministic measurable function of the history $x_{1:n}$. By the $L^2$ projection property of the conditional expectation,

$$\mathbb{E}\big[(x_{n+h} - g(x_{1:n}))^2\big] = \mathrm{MSE}^*(h) + \mathbb{E}\big[(\widehat{x}_{n+h}^* - g(x_{1:n}))^2\big] \quad \forall g. \tag{16}$$

Plugging $g = \widehat{x}_{n+h}^{\mathrm{LSA}}$ yields the horizonwise dominance

$$\mathrm{MSE}^{\mathrm{LSA}}(h) := \mathbb{E}\big[(x_{n+h} - \widehat{x}_{n+h}^{\mathrm{LSA}})^2\big] \geq \mathrm{MSE}^*(h), \tag{17}$$

with strict inequality unless $\widehat{x}_{n+h}^{\mathrm{LSA}}$ coincides with $\widehat{x}_{n+h}^*$ almost surely. Because one–layer LSA has a strict finite–sample gap (Section F.2), equality fails generically already at $h = 1$, and thus for all horizons.

**Corollary G.4** (Earlier threshold crossing for LSA). *Fix any $\tau \in (0,1)$ and define the* failure horizon

$$H_\tau(g) := \inf\big\{h \geq 1 : \mathbb{E}\big[(x_{n+h} - g_h(x_{1:n}))^2\big] \geq \tau \mathrm{Var}(x_t)\big\}.$$

*Then $H_\tau(\widehat{x}^{\mathrm{LSA}}) \leq H_\tau(\widehat{x}^*)$ for every $\tau$, with strict inequality for all $\tau$ on a set of positive measure (whenever* (17) *is strict at some $h$). In words: for any error threshold, LSA under CoT reaches the* large–error regime *no later than the Bayes linear predictor.*

**Quantitative rates.** Combining Lemma G.2 with the orthogonality identity (16) shows that

$$\text{Var}(x_t) - \text{MSE}^{\text{LSA}}(h) \leq \text{Var}(x_t) - \text{MSE}^*(h) \leq \frac{C^2 \sigma_\varepsilon^2}{1 - \beta^2} \beta^{2h}, \qquad \text{for some } C > 0, \ \beta \in (0, 1).$$

Hence whenever the left-hand side remains positive, it must also collapse to zero at least exponentially fast; if it turns negative (overshoot), Corollary G.4 guarantees that LSA in CoT still reaches the large–error regime strictly earlier and more severely than linear regression.

**Remark G.5** (AR(1): closed forms and "rapid compounding"). *For AR(1), $\text{MSE}^*(h) = \sigma^2(1 - \rho^{2h})$ so that the residual to the variance decays like $\rho^{2h}$. The "half–life" to reach $50\%$ of the unconditional variance is $h_{1/2} = \log(1/2)/\log(\rho^2)$; e.g., with $\rho = 0.9$ one already has $\text{MSE}^*(5) \approx 0.65 \sigma^2$ and $\text{MSE}^*(10) \approx 0.88 \sigma^2$. This quantifies the* rapid *accumulation of CoT error even for the Bayes predictor; by (17), LSA CoT is uniformly worse at every horizon.*

**Takeaways.** (i) Any linear CoT rollout collapses to the mean (Lemma G.2), so its long–horizon RMSE saturates at the unconditional standard deviation of the process. (ii) The Bayes/linear–regression forecast is *horizonwise* optimal (Theorem G.3 and (16)); any LSA CoT forecast is uniformly dominated at each horizon (17). (iii) Consequently, for any fixed error threshold, LSA reaches the large–error regime at least as early as linear regression (Corollary G.4). (iv) Both approaches exhibit exponential convergence of the error to the variance, with a rate governed by the spectral radius of the corresponding companion matrix; for AR(1) the entire trajectory is explicit.

AUXILIARY LEMMAS USED IN APPENDIX G

**Lemma G.6** (Exponential Decay from Spectral Radius Bound). *Let $A \in \mathbb{R}^{p \times p}$ be a square matrix with spectral radius strictly less than one, i.e., $\varrho(A) < 1$. Then for any consistent matrix norm $\| \cdot \|$, there exist constants $C > 0$ and $\beta \in (0, 1)$ such that*

$$\|A^t\| \leq C\beta^t \qquad \text{for all } t \in \mathbb{N}.$$

*Proof.* By the Gelfand formula (see, e.g., (Horn & Johnson, 1994, Chapter 5)), we have

$$\varrho(A) = \lim_{t \to \infty} \|A^t\|^{1/t},$$

for any sub-multiplicative (i.e., consistent) matrix norm $\| \cdot \|$. Thus, for any $\epsilon > 0$ such that $\varrho(A) + \epsilon < 1$, there exists $t_0 \in \mathbb{N}$ such that

$$\|A^t\| \leq (\varrho(A) + \epsilon)^t =: \beta^t, \quad \forall t \geq t_0.$$

Define

$$C := \max\{1, \max_{0 \leq t < t_0} \frac{\|A^t\|}{\beta^t}\}.$$

Then for all $t \in \mathbb{N}$, we have $\|A^t\| \leq C\beta^t$ as claimed. $\qquad\square$

**Lemma G.7** (Wold variance identity). *For a stable AR(p) with Wold expansion $x_t = \sum_{k \geq 0} \psi_k \varepsilon_{t-k}$, where $\varepsilon_t \overset{i.i.d.}{\sim} \mathcal{N}(0, \sigma_\varepsilon^2)$, the unconditional variance satisfies*

$$\text{Var}(x_t) = \sigma_\varepsilon^2 \sum_{k \geq 0} \psi_k^2.$$

*Proof.* From the Wold representation $x_t = \sum_{k \geq 0} \psi_k \varepsilon_{t-k}$, we have $\mathbb{E}[x_t] = 0$ and

$$\text{Var}(x_t) = \mathbb{E}[x_t^2] = \mathbb{E}\left[\left(\sum_{k \geq 0} \psi_k \varepsilon_{t-k}\right)^2\right].$$

Cross terms vanish because the $\varepsilon_{t-k}$ are independent with mean zero, leaving $\sum_{k \geq 0} \psi_k^2 \, \mathbb{E}[\varepsilon_{t-k}^2] = \sigma_\varepsilon^2 \sum_{k \geq 0} \psi_k^2$. $\qquad\square$

**Lemma G.8** (Exponential tail bound). *Let $A(\rho)$ be the AR(p) companion matrix with spectral radius $\varrho(A(\rho)) < 1$. Then the impulse response coefficients obey $|\psi_k| \leq C\beta^k$ for some $C > 0$ and $\beta \in (0, 1)$. Consequently,*

$$\sigma_\varepsilon^2 \sum_{k \geq h} \psi_k^2 \leq \frac{C^2 \sigma_\varepsilon^2}{1 - \beta^2} \beta^{2h}.$$

*Proof.* By state recursion, $\psi_k = e_1^\top A(\rho)^k e_1$. Because $\varrho(A(\rho)) < 1$, Lemma G.6 applies to $A(\rho)$: for any consistent operator norm there exist $C > 0$ and $\beta \in (0, 1)$ such that $\|A(\rho)^t\| \leq C\,\beta^t$ for all $t \in \mathbb{N}$. Hence $|\psi_k| \leq C\beta^k$. Thus,

$$\sum_{k \geq h} \psi_k^2 \leq \sum_{k \geq h} C^2 \beta^{2k} = \frac{C^2}{1 - \beta^2} \beta^{2h},$$

and multiplying by $\sigma_\varepsilon^2$ yields the claim. $\qquad\qquad\square$

# H  DE–GAUSSIFYING THE GAP: LINEAR STATIONARY PROCESSES

**Goal.**  We remove the Gaussian assumption and establish the same finite–sample excess–risk gap for one–layer LSA under a broad linear–stationary model. The strict gap requires only independence of innovations and finite moments; with absolutely summable autocovariances we also recover the $1/n$ first–order expansion.

**Model.**  Let $\{x_t\}_{t\in\mathbb{Z}}$ be a zero–mean *linear stationary* process with Wold representation

$$x_t \;=\; \sum_{k\geq 0}\psi_k\,\varepsilon_{t-k}, \qquad \sum_{k\geq 0}|\psi_k| < \infty, \tag{18}$$

where $\{\varepsilon_t\}$ are i.i.d. with $\mathbb{E}[\varepsilon_t]=0$, $\mathbb{E}[\varepsilon_t^2]=\sigma_\varepsilon^2>0$, and with a symmetric distribution. We assume $\mathbb{E}[\varepsilon_t^4]<\infty$ and $\mathrm{Var}(\varepsilon_t^2)<\infty$. Let $\gamma_h := \mathbb{E}[x_t x_{t+h}]$, so $\sum_{h\in\mathbb{Z}}|\gamma_h|<\infty$.

All Hankel/masking notation follows Section F.2:

$$G \;=\; \tfrac{1}{n}\sum_{m=1}^{n-p} x^{(m)}x^{(m)\top}, \quad x := x_{\,n-p+1:n}\in\mathbb{R}^p, \quad y := x_{n+1},$$

with the one–layer LSA predictor

$$\widehat{x}_{n+1}^{\mathrm{LSA}} \;=\; b^\top G A x, \qquad b\in\mathbb{R}^{p+1},\ A\in\mathbb{R}^{(p+1)\times p}.$$

As in Section F.2, introduce

$$Z := (\mathrm{vech}\,G)\otimes x, \qquad \widetilde{S} := \mathbb{E}[ZZ^\top], \quad \widetilde{r} := \mathbb{E}[Zx^\top], \quad \Gamma_p := \mathbb{E}[xx^\top].$$

**Linear regression predictor.**  For consistency with the Hankel construction, we restrict attention to the last $p$ lags $x := x_{\,n-p+1:n}\in\mathbb{R}^p$ as regression covariates. The best linear predictor of $y := x_{n+1}$ from $x$ is

$$y = w^{*\top}x + e_{n+1}, \qquad \mathbb{E}[x\,e_{n+1}]=0,$$

where $w^*\in\mathbb{R}^p$ is the least–squares coefficient vector and $e_{n+1}$ is the linear prediction error. We denote the corresponding fitted value by

$$\widehat{x}_{n+1}^{\mathrm{LR}} := w^{*\top}x.$$

**Lemma H.1** (Strict covariance of $(\mathrm{vech}\,G, x)$ without Gaussianity). *Let $g := \mathrm{vech}\,G$ and $x := x_{\,n-p+1:n}$. Under (18), $\mathrm{Cov}\big([\,g^\top,\,x^\top\,]^\top\big) \succ 0$ for every finite $n$.*

*Proof.* Let $Z_0 = [\,g^\top,\,x^\top\,]^\top$. Suppose $\mathrm{Var}(v^\top Z_0) = 0$ for some nonzero $v$. Traverse the tableau of Hankel–Gram sums from bottom (time $n$) upward. Collect coefficients of $\{x_n^2,\,x_{n-1}x_n,\ldots,x_{n-p}x_n,\,x_n\}$ to write

$$v^\top Z_0 \;=\; U + V\,x_n + W\,x_n^2,$$

with $U,V,W$ measurable w.r.t. $\mathcal{F}_{n-1} := \sigma(\varepsilon_s : s\leq n-1)$.

Write $x_n = \xi + \psi_0\varepsilon_n$, where $\xi$ is $\mathcal{F}_{n-1}$-measurable and $\varepsilon_n\perp\mathcal{F}_{n-1}$. By symmetry, $\mathrm{Cov}(\varepsilon_n,\varepsilon_n^2)=0$. Thus

$$\mathrm{Var}(v^\top Z_0 \mid \mathcal{F}_{n-1}) = \mathrm{Var}(V\psi_0\varepsilon_n + W\psi_0^2\varepsilon_n^2 \mid \mathcal{F}_{n-1}) = (V\psi_0)^2\sigma_\varepsilon^2 + W^2\psi_0^4\,\mathrm{Var}(\varepsilon^2).$$

Since both coefficients are strictly positive, this vanishes only if $V = W = 0$. Inductively repeating the elimination for $x_{n-1}, x_{n-2}, \ldots$ forces $v = 0$, a contradiction. Hence the covariance is strictly positive definite. □

**Lemma H.2** (Kronecker lift remains strictly PD). *With $Z = (\mathrm{vech}\,G)\otimes x$ and $x = x_{\,n-p+1:n}$,*

$$\Sigma^\otimes := \mathbb{E}\begin{bmatrix} Z \\ x \end{bmatrix}\begin{bmatrix} Z \\ x \end{bmatrix}^\top = \begin{bmatrix} \widetilde{S} & \widetilde{r} \\ \widetilde{r}^\top & \Gamma_p \end{bmatrix} \;\succ\; 0.$$

*Proof.* By Lemma H.1, $\mathrm{Cov}([g, x]) \succ 0$; hence the conditional covariance $\mathrm{Cov}(g \mid x) = \mathrm{Cov}(g) - \mathrm{Cov}(g, x)\Gamma_p^{-1}\mathrm{Cov}(x, g) \succ 0$. For any nonzero $u = \mathrm{vec}(U) \in \mathbb{R}^{qp}$ and $w \in \mathbb{R}^p$, put $Y := u^\top Z + w^\top x = x^\top (Ug + w)$. Then $\mathrm{Var}(Y) \geq \mathbb{E}[x^\top U \, \mathrm{Cov}(g \mid x) \, U^\top x] > 0$ unless $U = 0$; if $U = 0$ then $\mathrm{Var}(Y) = \mathrm{Var}(w^\top x) > 0$ since $\Gamma_p \succ 0$. Thus $\Sigma^\otimes \succ 0$. $\square$

**Theorem H.3** (Strict finite–sample gap under linear stationarity). *Under the model above, for every finite $n$,*

$$\min_{b, A} \; \mathbb{E}\left[(\widehat{x}_{n+1}^{\mathrm{LSA}} - \widehat{x}_{n+1}^{\mathrm{LR}})^2\right] \;=\; w^{*\top}\Delta_n w^*, \qquad \Delta_n \succ 0.$$

*Proof.* As in (5), the LSA risk relative to the regression predictor can be written

$$\begin{aligned}
\mathcal{L}(\eta) &= \mathbb{E}\left[(\widehat{x}_{n+1}^{\mathrm{LSA}} - \widehat{x}_{n+1}^{\mathrm{LR}})^2\right] \\
&= w^{*\top}\Gamma_p w^* + \eta^\top \widetilde{S}\, \eta - 2\, \eta^\top \widetilde{r}\, w^*.
\end{aligned}$$

The minimizer is $\eta^* = \widetilde{S}^{-1}\widetilde{r}\, w^*$, giving the optimal value

$$w^{*\top}(\Gamma_p - \widetilde{r}^\top \widetilde{S}^{-1}\widetilde{r})w^* = w^{*\top}\Delta_n w^*.$$

By Lemma H.2, $\widetilde{S} \succ 0$, hence $\Delta_n \succ 0$ by the Schur complement. $\square$

**Remarks.** (i) The gap above is defined relative to the best linear regression predictor $\widehat{x}_{n+1}^{\mathrm{LR}}$. (ii) If in addition the regression residual $e_{n+1}$ is independent of $(G, x)$ (as in an exact AR($p$) model with $p$ lags), then the same gap translates directly to a strict excess risk gap relative to $y$.

**Cumulant identities for linear processes.** Let $\kappa_r := \mathrm{cum}_r(\varepsilon_0)$ denote the order–$r$ cumulant of $\varepsilon_0$ (finite for the orders we use). For any $t_1, \ldots, t_r$,

$$\mathrm{cum}(x_{t_1}, \ldots, x_{t_r}) = \kappa_r \sum_{u \in \mathbb{Z}} \psi_{t_1 - u} \cdots \psi_{t_r - u}. \tag{19}$$

This follows from multilinearity of cumulants and independence of $\{\varepsilon_t\}$ (see, e.g., (Brillinger, 2001, Theorem 2.3.2))

**Lemma H.4** (Moment expansions via cumulants). *Assume $\sum_{h \in \mathbb{Z}} |\gamma_h| < \infty$, $\mathbb{E}|\varepsilon_t|^6 < \infty$ (so $\kappa_4, \kappa_6$ are finite). Let $u := \mathrm{vech}(\Gamma_{p+1})$ and set $S_\infty := (uu^\top) \otimes \Gamma_p$, $r_\infty := u \otimes \Gamma_p$. For $n \to \infty$ with fixed $p$,*

$$\widetilde{S}_n \;=\; S_\infty + \frac{1}{n}\, C_S + o(1/n), \qquad \widetilde{r}_n \;=\; r_\infty + \frac{1}{n}\, C_r + o(1/n), \tag{20}$$

*for finite matrices $C_S, C_r$ determined by $\{\gamma_h\}$ and $\kappa_4, \kappa_6$.*

*Proof.* We work entrywise. Write indices $i, j, k, \ell \in \{1, \ldots, p+1\}$, $s, t \in \{1, \ldots, p\}$, and

$$a = x_{m+i-1}, \; b = x_{m+j-1}, \; c = x_{m'+k-1}, \; d = x_{m'+\ell-1}, \; e = x_{n-p+s}, \; f = x_{n-p+t}.$$

Recall $G_{ij} = \frac{1}{n}\sum_{m=1}^{n-p} x_{m+i-1} x_{m+j-1}$. We analyze

$$(\widetilde{r}_n)_{(ij,s),t} = \mathbb{E}[G_{ij}\, x_s x_t] = \frac{1}{n}\sum_{m=1}^{n-p} \mathbb{E}[ab\, e\, f], \tag{21}$$

$$(\widetilde{S}_n)_{(ij,s),(k\ell,t)} = \mathbb{E}[G_{ij} G_{k\ell}\, x_s x_t] = \frac{1}{n^2}\sum_{m, m'=1}^{n-p} \mathbb{E}[ab\, cd\, e\, f]. \tag{22}$$

*(I) The $r_n$ expansion.* By the moment–cumulant formula (see Lemma H.6), for four variables,

$$\mathbb{E}[abef] = \mathbb{E}[ab]\mathbb{E}[ef] + \mathbb{E}[ae]\mathbb{E}[bf] + \mathbb{E}[af]\mathbb{E}[be] + \mathrm{cum}(a, b, e, f).$$

The pairwise terms equal

$$\gamma_{j-i}(\Gamma_p)_{st} + \gamma_{(n-p+s)-(m+i-1)}\gamma_{(n-p+t)-(m+j-1)} + \gamma_{(n-p+t)-(m+i-1)}\gamma_{(n-p+s)-(m+j-1)}.$$

Summing over $m$ gives

$$\frac{n-p}{n}\,\gamma_{j-i}(\Gamma_p)_{st} + \frac{1}{n}\sum_{k=1}^{n-p}\left(\gamma_{k+s-i}\gamma_{k+t-j} + \gamma_{k+t-i}\gamma_{k+s-j}\right).$$

Because $\sum_h |\gamma_h| < \infty$, Toeplitz summation (see Lemma F.16) implies that the convolutions are uniformly bounded and converge; thus

$$\frac{n-p}{n}\,\gamma_{j-i}(\Gamma_p)_{st} = \gamma_{j-i}(\Gamma_p)_{st} - \frac{p}{n}\gamma_{j-i}(\Gamma_p)_{st},$$

and

$$\frac{1}{n}\sum_{k=1}^{n-p}\left(\gamma_{k+s-i}\,\gamma_{k+t-j} + \gamma_{k+t-i}\,\gamma_{k+s-j}\right) = \frac{1}{n}\,c_{ij,st}^{(r,2)} + o(1/n),$$

for some absolutely convergent constant

$$c_{ij,st}^{(r,2)} := \sum_{k=1}^{\infty}\left(\gamma_{k+s-i}\,\gamma_{k+t-j} + \gamma_{k+t-i}\,\gamma_{k+s-j}\right).$$

For the fourth–order cumulant, (19) with $r = 4$ yields

$$\mathrm{cum}(a, b, e, f) = \kappa_4 \sum_{u\in\mathbb{Z}}\psi_{m+i-1-u}\psi_{m+j-1-u}\,\psi_{n-p+s-u}\psi_{n-p+t-u}.$$

Summing over $m$ (equivalently $k = n - p - m + 1$) and using

$$\sum_{k\geq 1}\sum_u |\psi_{i-u+k}\psi_{j-u+k}\psi_{s-u}\psi_{t-u}| \leq \|\psi\|_{\ell_1}^4 < \infty$$

gives

$$\frac{1}{n}\sum_{m=1}^{n-p}\mathrm{cum}(a, b, e, f) = \frac{1}{n}\,c_{ij,st}^{(r,4)} + o(1/n)$$

with an absolutely convergent constant $c_{ij,st}^{(r,4)} := \kappa_4 \sum_{k\geq 1}\sum_{u\in\mathbb{Z}}\psi_{i-u+k}\psi_{j-u+k}\psi_{s-u}\psi_{t-u}$. Collecting pieces and reorganizing in tensor form yields

$$r_n = (\mathrm{vec}\,\Gamma_{p+1})\otimes\Gamma_p + \frac{1}{n}C_r^{(\mathrm{vec})} + o(1/n),$$

and therefore $\widetilde{r}_n = (L_{p+1}\otimes I_p)\,r_n = r_\infty + \frac{1}{n}C_r + o(1/n)$.

*(II) The $S_n$ expansion.* For

$$S_n = \sum_{i,j,k,\ell}\sum_{s,t}\mathbb{E}\big[G_{ij}G_{k\ell}x_s x_t\big]\left((e_j\otimes e_i\otimes e_s)(e_\ell\otimes e_k\otimes e_t)^\top\right).$$

Then

$$\mathbb{E}\big[G_{ij}G_{k\ell}x_s x_t\big] = \frac{1}{n^2}\sum_{m=1}^{n-p}\sum_{m'=1}^{n-p}\mathbb{E}[abcdef].$$

**Sixth–order moment decomposition.** By the moment–cumulant formula in Lemma H.6,

$$\mathbb{E}[abcdef] = \sum_{P\in\mathfrak{M}_3}\prod_{(u,v)\in P}\mathbb{E}[uv] + \sum_{\pi\in\Pi_{4,2}}\mathrm{cum}_4\big(\pi^{(4)}\big)\,\mathbb{E}\big(\pi^{(2)}\big) + \mathrm{cum}_6(a, b, c, d, e, f), \quad (23)$$

where $\mathfrak{M}_3$ is the set of the 15 perfect matchings of $\{a, b, c, d, e, f\}$, $\Pi_{4,2}$ is the set of the 15 partitions into a 4–block and a 2–block, $\pi^{(4)}$ denotes the 4–tuple in the 4–block and $\pi^{(2)}$ the paired variables. Write $\gamma_h := \mathbb{E}[x_t x_{t+h}]$.

**(a) Triple–pairings.** The three pairings that keep $(e, f)$ together are

$$P_0 = \{(a,b),(c,d),(e,f)\}, \qquad P_1 = \{(a,c),(b,d),(e,f)\}, \qquad P_2 = \{(a,d),(b,c),(e,f)\}.$$

The leading pairing $P_0$ contributes

$$\frac{1}{n^2} \sum_{m,m'} \gamma_{j-i}\, \gamma_{\ell-k}\, (\Gamma_p)_{st} = \gamma_{j-i}\gamma_{\ell-k}(\Gamma_p)_{st} + \frac{1}{n}\, c_{ij,k\ell;st}^{(S,\mathrm{bd})} + o(1/n),$$

with

$$c_{ij,k\ell;st}^{(S,\mathrm{bd})} = -2p\, \gamma_{j-i}\, \gamma_{\ell-k}\, (\Gamma_p)_{st}. \tag{24}$$

The two additional pairings $P_1, P_2$ yield, after the change of variable $h = m - m'$ and Toeplitz summation (see Lemma F.16),

$$\frac{1}{n^2} \sum_{m,m'} \Big( \gamma_{i-k+h}\, \gamma_{j-\ell+h} + \gamma_{i-\ell+h}\, \gamma_{j-k+h} \Big)(\Gamma_p)_{st} = \frac{1}{n}\, c_{ij,k\ell;st}^{(S,0)} + o(1/n),$$

where

$$c_{ij,k\ell;st}^{(S,0)} := \sum_{h\in\mathbb{Z}} \Big( \gamma_{i-k+h}\, \gamma_{j-\ell+h} + \gamma_{i-\ell+h}\, \gamma_{j-k+h} \Big)(\Gamma_p)_{st} \quad \text{(absolutely convergent).} \tag{25}$$

Therefore, the total contribution of the three pairings with $(e, f)$ paired is

$$\gamma_{j-i}\gamma_{\ell-k}(\Gamma_p)_{st} + \frac{1}{n}\Big( c_{ij,k\ell;st}^{(S,\mathrm{bd})} + c_{ij,k\ell;st}^{(S,0)} \Big) + o(1/n).$$

All other 12 pairings necessarily contain at least one cross–pair between $\{a,b,c,d\}$ and $\{e,f\}$. After the change of variable $q := n - p - m + 1$ (or $q := n - p - m' + 1$ as appropriate) and absolute summability of $\{\gamma_h\}$, each such term equals $\frac{1}{n}$ times a finite constant plus $o(1/n)$. Collecting the 12 distinct cross–pairings (those where $e$ or $f$ pairs with one of $a, b, c, d$) gives the *explicit* constant via Toeplitz summation (see Lemma F.16)

$$c_{ij,k\ell;st}^{(S,2)} := \sum_{q=1}^{\infty} \Big[ \gamma_{s-i+q}\gamma_{t-j+q}\, \gamma_{\ell-k} + \gamma_{s-i+q}\gamma_{t-k}\, \gamma_{j-\ell+q} + \gamma_{s-i+q}\gamma_{t-\ell}\, \gamma_{j-k+q}$$

$$+ \gamma_{s-j+q}\gamma_{t-i+q}\, \gamma_{\ell-k} + \gamma_{s-j+q}\gamma_{t-k}\, \gamma_{i-\ell+q} + \gamma_{s-j+q}\gamma_{t-\ell}\, \gamma_{i-k+q}$$

$$+ \gamma_{t-i+q}\gamma_{s-j+q}\, \gamma_{\ell-k} + \gamma_{t-i+q}\gamma_{s-k}\, \gamma_{j-\ell+q} + \gamma_{t-i+q}\gamma_{s-\ell}\, \gamma_{j-k+q}$$

$$+ \gamma_{t-j+q}\gamma_{s-i+q}\, \gamma_{\ell-k} + \gamma_{t-j+q}\gamma_{s-k}\, \gamma_{i-\ell+q} + \gamma_{t-j+q}\gamma_{s-\ell}\, \gamma_{i-k+q} \Big].$$

Therefore the total contribution of triple–pairings is

$$\gamma_{j-i}\gamma_{\ell-k}(\Gamma_p)_{st} + \frac{1}{n}\, c_{ij,k\ell;st}^{(S,\mathrm{bd})} + \frac{1}{n}\, c_{ij,k\ell;st}^{(S,0)} + \frac{1}{n}\, c_{ij,k\ell;st}^{(S,2)} + o(1/n). \tag{26}$$

**(b) $\{4,2\}$ partitions.** For linear processes $x_t = \sum_{r\in\mathbb{Z}} \psi_{t-r}\varepsilon_r$ with i.i.d. innovations, the fourth cumulant satisfies $\mathrm{cum}_4(x_{t_1}, x_{t_2}, x_{t_3}, x_{t_4}) = \kappa_4 \sum_{r\in\mathbb{Z}} \psi_{t_1-r}\psi_{t_2-r}\psi_{t_3-r}\psi_{t_4-r}$, where $\kappa_4 = \mathrm{cum}_4(\varepsilon)$ and $\sum_k |\psi_k| < \infty$. Define the absolutely convergent series

$$S_{ab}(r) := \sum_{q\in\mathbb{Z}} \psi_{q+i-1-r}\, \psi_{q+j-1-r}, \qquad S_{cd}(r) := \sum_{q\in\mathbb{Z}} \psi_{q+k-1-r}\, \psi_{q+\ell-1-r}.$$

By stationarity and Toeplitz summation, only the three families of partitions in which the 4-block contains either $\{a,b\}$ or $\{c,d\}$ contribute at order $1/n$; all other $\{4,2\}$ partitions are $o(1/n)$. The non–vanishing $1/n$ constants are

$$c_{ij,k\ell;st}^{(S,4)} := \kappa_4 \sum_{r\in\mathbb{Z}} \Big[ \gamma_{t-s}\, S_{ab}(r)\, S_{cd}(r) \;+\; \gamma_{\ell-k}\, S_{ab}(r)\, \psi_{s-r}\psi_{t-r}$$

$$+ \gamma_{j-i}\, S_{cd}(r)\, \psi_{s-r}\psi_{t-r} \Big]. \tag{27}$$

**(c) Sixth–order cumulant.** Using $\mathrm{cum}_6(x_{t_1}, \dots, x_{t_6}) = \kappa_6 \sum_{r \in \mathbb{Z}} \prod_{u=1}^{6} \psi_{t_u - r}$ with $\kappa_6 = \mathrm{cum}_6(\varepsilon)$, the double sum over $(m, m')$ reduces (by Toeplitz summation) to

$$\frac{1}{n} c_{ij,k\ell;st}^{(S,6)} + o(1/n), \qquad c_{ij,k\ell;st}^{(S,6)} := \kappa_6 \sum_{r \in \mathbb{Z}} S_{ab}(r) \, S_{cd}(r) \, \psi_{s-r} \psi_{t-r}. \tag{28}$$

**(d) Collecting the pieces.** Combining (26), (27) and (28) in (23) yields

$$\mathbb{E}\big[G_{ij} G_{k\ell} x_s x_t\big] = \gamma_{j-i} \, \gamma_{\ell-k} \, (\Gamma_p)_{st} + \frac{1}{n}\Big(c_{ij,k\ell;st}^{(S,\mathrm{bd})} + c_{ij,k\ell;st}^{(S,0)} + c_{ij,k\ell;st}^{(S,2)} + c_{ij,k\ell;st}^{(S,4)} + c_{ij,k\ell;st}^{(S,6)}\Big) + o(1/n).$$

Therefore

$$S_n = \big(\mathrm{vec}\,\Gamma_{p+1}\big)\big(\mathrm{vec}\,\Gamma_{p+1}\big)^\top \otimes \Gamma_p + \frac{1}{n}\, C_S^{(\mathrm{vec})} + o(1/n),$$

with the explicit block

$$C_S^{(\mathrm{vec})} := \sum_{i,j,k,\ell} \sum_{s,t} \Big(c_{ij,k\ell;st}^{(S,\mathrm{bd})} + c_{ij,k\ell;st}^{(S,0)} + c_{ij,k\ell;st}^{(S,2)} + c_{ij,k\ell;st}^{(S,4)} + c_{ij,k\ell;st}^{(S,6)}\Big) \Big((e_j \otimes e_i \otimes e_s)\,(e_\ell \otimes e_k \otimes e_t)^\top\Big).$$

Finally, since $\widetilde{S}_n = (L_{p+1} \otimes I_p)\, S_n\, (L_{p+1} \otimes I_p)^\top$, we obtain $\widetilde{S}_n = S_\infty + \frac{1}{n} C_S + o(1/n)$, where $S_\infty = (uu^\top) \otimes \Gamma_p$ and $C_S = (L_{p+1} \otimes I_p)\, C_S^{(\mathrm{vec})} \,(L_{p+1} \otimes I_p)^\top$. Absolute summability of $\{\gamma_h\}$ and $\{\psi_k\}$, and finiteness of $\kappa_4, \kappa_6$, ensure that all series above converge absolutely and justify the $o(1/n)$ remainder. $\qquad\square$

**Theorem H.5** (Order of the non–Gaussian gap)**.** *Under Lemma H.4, let $Q = [u/\|u\|, Q_\perp]$ and $P := Q \otimes I_p$. Then the block–inverse expansion of Lemma F.13 applies verbatim, and*

$$\Delta_n = \Gamma_p - \widetilde{r}_n^\top \widetilde{S}_n^{-1} \widetilde{r}_n = \frac{1}{n} B_p + o(1/n),$$

*with $B_p \succeq 0$ given by the same closed form as in (9) after replacing the Gaussian $C_S, C_r$ by those from Lemma H.4. Generically $B_p \succ 0$.*

**Remarks.** (i) *No Gaussianity is needed* for strict PD and the positive Schur–complement gap: independence of innovations with finite fourth moment (and $\mathrm{Var}(\varepsilon^2) > 0$) suffices. (ii) If in addition $\sum_h |\gamma_h| < \infty$ and $\mathbb{E}|\varepsilon|^6 < \infty$, the exact $1/n$ order persists for general linear stationary processes; Gaussianity only simplifies the constants via Wick pairings.

**Discussion on AR/MA/ARMA models.** Stable AR, MA, and ARMA processes satisfy the assumptions above, so Theorems H.3 and H.5 apply directly. Nevertheless, caution is warranted in interpreting the result. For MA or ARMA models, the one–step prediction error $e_{n+1}$ still carries dependence on portions of the past beyond the last $p$ lags. This prevents a direct characterization of the mean–squared error gap between LSA and linear regression with respect to the true target $x_{n+1}$. Hence the finite–lag linear regression predictor $\widehat{x}_{n+1}^{\mathrm{LR}}$ does not coincide with the globally optimal (infinite–order) linear predictor. In particular, for MA models there may exist richer linear predictors that exploit the entire past more effectively. Our analysis should therefore be understood not as a claim of global optimality across all linear predictors, but rather as an insight into the structural gap that persists even when comparing LSA against the natural $p$–lag linear regression benchmark.

AUXILIARY LEMMAS USED IN APPENDIX H

**Lemma H.6** (Moment–cumulant formula)**.** *For random variables $X_1, \dots, X_r$ with finite moments up to order $r$, the joint moment can be expressed in terms of cumulants as*

$$\mathbb{E}\left[\prod_{i=1}^{r} X_i\right] = \sum_{\pi \in \mathcal{P}_r} \prod_{B \in \pi} \mathrm{cum}\big(X_j : j \in B\big),$$

*where $\mathcal{P}_r$ denotes the set of all partitions of $\{1, \dots, r\}$, and $\mathrm{cum}(\cdot)$ denotes the joint cumulant.*

