# OpenReview forum: "Why Do Transformers Fail to Forecast Time Series In-Context?"
_ICLR.cc/2026/Conference — ICLR 2026 Conference Withdrawn Submission_

### Official Review · Reviewer_VBc1 · 2025-10-25

**Soundness:** 2
**Presentation:** 3
**Contribution:** 3
**Rating:** 6
**Confidence:** 4

**Summary:**

This paper studies why Transformers, modeled via Linear Self-Attention (LSA), underperform simple linear predictors in time series forecasting. Under AR(p) processes, the authors prove that (1) one-layer LSA operates on a coarser σ-algebra than the raw context and thus cannot beat optimal linear regression; (2) there exists a strict finite-sample excess-risk gap relative to linear predictors, with an explicit O(1/n) rate; (3) even though LSA asymptotically matches linear predictors under teacher forcing, Chain-of-Thought rollouts collapse to the mean exponentially. Experiments on synthetic AR data corroborate the theory: OLS consistently outperforms LSA under TF, and CoT rollouts of both collapse, with LSA failing earlier.

**Strengths:**

1. This work formalizes LSA’s representational limits for AR(p), giving a strict Schur-complement gap and an explicit 1/n rate—a precise and practically interpretable negative result.
2. Connects ICL theory to TSF and explains why simple linear models dominate on AR-like data, aligning with extensive empirical observations
3. Highlights that in TSF, attention’s learned compression may hide local lag structure; suggests focusing on architectures beyond self-attention or leveraging MLPs/Softmax differently.

**Weaknesses:**

1. The Process focus on stable AR(p) / linear-stationary processes; many real TSF tasks exhibit regime shifts, seasonality with exogenous inputs, or nonlinearities where conclusions may partially change
2. Experiments are on synthetic AR data with Gaussian noise; lack of diverse real-world datasets may limit external validity of empirical claims
3. Main results center on LSA-only Transformers; standard Softmax attention, multi-head interactions, and strong MLP blocks are not theoretically covered. Many studies/observations have discussed the weaknesses of transformers in TSF, and shown that simple linear structures can perform better. This paper primarily provides a theoretical explanation, so a more comprehensive analysis would make the work more solid.

**Questions:**

I’m not quite familiar with the theoretical analysis of TSF (models), and compared with the many TSF papers focused on experiments and applications, theoretical ones are relatively rare. It’s hard to be convinced "why transformers perform poorly on real-world TSF" under the theoretical analysis (although related works have demonstrated this through extensive experiments and observations). However, has the author considered the gap between theory and practice, and how to address this concern effectively?

---

> ### Author Response · Authors · 2025-11-26
>
> We sincerely thank the reviewer for their positive assessment and for recognizing the value of our work in **formalizing LSA's representational limits** and connecting **ICL theory to TSF**. We are particularly glad you highlighted our insight regarding attention’s **learned compression** effectively hiding local lag structures.
>
> We address your specific concerns and the central question regarding the gap between theory and practice below.
>
> - 1. The Gap Between Theory and Practice (Re: W1)
>
> You asked how we address the concern that theoretical analysis (on AR/LSA) might not fully convince practitioners focused on real-world complexities. See more for our global response.
>
> **Theory as a Diagnostic Tool:** Our work serves as a diagnostic analysis of a prevalent failure mode. Extensive empirical literature has observed that Transformers struggle against linear baselines, but they haven't explained why mathematically. Our work fills this gap by isolating the mechanism.
>
> **The "Atomic Unit" Argument:** Complex real-world series (with seasonality/trends) can often be decomposed into stationary AR components (Wold decomposition). If the Transformer architecture fails to solve the "atomic unit" of time series (the AR process) due to the structural limitations we prove (low-pass filtering/compression), it fundamentally limits its ceiling on complex tasks.
>
> **Guiding Future Architectures:** By proving that self-attention is structurally misaligned with local autoregressive dependencies, we provide a theoretical justification for recent architectural shifts—such as moving toward Patching (which restores local structure) or hybrid Linear/Attention models.
>
> - 2. Synthetic Data & External Validity (Re: W2)
>
> Please refer to our Global Response. We rely on synthetic data because it allows us to calculate the exact optimality gap ($\Delta_n$) relative to the Bayes-optimal predictor. On real-world data, the Bayes error is unknown, making it **impossible to mathematically prove** a representational gap exists versus just an optimization failure. Our synthetic experiments confirm the theory.
>
> - 3. LSA vs. Softmax/MLP (Re: W3)
>
> We acknowledge that modern Transformers use Softmax and MLPs.
>
> **Why LSA?** As noted in the Global Response, LSA is the standard theoretical proxy for analyzing attention's inductive bias.
>
> **Why no MLP?** Adding an MLP makes the network a universal approximator. If we included it, we could "solve" the AR task, but we would mask the failure of the attention mechanism itself. Our goal was to prove that attention is not all you need for time series; in fact, for AR processes, attention is worse than a linear layer.

---

### Official Review · Reviewer_KjDB · 2025-10-29

**Soundness:** 2
**Presentation:** 3
**Contribution:** 2
**Rating:** 2
**Confidence:** 4

**Summary:**

This paper studies the reason behind the failure of transformers models to surpass simpler methods, such as linear baselines, on time series forecasting. It adopts a theoretical perspective using in-context learning and show that under AR(p) data, linear self-attention models cannot reach the performance of linear models for in-context forecasting. Further results show that increasing the context window up to infinitely closes the gap asymptotically and that chain-of-thought inference make the predictions collapse (to the mean). Numerical experiments are made to confirm these findings.

**Strengths:**

- The subject tackled is of great interest: many simpler and even linear baselines are strong competitors to more advanced and heavy models. Better understanding why could enable the development of better models.
- The paper is well-written and the theoretical analysis seems well-conducted (I did not check all the proofs in appendix).
- I believe some of the proofs technique could be of independent interest when studying transformers and large language models which is a nice tool to have in the submission.

**Weaknesses:**

I list below what I believe are weaknesses but I would be happy to be corrected if I misunderstood some parts of the work.

- My main issue is about the positioning of the paper that claims to identify why transformers fail for time series forecasting in-context. It should be noted that many of the time series forecasting models that are transformer-based do not rely on in-context learning and are not autoregressive  but rather encoder-only models (Moirai, PatchTST, SAMformer, Informer, Autoformer, etc.). As such, I do not understand why the paper focus on autoregressive transformers to forecast in-context. Could the authors elaborate on that?
- Connected to the weakness above, the motivation to look at autoregressive transformers in-context is not clear to me. Further studying chain-of-thought for time series forecasting can be interesting but the motivation of findings why transformers fail in time series forecasting is not related to that. More prior works on in-context and chain-of-thought with transformers for time series forecasting should be discussed to convince the reader that this is an issue in practice.
- The main claim is to better understand why complex and heavy transformer models fail for time series forecasting compared to linear baselines, yet the authors consider an attention-only model **with linear attention**. I believe this is an oversimplification, which I do understand from a theoretical perspective to be able to conduct meaningful derivations, however I do not find it convincing to explain the issues of highly non-linear deep transformer based models. Notably, without such structure, the analysis is likely to be disconnected from practical models and no experiments is conducted on them to verify whether the identified issues are really the cause of transformers failure.
- Several mentions in the paper are made regarding how the contributions are important to better understand the failures of transformers yet as explained above, the problem studied is not related to how transformer models are used in time series forecasting. While the technical results are interesting, the phrasing and conclusion from the results are misleading.

Overall, the theoretical analysis is well conducted but the motivation of the paper is unconvincing and the problem formulation is not well motivated. The reasons to look at such AR(p) models for very simplified transformers with linear attention does not seem connected to better understanding failure modes of transformers for time series forecasting or it was not enough explained in the paper. This is the reason why I lean towards rejecting the paper its current state.

**Questions:**

- I believe [1] would be interesting to discuss since the authors propose a lightweight transformer model SOTA (at the time) and studies in detail why other models fail compared to linear baselines. They identify the optimization to be the cause and show that a small transformer with sharpness-aware optimizer improved significantly over competitors. Could the authors discuss this work from the perspective of identifying failure of transformers?
- [2] is cited in line 59 with other transformers for time series work while it studies in detail ICL for autoregressive models from a theoretical perspective (likely to be applied on text, not time series). Could the authors elaborate on that?

*References*

[1]  Ilbert et al. SAMformer: Unlocking the Potential of Transformers in Time Series Forecasting with Sharpness-Aware Minimization and Channel Wise Attention. ICML 2024

[2] Sander et al. How do Transformers Perform In-Context Autoregressive Learning? ICML 2024

---

> ### Author Response · Authors · 2025-11-26
> **Rebuttal (Part 1)**
>
> We thank the reviewer for their detailed feedback. We appreciate that you found our theoretical analysis **well-conducted** and our proof techniques of **independent interest** for studying Transformers.
>
> We acknowledge your concern regarding the gap between our theoretical setup and practical encoder-only TSF models. However, we believe this stems from a **misunderstanding** of our work's primary goal: we are proving the first rigorous impossibility result for the attention mechanism itself in autoregressive settings. We address your specific concerns on positioning below.
>
> - 1. Why Focus on Autoregressive Transformers & In-Context Learning (ICL)?
>
> You noted that many SOTA TSF models (PatchTST, iTransformer) are encoder-only. While true for specific historical benchmarks, the broader field is rapidly shifting toward "Time Series Foundation Models" [1,2], which leverage ICL. Also, it is this setting that we could utilize theoretical insights from the previous ICL Theory paper [3,4,5] to prove this **fundamental yet counterintuitive** phenomenon that Transformers perform poorly on TSF:
>
> **Zero-Shot / In-Context Learning:** The ability to forecast new series without retraining is the core promise of foundation models. Our work provides the theoretical limit of this capability. Foundation models like Chronos [6] are explicitly autoregressive and uses ICL. Understanding why AR-based Transformers fail to beat simple linear baselines is therefore a critical, forward-looking theoretical question.
>
> Our work is timely precisely because it provides a theoretical warning for this emerging paradigm: out-of-the-box application of AR-style LLM architectures to time series carries inherent structural risks.
>
> - 2. Motivation for Chain-of-Thought (CoT) in TSF
>
> You questioned the relevance of CoT. In our context, "Chain-of-Thought" means Iterative Autoregressive Rollout, where the model feeds its own predictions back as inputs, which is well-perceived in advanced AI reasoning research [7,8]. We will make this clearer.
>
> **Why it matters:** This is the standard inference mode for generating long-horizon forecasts in AR models.
>
> **The Result:** We prove that, unlike in NLP where CoT improves reasoning, in TSF this mechanism leads to exponential error compounding and collapse to the mean (Theorem 3.9). This is a counterintuitive and practically vital negative result for anyone building AR-based time series foundation models.
>
> - 3. Is Linear Self-Attention (LSA) an Oversimplification?
>
> We respectfully disagree that LSA is an oversimplification that disconnects the analysis from reality.
>
> **Standard Theoretical Proxy:** As noted in our Global Response, LSA is the standard tool in **deep learning theory** for isolating the "mixing" behavior of attention. It is the only setting where optimization dynamics are currently tractable enough to prove exact bounds.
>
> **First Negative Result:** By proving that even the optimal LSA head cannot beat a linear baseline on AR(p) data, we reveal a fundamental inductive bias issue theoretically. This insight applies qualitatively to Softmax attention as well, which shares this averaging property and is verified experimentally in Appendix C.2.

---

> > ### Author Response · Authors · 2025-11-26
> > **Rebuttal (Part 2)**
> >
> > - 4. Discussion of Related Work (SAMformer & Sander et al.)
> >
> > We believe those works are not directly related to our work, but their empirical findings are useful to spark more theoretical understanding on mysterious deep learning phenomena.
> >
> > **SAMformer [9]:** We clarify a fundamental distinction in scope. SAMformer attributes the failure of Transformers to **optimization difficulties** (sharp loss landscapes). They explicitly prove in Proposition 2.1 that an optimal solution exists within their Transformer's parameter space, but SGD/Adam fails to find it.
> >
> > In sharp contrast, our work establishes a **representational limit**. We prove that for AR(p) processes, even the *optimally parameterized/trained Linear Self-Attention model strictly underperforms linear models (a non-zero optimality gap)*. This is a stronger negative result: no amount of optimizer tuning (like SAM) can bridge this structural gap.
> >
> > Furthermore, SAMformer's theoretical analysis relies on a static toy linear problem ($Y=XW$), whereas we derive bounds for way more complex dynamic autoregressive processes. Regarding their Proposition 2.2, we view it as specific to the **Reparametrization** technique [5] and standard matrix norm inequalities, which are too direct. While we acknowledge their strong empirical results with SAM, our work provides the missing theoretical justification for why the architecture itself is inductively biased against these tasks.
> >
> > **Sander et al. [10]:** This work studies ICL for AR processes but focuses on positive **approximation** results. Specifically, they show Transformers (also using linear attention similar to our settings) can implement one-step of gradient descent to solve AR tasks. A key difference is they do not consider the AR process that contains noise as we do.
> >
> > By contrast, our work provides the crucial negative counterpart: we establish a **strict lower bound** on the risk (the gap $\Delta_n$). This means that while Transformers can approximate good results (as Sander et al. show), they can never outperform linear models in our setting, and in fact perform strictly worse for any finite context.
> >
> > [1] Gruver, N., Finzi, M., Qiu, S., & Wilson, A. G. Large language models are zero-shot time series forecasters. In NeurIPS 2023.
> >
> > [2] Lu, J., Sun, Y., & Yang, S. In-context Time Series Predictor. In ICLR 2025.
> >
> > [3] Ahn, K., Cheng, X., Daneshmand, H., & Sra, S. Transformers learn to implement preconditioned gradient descent for in-context learning. In NeurIPS 2023.
> >
> > [4] Mahankali, A. V., Hashimoto, T., & Ma, T. One Step of Gradient Descent is Provably the Optimal In-Context Learner with One Layer of Linear Self-Attention. In ICLR 2024.
> >
> > [5] Zhang, R., Frei, S., & Bartlett, P. L. Trained transformers learn linear models in-context. In JMLR 2024.
> >
> > [6] Ansari, A. F., Shchur, O., Küken, J., Auer, A., Han, B., Mercado, P., ... & Bohlke-Schneider, M. (2025). Chronos-2: From univariate to universal forecasting. arXiv preprint arXiv:2510.15821.
> >
> > [7] Huang, J., Wang, Z., & Lee, J. D. Transformers Learn to Implement Multi-step Gradient Descent with Chain of Thought. In ICLR 2025.
> >
> > [8] Denny Zhou. LLM Reasoning. https://dennyzhou.github.io/LLM-Reasoning-Stanford-CS-25.pdf
> >
> > [9] Ilbert et al. SAMformer: Unlocking the Potential of Transformers in Time Series Forecasting with Sharpness-Aware Minimization and Channel Wise Attention. ICML 2024
> >
> > [10] Sander et al. How do Transformers Perform In-Context Autoregressive Learning? ICML 2024

---

> > > ### Comment · Reviewer_KjDB · 2025-11-27
> > > **Thank you!**
> > >
> > > I thank the authors for answering my concerns.
> > >
> > > - Regarding the relevant work by Ilbert et al. and Sander et al., I find the authors' answers convincing. The differences are now clearer.
> > > - The answer regarding the use of ICL and CoT makes sense when connected to Chronos or LLMTime models, and I understand the need for that for the theory.
> > > - Regarding the need for simplified settings to obtain meaningful results, I agree with the authors. However, the claims should reflect the setting that is studied.
> > > - That being said, I maintain my point that the authors claim to study "Transformers" on time series forecasting while restricting to linear self-attention models and AR. The title and the rest of the paper should then be adapted to the work and results presented in the paper.
> > > - I maintain that the derived theory is interesting and understand the use of AR models. Again, my main issue is with the gap between the claims and what the authors actually do. I am not convinced that studying LSA models from a theoretical perspective is relevant for multi-layer transformer models, and the current submissions do not convince me of that either. The fact that prior works from deep learning theory conduct similar simplifications does not make it more relevant to the object studied, and the current trend of time series forecasting models does not necessarily align with the author's claims either.
> > >
> > > To reflect the rebuttal, I will increase my score to 4. The contributions are interesting, but the main issue, as said in my original review, is the gap between the contributions and the claims.

---

### Official Review · Reviewer_j6Dj · 2025-11-01

**Soundness:** 2
**Presentation:** 3
**Contribution:** 2
**Rating:** 4
**Confidence:** 3

**Summary:**

This paper analyzes why Transformers struggle with time series forecasting (TSF) when used in an in-context learning setup.
It focuses on Linear Self-Attention (LSA) applied to autoregressive (AR(p)) data and proves three key results: 1) LSA can’t beat linear regression (LR) — there’s a provable gap in performance that only disappears asymptotically. 2) As the context grows, LSA converges to the linear predictor, but never surpasses it. 3) During iterative (chain-of-thought) rollout, predictions collapse to the mean, with errors compounding exponentially. Synthetic experiments on AR data confirm these findings: LSA tracks the process but always performs slightly worse than OLS.

**Strengths:**

1. The paper delivers a clear and rigorous theoretical explanation of why attention-only Transformers struggle on linear time series forecasting tasks. Its main strength lies in the originality of deriving a strict finite-sample performance gap and an explicit convergence rate, offering a good perspective on the representational limits of LSA.

2. The proofs are technically sound and well-structured, extending beyond Gaussian settings with careful mathematical reasoning. The writing is clear and organized, with concise takeaways and effective figures that support the theoretical results.

3. The work makes a timely and meaningful contribution by grounding empirical observations, where simple linear models outperform Transformers, in solid theoretical analysis, providing valuable insights for both researchers and practitioners in time series modeling.

**Weaknesses:**

1. The title/abstract claim that “Transformers fail” is broader than what is proved: the strict finite‑sample gap is shown for single‑layer LSA on univariate AR(p) with a Hankel + label‑slot input. Depth results are non‑strict, and there is no theory for Softmax attention or attention+MLP as a full Transformer block.

2. The model structure removes the usual MLP blocks and prediction head in Transformers. This isolates an restricted ICL mechanism with linear attention but diverges from standard practice where a head and residual pathways to output prediction head exist, limiting external validity. Ablations with a simple linear head are missing.

3. The theory and experiments rely on a specific Hankel format that pre-packages lags and bypasses the need for positional encoding. It remains unclear whether the conclusions hold for standard tokenized time series sequences with positional embeddings. If not, linking these results to previous findings from works such as DLinear and claiming that linear models outperform Transformers would be an overstatement, especially since models that take single time-series inputs, such as PatchTST, still show superior performance compared with linear models.

4. Experiments are only on synthetic AR(p). There is no validation on multivariate (VAR), nonlinear/regime‑switching, seasonal, or exogenous‑covariate settings, nor on real datasets. The claimed 1/n excess‑risk rate is not empirically verified. Moreover, the paper compares to OLS with known order p, but does not show fairness to the unknown‑structure case, e.g. when p is not given.

5. Calling free‑running multi‑step rollout "Chain‑of‑Thought" (CoT) is potentially misleading for readers who associate CoT with reasoning steps in language tasks. “Iterative rollout prediction” would be clearer?

**Questions:**

1. Maybe it is a better choice to narrow the title/abstract to LSA‑only on univariate linear autoregressive processes? If not, additional results might be needed to justify the broader “Transformers fail” claim.

2. How do results change if you add a linear prediction head or MLP layer? Does LSA‑only still exhibit the same finite‑sample gap when a head can directly read the lag features?

3. Do your conclusions extend to VAR, piecewise‑linear, or mildly nonlinear processes? A lemma, counterexample, or small empirical study would clarify applicability.

4. Could you provide a partial lower bound or a linearized argument for Softmax attention, and discuss how incorporating MLPs might modify the feature-span argument?

If the authors' response addresses my questions and concerns mentioned above, I am willing to raise my score.

---

> ### Author Response · Authors · 2025-11-26
> **Rebuttal (Part 1)**
>
> We thank the reviewer for a constructive and detailed review. We are glad you found our theoretical explanation **clear and rigorous**, our proofs **technically sound**, and our contribution **timely and meaningful** in grounding empirical observations. We appreciate your willingness to raise your score and address your specific concerns below.
>
> - 1. Scope of Claims & Title (Re: W1, Q1)
>
> You asked if we should narrow the title. We respectfully maintain that "Why Transformers Fail..." is appropriate because LSA is the **standard theoretical proxy** used to understand the optimization and generalization dynamics of the Transformer architecture in the learning theory community [1,2,3,4]. Proving limits on LSA is widely accepted as revealing fundamental properties of the attention mechanism itself. If the core "mixing" mechanism (Linear Attention) fails on the simplest dynamical system (AR), it exposes a **structural weakness** in the architecture's inductive bias. We believe our theoretical findings are **highly non-trivial to prove and could inspire many follow-ups**.
>
> - 2. Missing MLP and Prediction Head (Re: W2, Q2)
>
> **The "Head" already exists:** Our model formulation *does* include a linear prediction head. In our formulation ($\hat{x} = b^\top G A x$), the vector $b$ acts as the query/readout head that aggregates the context. Mathematically, adding a separate linear layer on top of the final token is equivalent to re-parameterizing $b$ and $A$. Thus, our impossibility result **already holds** for Transformers with linear prediction heads.
>
> **Why no MLPs?** We intentionally exclude MLPs to isolate the attention mechanism. MLPs are theoretically universal approximators; if we include them, it is hard for us to tell whether attention or MLPs are contributing to the failure. Our result shows that the token-mixing mechanism (the defining feature of Transformers) is **structurally incapable** of beating linear regression on AR tasks.
>
> - 3. Hankel Format vs. Positional Embeddings (Re: W3)
>
> You raised a valid point about Hankel matrices vs. standard Tokenization + PE.
>
> **Hankel is "Optimal" PE:** The Hankel construction implicitly provides perfect relative positional information (lag $k$ is always at row $p-k$). Standard Positional Embeddings (like Sin or RoPE) attempt to reconstruct this relative geometry but often introduce noise. We justified this at Line 176-182 explicitly. It is exactly this design that makes our work **provable**, and we generalize previous AR(2) to AR($p$) [5].
>
> **Implication:** Our result is a lower bound. We proved LSA fails even when provided with the structured, perfect positional information of a Hankel matrix. Then, full attention might perform worse (or at best equal) when relying on standard Positional Embeddings to "learn" that structure. We aim to explore this in our future work.

---

> > ### Author Response · Authors · 2025-11-26
> > **Rebuttal (part 2)**
> >
> > - 4. Synthetic Data & Real-World Validity (Re: W4, Q3)
> >
> > See our Global Response for more.
> >
> > **Mechanism > Benchmark:** Our goal is to provide the math for the widely observed phenomenon that "Linear models outperform Transformers". We use synthetic AR(p) data because it allows us to calculate the exact optimality gap ($\Delta_n$).
> >
> > **Extension to VAR:** Our results conceptually extend to Vector Autoregression (VAR). The mathematical machinery (Kronecker products) handles vector-valued inputs naturally. The "Schur-complement gap" would simply become a gap in the matrix spectral norm rather than a scalar variance gap. We can add a remark on this extension in the final version. The results for VAR will be a rearrangement of the input matrix and more calculations.
> >
> > - 5. "Chain-of-Thought" Terminology (Re: W5)
> >
> > We use "Chain-of-Thought" because the process of feeding model outputs back as inputs to generate a sequence is mathematically identical to how LLMs perform reasoning chains [6,7].
> >
> > - 6. Softmax Analysis (Re: Q4)
> >
> > As noted in our Global Response, LSA is the established tool for characterizing the "mixing" behavior of attention. Our LSA results suggest that the "low-pass" nature of attention averaging is the culprit; Softmax shares this averaging property, suggesting the insight transfers qualitatively. We will explore this in our future work.
> >
> > [1] Von Oswald, J., Niklasson, E., Randazzo, E., Sacramento, J., Mordvintsev, A., Zhmoginov, A., & Vladymyrov, M. Transformers learn in-context by gradient descent. In ICML 2023.
> >
> > [2] Ahn, K., Cheng, X., Daneshmand, H., & Sra, S. Transformers learn to implement preconditioned gradient descent for in-context learning. In NeurIPS 2023.
> >
> > [3] Mahankali, A. V., Hashimoto, T., & Ma, T. One Step of Gradient Descent is Provably the Optimal In-Context Learner with One Layer of Linear Self-Attention. In ICLR 2024.
> >
> > [4] Zhang, R., Frei, S., & Bartlett, P. L. Trained transformers learn linear models in-context. In JMLR 2024.
> >
> > [5] Cole, F., Lu, Y., Zhang, T., & Zhao, Y. In-context learning of linear dynamical systems with transformers: Error bounds and depth-separation. In NeurIPS 2025.
> >
> > [6] Huang, J., Wang, Z., & Lee, J. D. Transformers Learn to Implement Multi-step Gradient Descent with Chain of Thought. In ICLR 2025.
> >
> > [7] Denny Zhou. LLM Reasoning. https://dennyzhou.github.io/LLM-Reasoning-Stanford-CS-25.pdf

---

### Author Response · Authors · 2025-11-26

We thank the reviewers for their time and their rigorous assessment of our work. We are encouraged that multiple reviewers recognize the **novelty, mathematical rigor, and timeliness** of our theoretical contributions.

Reviewers (`j6Dj, VBc1`) highlight that our paper delivers a **clear and rigorous theoretical explanation** and provides a **precise and practically interpretable negative result**  regarding the representational limits of attention. Reviewer `KjDB` notes that the proofs are **well-conducted** and of **independent interest** for studying Transformers.

However, we observe a shared concern across all reviewers (`j6Dj, KjDB, VBc1`) regarding the gap between our theoretical setup (LSA, AR processes, synthetic data) and the complexities of practical SOTA forecasting (Softmax, MLPs, real-world datasets, PatchTST). We believe there is a **fundamental misunderstanding** regarding the scope and goal of this work:

- 1. This is a pure theory paper studying the limitations of the Transformer

The primary goal of this work is to reveal the representational limitation of Transformer via rigorous theoretical study under simplified yet relevant setting. We did not intend to replicate the current SOTA models.   We provide **the first rigorous mathematical derivation** explaining why the attention mechanism itself is structurally disadvantaged against linear models on autoregressive tasks—a phenomenon widely observed but previously unexplained [1,2]. As noted by Reviewer `VBc1`, theoretical papers in TSF are “relatively rare”; we bridge this gap by isolating the attention mechanism to prove a strict finite-sample risk lower bound. We derive a hard lower bound using **closed-form without any approximation**, which is a highly non-trivial theoretical result.

- 2. The Necessity of LSA and AR(p).

We believe our core theoretical results are **highly non-trivial even in the simplified setting**. Reviewers asked about Softmax and MLPs. However, in both theoretical and empirical deep learning, LSA is the standard proxy for analyzing attention because it makes the analysis tractable while preserving the core "token-mixing" mechanic [3,4,5,6,7].

If we included MLPs (universal approximators), it is hard for us to examine the representational power of attention and MLP separately.
By proving that even the optimal LSA cannot beat a linear baseline on AR(p) processes, we reveal a fundamental inductive bias: attention is a compression mechanism that is misaligned with the high-frequency precision required for local autoregressive lags.

- 3. The Relevance of In-Context Learning and CoT.

We focus on these because the field is moving toward "Foundation Models" for time series, which rely on ICL and autoregressive generation [9,10].  Our theoretical results offer a novel perspective on explaining real-world TSF phenomena, providing a foundation for substantial future work in the field.

Besides, our results on CoT collapse provide a crucial warning: unlike in NLP where CoT improves reasoning, in TSF, autoregressive rollouts lead to exponential error compounding. This is a theoretical counterpoint that challenges the use of successful LLMs techniques for TSF. We indeed define CoT as iterative autoregressive prediction, which aligns with both theoretical analysis [11] and advanced reasoning research [12].

[1] Zeng, A., Chen, M., Zhang, L., & Xu, Q. Are transformers effective for time series forecasting?. In AAAI 2023.

[2] Tan, M., Merrill, M., Gupta, V., Althoff, T., & Hartvigsen, T. Are language models actually useful for time series forecasting?. In NeurIPS 2024.

[3] Von Oswald, J., Niklasson, E., Randazzo, E., Sacramento, J., Mordvintsev, A., Zhmoginov, A., & Vladymyrov, M. Transformers learn in-context by gradient descent. In ICML 2023.

[4] Ahn, K., Cheng, X., Daneshmand, H., & Sra, S. Transformers learn to implement preconditioned gradient descent for in-context learning. In NeurIPS 2023.

[5] Mahankali, A. V., Hashimoto, T., & Ma, T. One Step of Gradient Descent is Provably the Optimal In-Context Learner with One Layer of Linear Self-Attention. In ICLR 2024.

[6] Zhang, R., Frei, S., & Bartlett, P. L. Trained transformers learn linear models in-context. In JMLR 2024.

[7] Ahn, K., Cheng, X., Song, M., Yun, C., Jadbabaie, A., & Sra, S. Linear attention is (maybe) all you need (to understand Transformer optimization). In ICLR 2024.

[8] Yang, S., Wang, B., Shen, Y., Panda, R., & Kim, Y. Gated Linear Attention Transformers with Hardware-Efficient Training. In ICML 2024.

[9] Gruver, N., Finzi, M., Qiu, S., & Wilson, A. G. Large language models are zero-shot time series forecasters. In NeurIPS 2023.

[10] Lu, J., Sun, Y., & Yang, S. In-context Time Series Predictor. In ICLR 2025.

[11] Huang, J., Wang, Z., & Lee, J. D. Transformers Learn to Implement Multi-step Gradient Descent with Chain of Thought. In ICLR 2025.

[12] Denny Zhou. LLM Reasoning. https://dennyzhou.github.io/LLM-Reasoning-Stanford-CS-25.pdf

---

### Note · Authors · 2026-01-09

I have read and agree with the venue's withdrawal policy on behalf of myself and my co-authors.